# A high-quality bonobo genome refines the analysis of hominid evolution

Yafei Mao[1,12], Claudia R. Catacchio[2,12], LaDeana W. Hillier[1], David Porubsky[1], Ruiyang Li[1], Arvis Sulovari[1], Jason D. Fernandes[3], Francesco Montinaro[2,4], David S. Gordon[1,5], Jessica M. Storer[6], Marina Haukness[3], Ian T. Fiddes[3], Shwetha Canchi Murali[1,5], Philip C. Dishuck[1], PingHsun Hsieh[1], William T. Harvey[1], Peter A. Audano[1], Ludovica Mercuri[2], Ilaria Piccolo[2], Francesca Antonacci[2], Katherine M. Munson[1], Alexandra P. Lewis[1], Carl Baker[1], Jason G. Underwood[7], Kendra Hoekzema[1], Tzu-Hsueh Huang[1], Melanie Sorensen[1], Jerilyn A. Walker[8], Jinna Hoffman[9], Françoise Thibaud-Nissen[9], Sofie R. Salama[3,10], Andy W. C. Pang[11], Joyce Lee[11], Alex R. Hastie[11], Benedict Paten[3], Mark A. Batzer[8], Mark Diekhans[3], Mario Ventura[2 ✉] & Evan E. Eichler[1,5 ✉]

The divergence of chimpanzee and bonobo provides one of the few examples of recent hominid speciation[1,2]. Here we describe a fully annotated, high-quality bonobo genome assembly, which was constructed without guidance from reference genomes by applying a multiplatform genomics approach. We generate a bonobo genome assembly in which more than 98% of genes are completely annotated and 99% of the gaps are closed, including the resolution of about half of the segmental duplications and almost all of the full-length mobile elements. We compare the bonobo genome to those of other great apes[1,3–5] and identify more than 5,569 fixed structural variants that specifically distinguish the bonobo and chimpanzee lineages. We focus on genes that have been lost, changed in structure or expanded in the last few million years of bonobo evolution. We produce a high-resolution map of incomplete lineage sorting and estimate that around 5.1% of the human genome is genetically closer to chimpanzee or bonobo and that more than 36.5% of the genome shows incomplete lineage sorting if we consider a deeper phylogeny including gorilla and orangutan. We also show that 26% of the segments of incomplete lineage sorting between human and chimpanzee or human and bonobo are non-randomly distributed and that genes within these clustered segments show significant excess of amino acid replacement compared to the rest of the genome.

The bonobo or pygmy chimpanzee (*Pan paniscus*) and the common chimpanzee (*Pan troglodytes*) are among the most-recently diverged ape species (around 1.7 million years ago)[1,2]. Both species represent the closest living species to humans and, therefore, offer the potential to pinpoint genetic changes that are also unique to human. The first bonobo sequence, which was generated using short-read whole-genome sequencing[1], resulted in a genome assembly (panpan1.1) with more than 108,000 gaps in which the vast majority of segmental duplications were not incorporated and few structural variants were identified (Supplementary Table 1). As a result of the lower accuracy of early next-generation sequencing technology and the fragmentary nature of the original chimpanzee genome, large fractions of the genomes of great apes could not be compared and gene models were often incomplete[3–8]. In the past few years, long-read genome-sequencing technologies have considerably enhanced our ability to generate contiguous, high-quality genomes in

which most genes and common repeat elements are fully annotated[9]. Here, we apply a multiplatform approach to produce a highly contiguous, accurate bonobo reference genome. Our analysis highlights the extent to and rapidity at which hominid genomes can differ and provides insights into incomplete lineage sorting (ILS) and its relevance to gene evolution and the genetic relationship among living hominids.

## Sequence and assembly

We sequenced DNA from a female bonobo (Mhudiblu, *P. paniscus*) to 74-fold sequence coverage using the long-read PacBio RS II platform (Supplementary Tables 2, 3 and Supplementary Fig. 1). We generated a 3.0-gigabase assembly (contig N50 of 16.58 megabases (Mb)) (Supplementary Table 4) and constructed a chromosomal-level AGP (a golden path) assembly (Mhudiblu_PPA_v0) using Bionano Genomics optical

[1]Department of Genome Sciences, University of Washington School of Medicine, Seattle, WA, USA. [2]Department of Biology, University of Bari, Bari, Italy. [3]UC Santa Cruz Genomics Institute, University of California, Santa Cruz, Santa Cruz, CA, USA. [4]Estonian Biocentre, Institute of Genomics, Tartu, Estonia. [5]Howard Hughes Medical Institute, University of Washington, Seattle, WA, USA. [6]Institute for Systems Biology, Seattle, WA, USA. [7]Pacific Biosciences (PacBio) of California, Menlo Park, CA, USA. [8]Department of Biological Sciences, Louisiana State University, Baton Rouge, LA, USA. [9]National Center for Biotechnology Information, National Library of Medicine, National Institutes of Health, Bethesda, MD, USA. [10]Howard Hughes Medical Institute, University of California, Santa Cruz, Santa Cruz, CA, USA. [11]Bionano Genomics, San Diego, CA, USA. [12]These authors contributed equally: Yafei Mao, Claudia R. Catacchio. ✉e-mail: mario.ventura@uniba.it; eee@gs.washington.edu

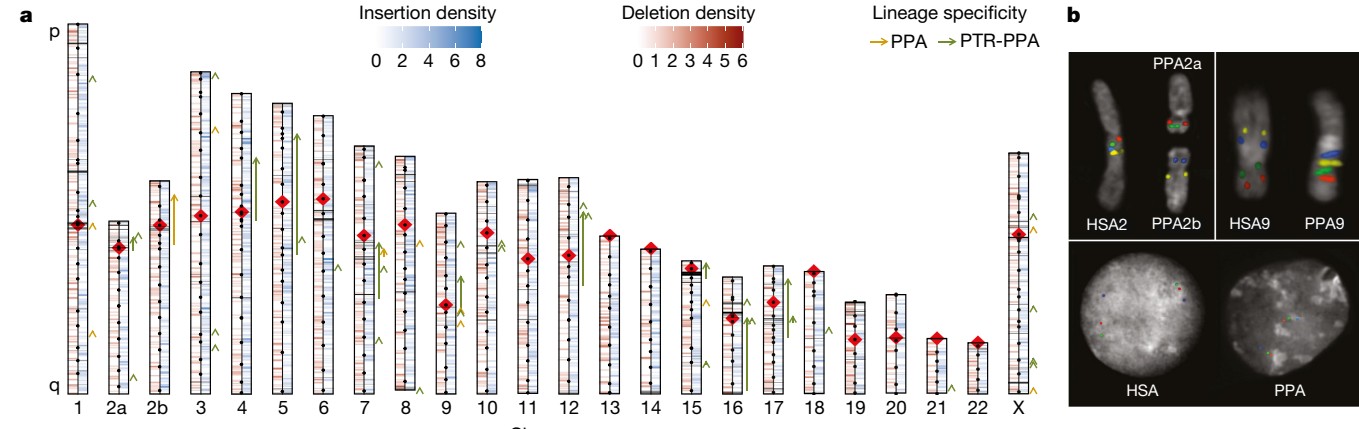

**Fig. 1 | Sequence and assembly of the bonobo genome. a**, Schematic of the Mhudiblu_PPA_v0 assembly depicting the centromere location (red rhombus), FISH probes used to create assembly backbone (black dots), fixed bonobo-specific insertions (blue) and deletions (red) (Supplementary Data), remaining gaps (black horizontal lines) and large-scale inversions (arrows). We distinguish bonobo-specific inversions (dark orange, PPA) from *Pan*-specific inversions (dark green, PTR-PPA). **b**, FISH validation of the bonobo

chromosome 2a and 2b fusion and the 2b pericentric inversion (probes: RP11-519H15 in red, RP11-67L14 in green, RP11-1146A22 in blue, RP11-350P7 in yellow) (top left); the chromosome 9 pericentric inversion (probes: RP11-1006E22 in red, RP11-419G16 in green, RP11-876N18 in blue, RP11-791A8 in yellow) (top right); and the inversion Strand-seq_chr7_inv4a (probes: RP11-118D11 in green, WI2-3210F8 in red, RP11-351B3 in blue) (bottom).

maps and a clone-order framework using fluorescent in situ hybridization (FISH) of bacterial artificial chromosomes (BACs)[10] (Fig. 1). The Mhudiblu_PPA_v0 assembly assigns 74 Mb of new sequence to chromosomes, closing 99.5% of the original 108,095 gaps (Supplementary Table 5). This assembly has been annotated by NCBI and is available in the UCSC Genome Browser (panPan3, Methods, Supplementary Data and Extended Data Fig. 1). We estimate the sequence accuracy of the bonobo assembly to be 99.97–99.99% (Supplementary Table 6 and Supplementary Data). The overall nucleotide divergence between chimpanzee and bonobo based on these new long-read assemblies is $0.421 \pm 0.086\%$ for autosomes and $0.311 \pm 0.060\%$ for the X chromosome (Supplementary Table 7). Using these new assemblies, we genotyped 27 previously sequenced great ape genomes, which resulted in slight adjustments in median effective population sizes for the great apes (Extended Data Fig. 2).

## Gene annotation

We predict 22,366 full-length protein-coding genes and 9,066 noncoding genes using the NCBI Eukaryotic Genome Annotation Pipeline. We also generated 867,690 full-length bonobo cDNAs (Supplementary Table 8) and applied the Comparative Annotation Toolkit[11] to identify 20,478 protein-coding and 36,880 noncoding bonobo gene models; 99.5% of the protein-encoding models show no frameshift errors[12] and 38.4% of the protein-coding isoforms are now more complete. We identify 119 genes that have potential frameshifting insertions or deletions that disrupt the primary isoform relative to the human reference (GRCh38) (Supplementary Table 9). Respectively, 206 and 1,576 protein-coding genes are part of gene families that contracted or expanded in the bonobo genome compared to the human genome (Supplementary Tables 10, 11). We identify 65 putatively previously undescribed exons with support from full-length cDNA (Supplementary Tables 12–14), such as the protein-coding exon in *ANAPC2*, which is found in the bonobo but not in the chimpanzee sequence (Supplementary Fig. 2). Using other great ape genomes[13,14] and a genome-wide analysis from 20 bonobo and chimpanzee samples, we identified genes that showed an excess of amino acid replacement, balancing selection and potential selective sweeps (Tajima's *D* and SweepFinder2)[15]. Most of the genes that showed selective sweeps in bonobo (*DIRC1*, *GULP1* and *ERC2*) (Supplementary Tables 15–18) or chimpanzee (*KIAAO40*, *TM4SF4* and *FOXP2*) (Supplementary Tables 19–22) genomes are novel.

## Mobile element insertions

The number of full-length (retrotransposition-competent), lineage-specific long interspersed nuclear element-1 (L1) in the bonobo genome (413 chimpanzee-specific L1 elements (L1Pt)) is similar to that in the chimpanzee genome (383 L1Pt) and 15–25% greater than the number of elements in the human genome (330 human-specific L1 elements (L1Hs)) (Supplementary Figs. 3–5). An analysis of Alu short interspersed nuclear element (SINE) repeats leads to a refined subfamily classification and we find that the number of bonobo-specific elements ($n = 1,492$) is nearly identical to that in the chimpanzee genome ($n = 1,431$). *Pan* lineages, therefore, show among the lowest rates of Alu insertions compared to the human genome (in which the rate has doubled) and the rhesus macaque genome (which shows a tenfold increased rate) (Extended Data Fig. 3). Although the bonobo genome shows a reduced genetic diversity of single-nucleotide variants[7,16] compared to the chimpanzee genome, we find that bonobo SINE–variable number tandem repeat (VNTR)–Alu (SVA) elements are more copy number polymorphic (45%) (Extended Data Fig. 3) compared to the chimpanzee genome (35%; $P < 6.5 \times 10^{-4}$). By contrast, the chimpanzee-specific endogenous retrovirus (PtERV1) shows an indistinguishable low rate of polymorphism for PtERV1 in both species (7% for bonobo and 9% for chimpanzee), which suggests relatively little activity since the divergence of *Pan* (Supplementary Data).

## Segmental duplications

We identified 87.4 Mb of segmental duplications (≥1 kilobase (kb) and ≥90% identity) (Extended Data Fig. 3, Supplementary Figs. 6, 7 and Supplementary Table 23), most of which was previously unassembled. Segmental duplications are interspersed with an excess of large (≥10 kb) intrachromosomal duplications, which is consistent with the burst of segmental duplications that occurred at the root of the hominid lineage[17]. Despite the approximately sixfold improvement, the largest and most identical duplications were still not initially resolved (around 84 Mb). Using the Segmental Duplication Assembler algorithm[18,19], we successfully resolved an additional 56 Mb (Supplementary Table 24) and used these data to identify recent gene family expansions (Extended Data Fig. 4 and Supplementary Tables 25–31). We show, for example, that the eukaryotic translation initiation factor 4 subunit A3 (*EIF4A3*) gene family has expanded in both chimpanzee and

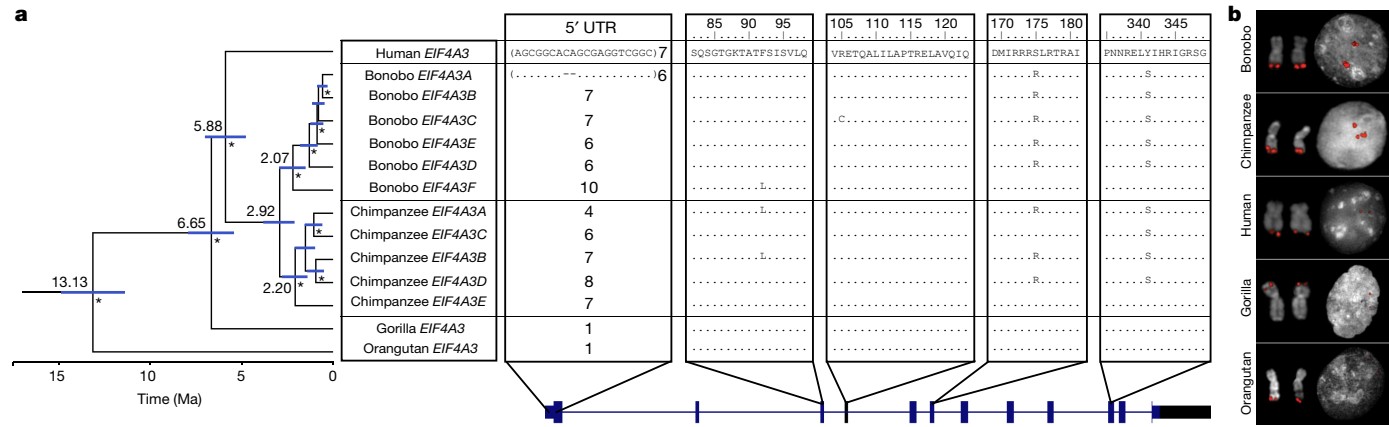

**Fig. 2 | EIF4A3 gene family expansion and sequence resolution. a**, Multiple sequence alignment shows *EIF4A3* amino acid differences between the human, Mhudiblu_PPA and chimpanzee assembled paralogues, and sequences of other great apes. A polymorphic 18-bp motif VNTR is located at the 5′ UTR of nonhuman primate *EIF4A3* and accounts for most of the differences between various isoforms. A phylogenetic tree is built from neutral sequences of *EIF4A3* paralogues using Bayesian phylogenetic inference. This analysis is conducted using BEAST2 software. Numbers on each major node denote estimated divergence time. Ma, million years ago. The blue error bar on each node indicates the 95% confidence interval of the age estimation. Bayesian posterior probabilities are reported using asterisks for nodes with posterior probability >99%. **b**, FISH on metaphase chromosomes and interphase nuclei with human probe WI2-3271P14 confirms an *EIF4A3* subtelomeric expansion of chromosome 17 in bonobo and chimpanzee relative to human, gorilla and orangutan.

bonobo genomes. There is evidence that five out of the six paralogues are expressed and encode a full-length open-reading frame (Fig. 2 and Extended Data Fig. 5). We estimate that the initial *EIF4A3* gene duplication occurred in the ancestral lineage approximately 2.9 million years ago. It then subsequently expanded and experienced gene conversion events independently in the chimpanzee and bonobo lineages, creating five and six copies of the *EIF4A3* gene family, respectively. Notably, some of the gene conversion signals correspond to a set of specific amino acid changes in the basic ancestral structure that are now common to only chimpanzee and bonobo (Fig. 2 and Extended Data Fig. 5).

## Structural variation and gene disruption

As part of the assembly curation, we validated nine larger inversions that distinguish human and bonobo karyotypes, created a FISH-based chromosomal backbone (Fig. 1) and used single-cell DNA template strand sequencing (Strand-seq) to assign orphan contigs to chromosomes (36 Mb) (Mhudiblu_PPA_v1) (Supplementary Tables 32–38). We identify 17 fixed inversions that differentiate bonobo from chimpanzee, of which 11 are bonobo-specific (Supplementary Table 39) and 22 regions that probably represent bonobo inversion polymorphisms (Supplementary Table 40). Moreover, we assign 38 fixed inversions that occurred in the common *Pan* ancestor (Supplementary Table 39). We annotated and validated the breakpoint intervals of each tested inversion (Supplementary Table 41) and found segmental duplications or long interspersed nuclear elements at the breakpoints of inversions in 82% and 86% of cases, respectively (Supplementary Table 40). We also compared the bonobo genome to the human, chimpanzee and gorilla genomes to identify deletions and insertions (>50 base pairs (bp)). We classify 15,786 insertions and 7,082 deletions as bonobo-specific and genotyped these in a population of great ape samples[7,16,20] to identify 3,604 fixed insertions and 1,965 fixed deletions, of which only a small fraction (2.66% or 148 out of 5,569) intersect with genic functional elements (Supplementary Tables 42–45).

Bonobo-specific events that delete ENCODE regulatory elements[21] (*n* = 381), for example, are enriched in membrane-associated genes with extracellular domains whereas chimpanzee-specific events (*n* = 187) are associated with cadherin-related genes (Supplementary Table 46). Deletions (*n* = 1,040) shared between the chimpanzee and bonobo genomes show an enrichment of the loss of putative regulatory elements associated with post-synaptic genes (3.32 enrichment; $P = 1.2 \times 10^{-7}$) and

pleckstrin homology-like domains (6.15 enrichment; $P = 1.20 \times 10^{-9}$). We validate 110 events that disrupt protein-coding genes by generating high-fidelity genomic sequencing for each of the great ape reference genomes and restricting to those events that could be genotyped in a population of genomes (Supplementary Data). As expected, many fixed gene-loss events occurred in genes that are tolerant to mutation, redundant duplicated genes or genes in which the event simply altered the structure of the protein. For example, we validate a 25.7-kb gene loss of one of the keratin-associated genes (*KRTAP19-6*) associated with hair production in the ancestral lineage of chimpanzee and bonobo (Supplementary Fig. 8). In the bonobo lineage, we identify five fixed structural variants that affect protein-coding genes (Supplementary Table 47), but only two of which completely ablate the gene. For example, *LYPD8*, which encodes a secreted protein that prevents invasion of the colonic epithelium by Gram-negative bacteria, has been completely deleted by a 24.3-kb bonobo-specific deletion. Similarly, *SAMD9* (SAMD family member 9) is a fixed gene loss in bonobo as a result of a 41.46-kb bonobo-specific deletion. The other three bonobo-specific fixed structural variant events in protein-coding regions all maintain the open-reading frame, including a 49-amino acid deletion of *ADAR1*, which encodes a protein that is critical for RNA editing and is implicated in human disease[22–24] (Extended Data Fig. 6).

## A comparison of ILS in hominids

The higher quality and more contiguous nature of the bonobo genome provide an opportunity to generate a higher-resolution ILS map. In comparison to the original bonobo assembly in which only around 800 Mb (27%) could be analysed, it is now possible to align approximately 76% of the genome in a four-way ape genome alignment (2,357 Mb within 10-kb windows) (Supplementary Table 48) owing to long-read genome assemblies[14]. We performed a genome-wide phylogenetic window-based analysis to systematically identify regions that are inconsistent with the species tree and classified these as human–bonobo and human–chimpanzee ILS topologies (Fig. 3). We predict that 5.07% of the human genome is genetically closer to chimpanzee or bonobo (Table 1); 2.52% of the human genome is more closely related to the bonobo genome (human–bonobo ILS segments) than the chimpanzee genome whereas 2.55% of the human genome is more closely related to the chimpanzee genome (human–chimpanzee ILS) than the bonobo genome (Fig. 3a). This proportion of ILS nearly doubles previous estimates (3.3%)[1]

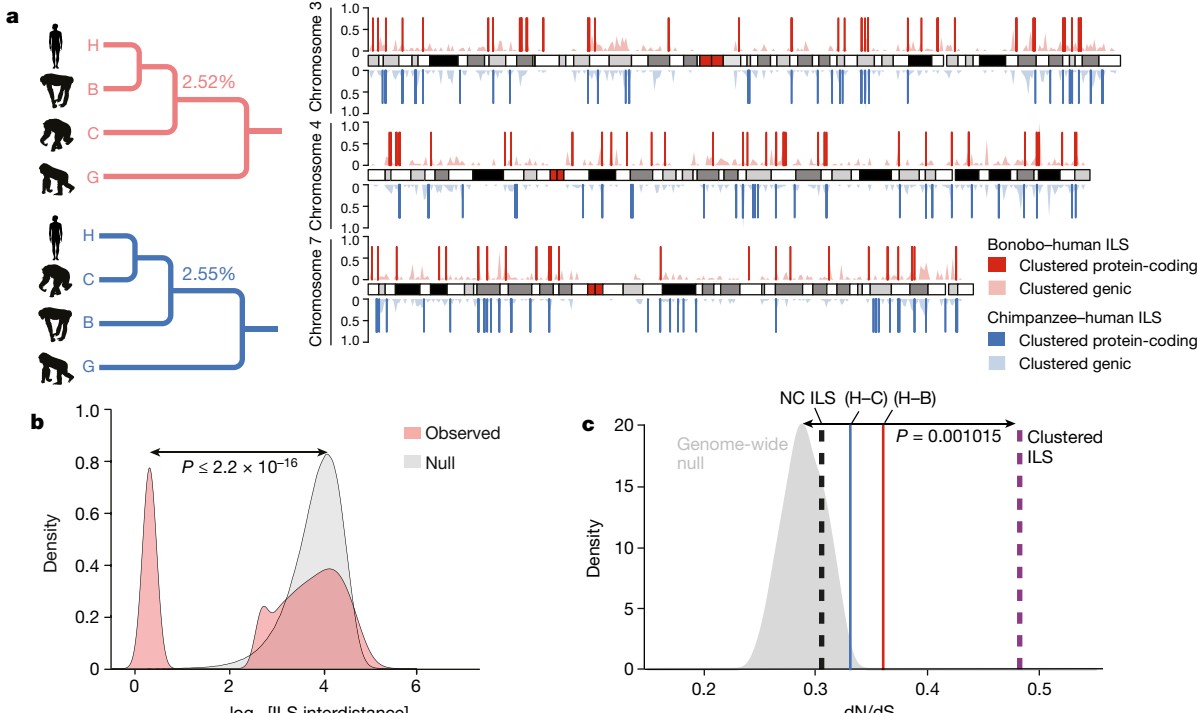

**Fig. 3 | Hominid ILS. a**, A whole-genome ILS cladogram analysis (left) for bonobo–human (red) and chimpanzee–human (blue) and a schematic map (right) of clustered ILS segments (500-bp resolution) specifically for chromosomes 3, 4 and 7. The lighter density plot represents the clustered ILS events mapping to intragenic regions, whereas the vertical lines represent the subset that overlap with protein-coding exons. **b**, Distribution of distances between ILS segments (inter-ILS) (500-bp resolution) compared with a simulated (null) expectation (from 400,000 simulations) reveals a bimodal pattern with a subset (26%) that is clustered and significantly non-randomly distributed. A two-sample Wilcoxon rank-sum test was used to calculate the *P* value in R. **c**, ILS exons show a significant excess of amino acid replacement

(dN/dS) for both human–bonobo (H–B; red line; *P* = 0.004778) and human–chimpanzee (H–C; blue line; *P* = 0.03924) ILS. In particular, exons mapping to the ILS clustered segments (**b**) show the most significant excess of amino acid replacements dN/dS (dotted purple line; *P* = 0.001015) compared to the genome-wide null distribution (grey density plot). This shift is not observed for the non-clustered ILS segments (NC ILS; dotted black line; *P* = 0.3161). Significance was analysed using the one-sample Student's *t*-test in R. The silhouette of the chimpanzee in **a** is created by T. Michael Keesey and Tony Hisgett (http://phylopic.org/; image is under a Creative Commons Attribution 3.0 Unported licence); silhouettes of bonobo and gorilla are from http://phylopic.org/ under a Public Domain Dedication 1.0 licence.

(Supplementary Table 1). Consistent with previous observations[1], the largest ILS segments are biased (around 1.8-fold) to intergenic regions, depleted for genes (>35%) and are particularly enriched in L1 content. Notably, the distribution of ILS segments is highly non-random based on simulation experiments. We specifically measured the distance between ILS segments (see below) and identified a subset (around 26%) of sites that are significantly more clustered than expected by chance (Fig. 3b).

We focused specifically on protein-coding exons based on the human RefSeq annotation[25] and identified 1,446 exons that mapped to ILS topologies (713 exons to a human–bonobo topology and 733 exons to a human–chimpanzee topology) (Supplementary Table 49). As a whole, genes corresponding to these ILS exons are significantly enriched in both glycoprotein function ($P = 1.30 \times 10^{-14}$ for human–bonobo and $P = 5.60 \times 10^{-11}$ for human–chimpanzee) and calcium-binding epidermal growth factor (EGF) domain function ($P = 4.40 \times 10^{-12}$ for human–bonobo and $P = 9.40 \times 10^{-7}$ for human–chimpanzee) (Supplementary Table 50). We considered multiple occurrences in the same gene and identified 84 genes with at least two exons under ILS (Supplementary Table 51) with some enrichment in photoreceptor activity ($P = 1.6 \times 10^{-4}$) (Supplementary Table 51 and Supplementary Fig. 9) as well as EGF-like ($P = 1.9 \times 10^{-6}$) and transmembrane ($P = 2.4 \times 10^{-3}$) functions. Overall, we observe a significant excess of amino acid replacement (dN/dS) for all 1,446 ILS exons compared to non-ILS exons ($P = 0.0048$ for human–bonobo, $P = 0.039$ for human–chimpanzee) (Fig. 3c), which is consistent with either the action of relaxed selection or positive selection. Exons mapping to the clustered ILS segments show greater dN/dS with respect

to exons in the non-clustered ILS segments, which suggests that these clustered ILS segments are contributing disproportionately to accelerated amino acid evolution in the hominid genome.

We extended the ILS analysis (Supplementary Data) across 15 million years of hominid evolution through the inclusion of genome data from orangutan and gorilla. As expected, ILS estimates for the human genome increase to more than 36.5% (Extended Data Fig. 7 and Supplementary Table 52) similar to (albeit still greater than) previous estimates[3,14]. We measured the inter-ILS distance and observed a consistent non-random pattern of clustered ILS for these deeper topologies with more ancient ILS showing an even greater proportion of clustered sites (Extended Data Fig. 7). Once again, we observe a significantly increased mean dN/dS in clustered human–chimpanzee and human–bonobo topologies ($P < 2.2 \times 10^{-16}$, mean = 0.366) as well as clustered orangutan–human and orangutan–gorilla–human topologies ($P < 2.2 \times 10^{-16}$, mean = 0.316) compared to the null distribution (Supplementary Fig. 10). A Gene Ontology analysis[26] of the genes that intersect these combined data confirm not only the most significant signals for immunity (for example, glycoprotein ($P = 1.3 \times 10^{-25}$) and immunoglobulin-like fold/FN3 ($P = 2.4 \times 10^{-20}$)), but also genes related to EGF signalling ($P = 1.6 \times 10^{-13}$), solute transporter function (for example, transmembrane region ($P = 1.3 \times 10^{-25}$)) and, specifically, calcium transport ($P = 3.7 \times 10^{-8}$) (Supplementary Table 53). Although ILS regions, in general, show diversity patterns of single-nucleotide polymorphisms that are consistent with balancing selection, it is noteworthy that both clustered and non-clustered ILS exons show a significant excess of polymorphic gene-disruptive events that are consistent with the action

## Table 1 | Hominid genome-wide ILS estimates

| Window size | Number of ILS segments | | Percentage of ILS | | Total ILS[a] | Genomic properties | | | | |
|---|---|---|---|---|---|---|---|---|---|---|
| | (G, ((B, H), C)) | (G, ((H, C), B)) | (G, ((B, H), C)) | (G, ((H, C), B)) | | GC[a] | Intergenic/intragenic | Alu[a] | L1[a] | Exon[a] |
| 20 kb | 218 | 218 | 0.19 | 0.19 | 0.38 | 37.7 | 1.79 | 6.37 | 31.44 | 0.49 |
| 10 kb | 1,143 | 1,138 | 0.49 | 0.48 | 0.97 | 38.39 | 1.73 | 7.35 | 27.08 | 0.47 |
| 5 kb | 4,314 | 4,373 | 0.91 | 0.92 | 1.83 | 38.95 | 1.64 | 7.85 | 24.67 | 0.58 |
| 2 kb | 18,218 | 18,334 | 1.52 | 1.53 | 3.05 | 39.58 | 1.49 | 8.71 | 21.51 | 0.72 |
| 1 kb | 46,584 | 46,938 | 2.06 | 2.07 | 4.13 | 40.06 | 1.37 | 9.8 | 19.85 | 0.8 |
| 500 bp | 102,197 | 103,338 | 2.52 | 2.55 | 5.07 | 40.54 | 1.33 | 11.24 | 18.66 | 0.75 |
| Genome average | | | | | | 40.89 | 1.21 | 10.17 | 17.42 | 1.17 |

B, bonobo; C, chimpanzee; G, gorilla; H, human. (G, ((B, H), C)) and (G, ((H, C), B)) represent two different ILS topologies. Intergenic/intragenic indicates the intergenic to intragenic ratio.
[a]Content is shown as a percentage; the GC, Alu, L1 and exon contents are based on the GRCh38 genome.

of relaxed as well as balancing selection (Supplementary Fig. 11). An examination of these gene-rich clustered ILS regions reveals a complex pattern of diverse ILS topologies that suggests deep coalescence operating across specific regions of the human genome as has previously been reported for the major histocompatibility complex[1,3] (Extended Data Fig. 8).

## Discussion

High-quality hominid genomes are a critical resource for understanding the genetic differences that make us human as well as the diversification of the *Pan* lineage over the past two million years of evolution. The bonobo represents the last of the great ape genomes to be sequenced using long-read sequencing technology. Its sequence will facilitate more systematic genetic comparisons between human, chimpanzee, gorilla and orangutan without the limitations of technological differences in sequencing and assembly of the original reference[1,3–5,14]. As a result, we now predict that a greater fraction (around 5.1%) of the human genome is genetically closer to chimpanzee or bonobo compared to previous studies (3.3%)[1]. We estimate that more than 36.5% of the hominid genome shows ILS if we consider a deeper phylogeny that includes gorilla and orangutan. Notably, 26% of the ILS regions are clustered and exons that underlie these clustered ILS signals show elevated rates of amino acid replacement. These findings support a previous study in gorilla that showed a subtler correlation in which genes with higher dN/dS values are enriched in ILS segments[3]. In that study, however, the authors explained the observation as a result of stronger purifying selection in non-ILS sites or background selection that reduced the effective population size and, as a result, led a depletion of ILS. Our genome-wide exon analyses specifically show that only a subset of clustered ILS exons are driving this effect and that these genes are enriched in glycoprotein and EGF-like calcium signalling functions owing to the action of either relaxed selection or positive selection of genes in these pathways (Supplementary Data).

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

## Methods

We sequenced and assembled the genome of a single female bonobo (Mhudiblu, also known as Mhudibluy, who was obtained from the San Diego Zoo, ISIS 601152, born 15 April 2001 and who was later transferred to the Wuppertal Zoo in Germany where she was referred to as Muhdeblu) using long-read PacBio RS II sequencing chemistry and the Falcon genome assembler. The assembly was error-corrected using Quiver[27], Pilon[28] and an in-house FreeBayes-based[29] insertion or deletion correction pipeline optimized to improve continuous long-read assemblies[14]. We also generated Illumina whole-genome sequencing (WGS) data using the Illumina TruSeq PCR-Free library preparation kit. Genome assembly contigs were ordered and oriented into scaffolds using Bionano optical maps (Supplementary Table 54 and Supplementary Data) (HybridScaffolds suite, Bionano Genomics Saphyr platform) and four-colour FISH of 324 BAC clones. Cell lines from chimpanzee, bonobo, gorilla and orangutan were obtained from Coriell (S006007) or from a collection developed by M. Rocchi; no approval from ethics committees were required for use of these established lines. We assigned each contig and scaffold into unique groups corresponding to individual chromosomal homologues using SaaRclust[30,31] while applying Strand-seq to detect inversions, assign orphan contig and orient contigs[32,33]. To estimate genome-wide sequence accuracy, we applied Merqury[34] using Illumina WGS data. We also generated a bonobo large-insert BAC library (VMRC74) and selected at random 17 clones for complete PacBio insert sequencing[35]. The Comparative Annotation Toolkit (CAT)[11] was used for genome annotation using human GENCODE v.33 and RNA-sequencing data. We also generated more than 860,000 full-length non-chimeric transcripts from full-length isoform sequencing (Iso-Seq) data generated from induced pluripotent stem cell and derived neuronal progenitor cell lines[36] from bonobo sample AG05253 and we searched for gene structures split over multiple contigs (Supplementary Table 55). Repeat content of the assembled genome was analysed using RepeatMasker (RepeatMasker-Open-4.1.0) and the Dfam3 repeat library. We assigned lineage-specific Alu and full-length long interspersed nuclear element, SVA_D and PtERV elements to subfamilies by applying COSEG (http://www.repeatmasker.org/COSEGDownload.html) to determine the lineage-specific subfamily composition. For cross-species analysis of mobile element insertions (MEIs), we performed liftOver on the basis of the chains built from the Cactus whole-genome alignments generated during CAT annotation. For cross-assembly analyses of bonobo MEI insertions and a specific subset of other analyses (Supplementary Data), we used Bowtie 2 to map MEI flanking sequences between genomes. We estimated the duplication content in the bonobo assembly, applying the whole-genome analysis comparison method[37] and targeted collapsed duplications for assembly using Segmental Duplication Assembler[19]. Insertions and deletions were detected in bonobo, chimpanzee and gorilla using PBSV, Sniffles[38] and Smartie-sv[14] and genotyped using Paragraph[39] against a panel of 27 Illumina WGS genomes. We searched for evidence of ILS among the chimpanzee, gorilla and human lineages applying Prank (v.140110) to construct multiple sequence alignments and using ete3 module to identify segments and exons under ILS (Supplementary Table 56). For consistency, NCBI reference genome nomenclature has been used throughout the manuscript and corresponds to the following UCSC IDs (NCBI/UCSC): panpan1.1/panPan2, Mhudiblu_PPA_v0/panPan3, Clint_PTRv2/panTro6, Kamilah_GGO_v0/gorGor6, Susie_PABv2/ponAbe3 and GRCh38/hg38 (details of the methods used are provided in the Supplementary Data).

### Reporting summary

Further information on research design is available in the Nature Research Reporting Summary linked to this paper.

## Data availability

The Mhudiblu_PPA_v0 (GCA_013052645.1), Mhudiblu_PPA_v1 (GCA_013052645.2) and Mhudiblu_PPA_v2 (GCA_013052645.3) assemblies are deposited in the NCBI under BioProject accession number PRJNA526933. The raw PacBio continuous long-read, Strand-seq, Illumina and Iso-Seq data of bonobo are deposited in the NCBI under SRA accession number SRP188441. The Bionano map of bonobo Mhudiblu is deposited in the NCBI under BioProject accession number PRJNA526933. The raw PacBio HiFi data of bonobo Mhudiblu and gorilla Kamilah are deposited in the NCBI under SRA accession number SRP301932 under BioProject accession number PRJNA691628. The BACs used in this study are listed in Supplementary Table 57 in the NCBI with BioProject accession PRJNA634395.

## Code availability

Custom scripts used in this study are available at GitHub (https://github.com/EichlerLab and https://github.com/MaoYafei).

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

**Acknowledgements** We thank L. Carbone and O. Ryder for providing the Mhudiblu bonobo cell line; R. Gage and C. Marchetto for the preparation of the RNA sequence and access to bonobo induced pluripotent stem cell lines; A. Rhie for assistance with Merqury analysis; and T. Brown for manuscript proofreading and editing. The silhouettes of the bonobo and gorilla in Fig. 3 and Extended Data Fig. 2 (and, additionally, the human silhouette in Extended Data Fig. 2) are downloaded from phylopic.org under a Public Domain Dedication 1.0 licence. The silhouette of the chimpanzee in Fig. 3 and Extended Data Fig. 2 is downloaded from phylopic.org under a Creative Commons Attribution 3.0 Unported licence (https://creativecommons.org/licenses/by/3.0/, credit to T. M. Keesey and T. Hisgett). This work was supported, in part, by National Institutes of Health (NIH) grants HG002385 and 1U24HG009081 to E.E.E.; Futuro in Ricerca 2010-RBFR103CE3 to M.V.; R01 GM59290 to M.A.B.; 2U41HG007234 to M.D. and B.P.; U01HG010961, U41HG010972, R01HG010485, U01HL137183 and 5U54HG007990 to B.P.; Arian Smit's NHGRI grant RO1 HG002939 to J.M.S.; 5T32HG008345-04 to B.P.; R01HG010329-01 to S.R.S.; and European Regional Development Fund 2014-2020.4.01.16-0030 to F.M. E.E.E. is an investigator of the Howard Hughes Medical Institute. The work of J.H. and F.T.-N. was supported by the Intramural Research Program of the National Library of Medicine, National Institutes of Health. P.H. is supported by an NIH Pathway to Independence Award (NHGRI, K99HG011041).

**Author contributions** K.M.M. and A.P.L. generated long-read sequencing data; C.B. created the BAC library (VMRC74); K.H., M.S., P.A.A., D.S.G., L.W.H., S.C.M., M.D., C.R.C., L.M., I.P., M.V., E.E.E. and D.P. completed the de novo assembly, its curation and quality assessment; P.C.D., R.L., Y.M., W.T.H., T.-H.H., D.S.G., M.V. and E.E.E. performed segmental duplication and gene family analyses; D.P. performed Strand-seq single-cell data analysis; C.R.C., I.P., L.M., F.A. and M.V. performed FISH analyses; A.W.C.P., J.L. and A.R.H. led the Bionano Genomics analyses; I.T.F., M.D., Y.M., M.H. and B.P. performed gene annotation and gene analyses; J.H. and F.T.-N. performed the RefSeq annotation; J.G.U. performed Iso-Seq; F.M., P.H. and Y.M. performed population genetic analyses; J.D.F., J.M.S., S.R.S., Y.M., J.A.W. and M.A.B. performed MEI analyses; Y.M. performed structural variant analyses; Y.M., A.S. and P.H. performed ILS analyses; M.V. and E.E.E. supervised the project and finalized the manuscript. All authors read and approved the manuscript.

**Competing interests** J.G.U. is an employee of Pacific Biosciences. A.W.C.P., J.L. and A.R.H. are employees of Bionano Genomics. The other authors declare no competing interests.

**Additional information**

**Correspondence and requests for materials** should be addressed to M.V. or E.E.E.

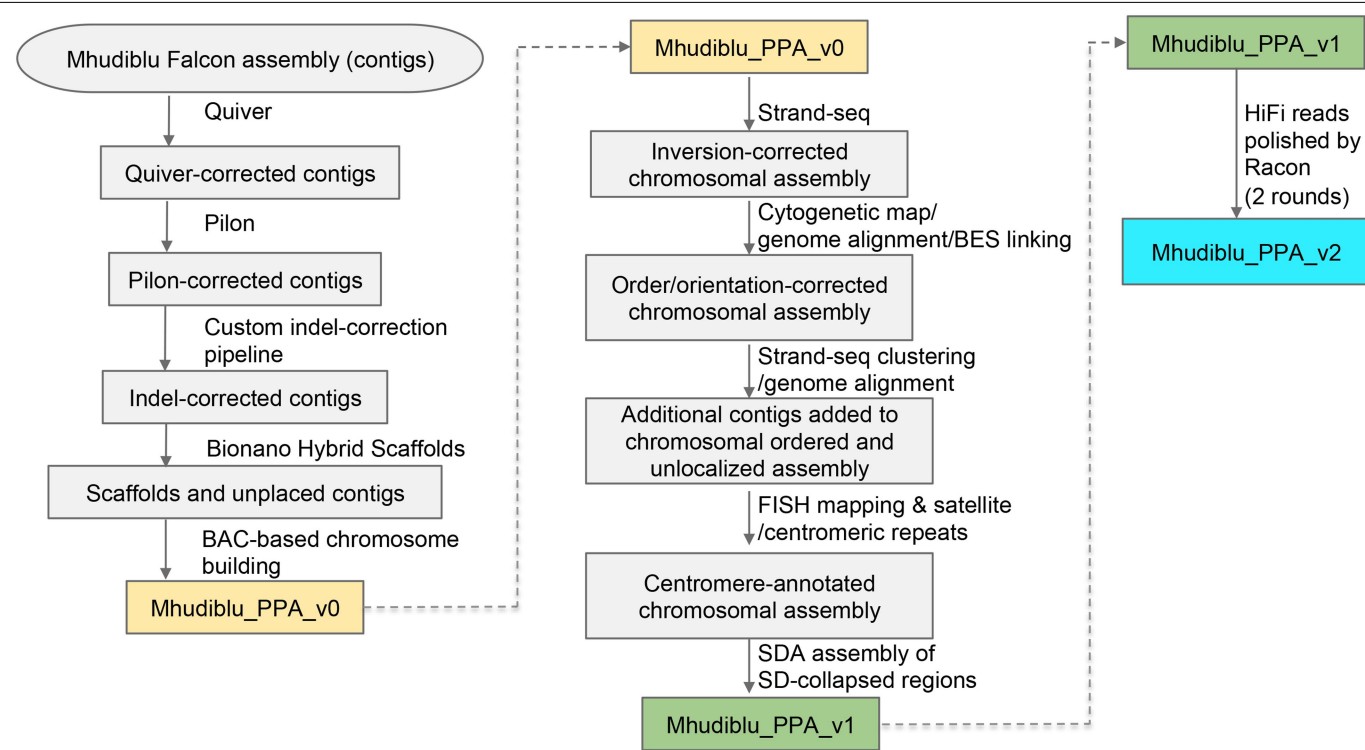

**Extended Data Fig. 1 | Workflow schematic of bonobo assembly pipeline.** Processing steps to create the reference sequences Mhudiblu_PPA_v0, Mhudiblu_PPA_v1 and Mhudiblu_PPA_v2.

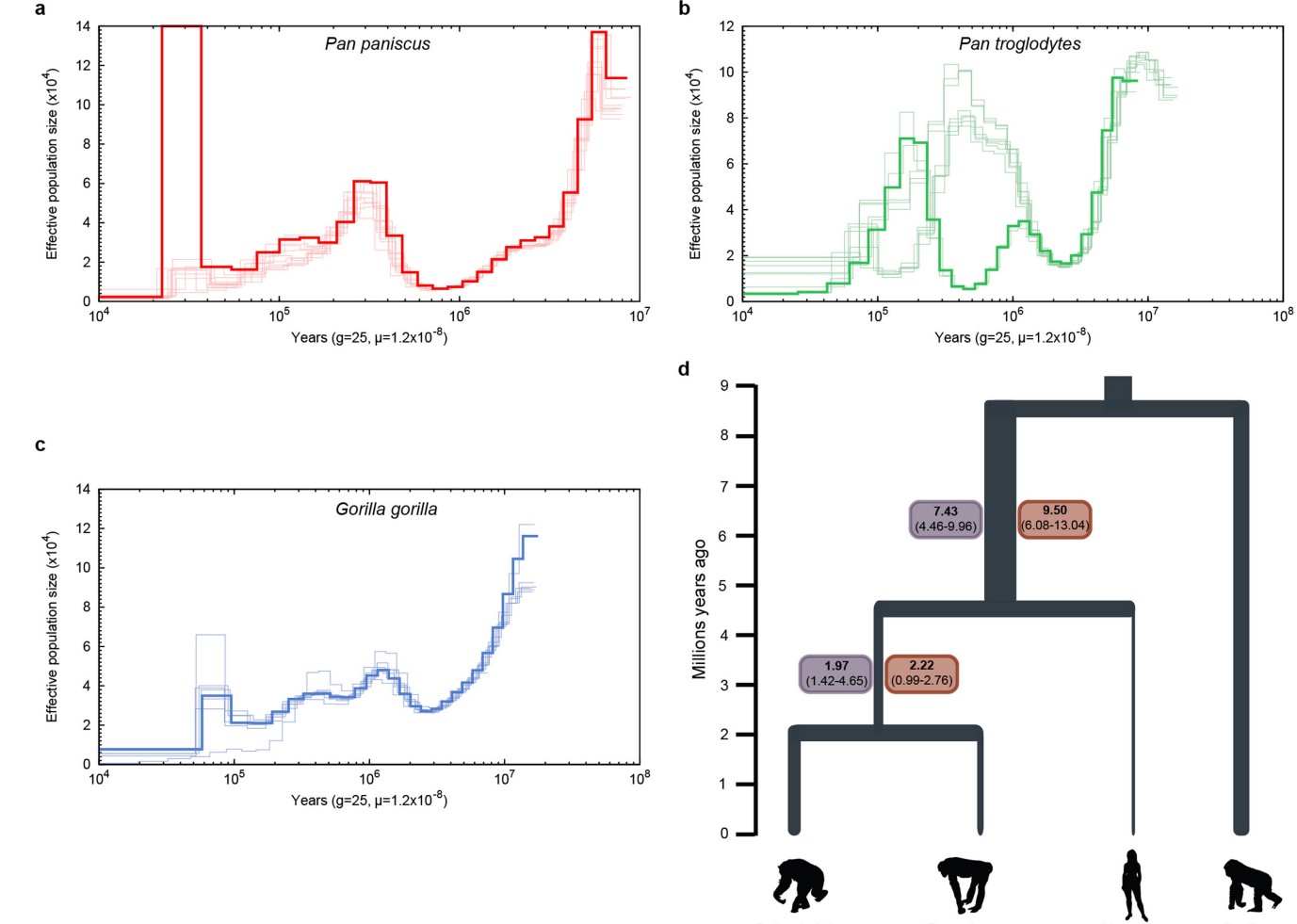

**Extended Data Fig. 2 | Pairwise sequentially Markovian coalescent analysis and estimates of the effective population size predating the divergence in *Homo* and *Pan*. a–c**, Pairwise sequentially Markovian coalescent (PSMC) plots based on an analysis of Illumina WGS genomes of 10 bonobos (**a**; red), 10 chimpanzees (**b**; green) and 7 gorillas (**c**; blue). The $y$ axis represents the effective population size ($N_e$) ($\times 10^4$) inferred by the PSMC and the $x$ axis represents the time in years. $N_e$ values and time are scaled with generation time $g = 25$ years and a mutation rate of $\mu = 1.2 \times 10^{-8}$ per bp per generation[16]. **d**, Values in boxes refer to median and 95% confidence interval $N_e$ ($\times 10^4$) values inferred through PSMC analysis considering bonobo (red boxes) and chimpanzee (purple). We extracted size estimates from time intervals between 4 and 7 million years ago for the *Homo*, *Pan* $N_e$ and been 1 and 2.5 million years ago for the *P. paniscus*, *P. troglodytes* $N_e$, considering $\mu = 0.5 \times 10^{-9}$ mutations (bp × year) and a generation time of 25 years. Values using $\mu = 1 \times 10^{-9}$ mutations (bp × year) are reported in Supplementary Data.

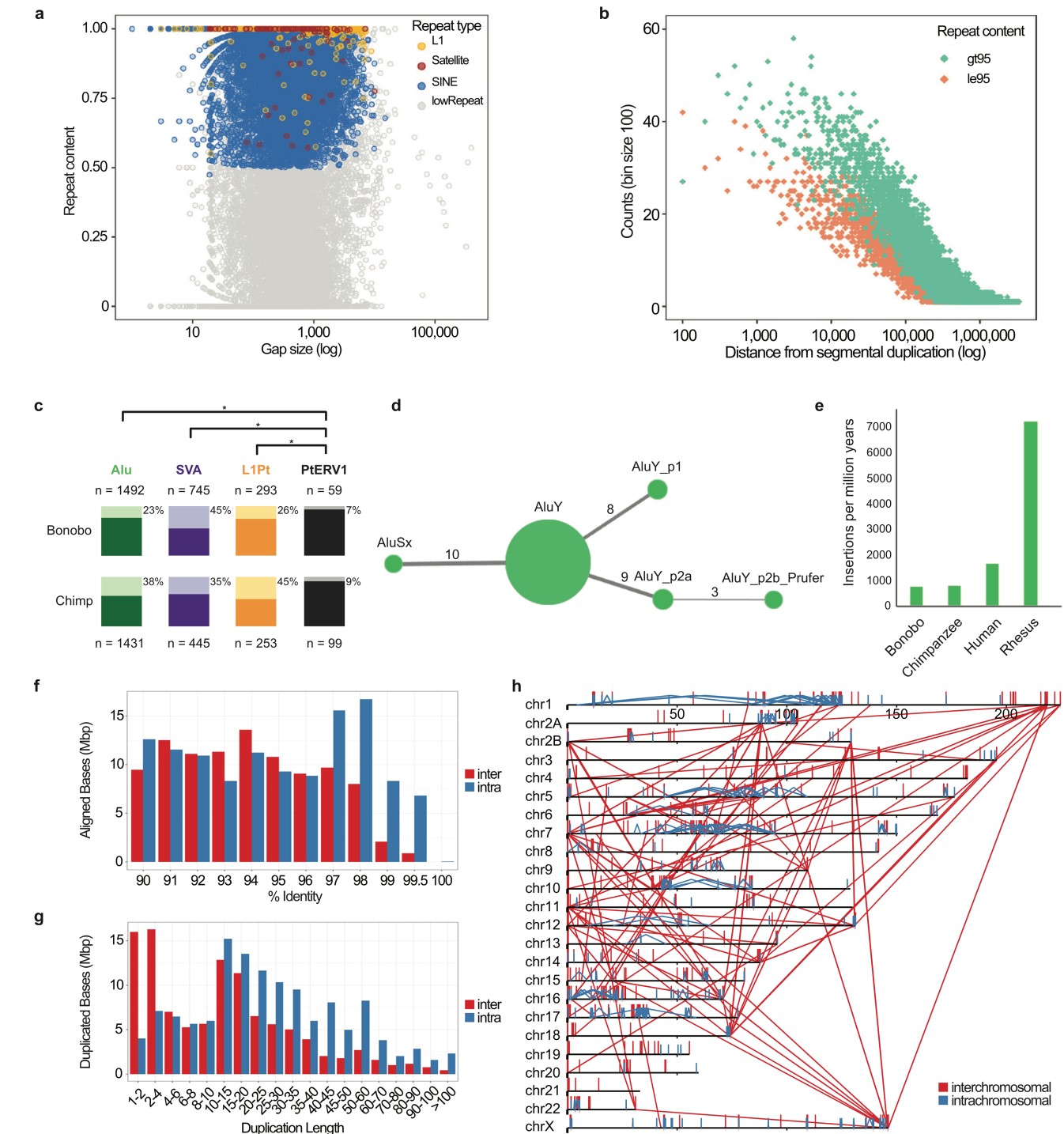

**Extended Data Fig. 3 | Sequence and assembly of the bonobo genome and bonobo genome repeat structure. a**, The size (x axis is shown on a log scale) and repeat content of gaps filled in the new bonobo assembly compared with the panpan1.1 assembly[1]. Gaps composed of more than 50% repeat content for any particular class of repeat are coloured. **b**, Distance from filled gaps to the nearest segmental duplication (x axis) versus the counts of highly repetitive (>95%, green) and less repetitive (≤95%, orange) filled gaps in 100 base-pair bins (y axis). An additional 2,600 and 1,755 filled gaps map directly within segmental duplication sites with ≤95% and >95% repeat content, respectively. **c**, Polymorphism rates for lineage-specific MEIs. Alu, SVA, L1Pt and PtERV1 insertions that do not 'lift over' between chimpanzee and bonobo reference genomes were identified and genotyped for deletions using data from 10 bonobos and 10 chimpanzees. Light-coloured bars and percentages represent the fraction of instances of the MEI type that display support for polymorphism; dark-coloured bars represent the fraction of fixed insertions in

these populations. PTERV1 displays a significantly less polymorphic fraction than Alu ($P = 2.6 \times 10^{-74}$, chimpanzee; $P = 6.9 \times 10^{-35}$, bonobo; $\chi^2$ test, Bonferroni correction), SVA ($P = 3.8 \times 10^{-19}$; $P = 1.9 \times 10^{-62}$) or L1Pt ($P = 2.2 \times 10^{-18}$; $P = 1.3 \times 10^{-8}$), reflecting its lack of activity since the divergence of *Pan*. SVA is the only MEI type with a greater polymorphism rate in bonobo. **d**, A COSEG network of bonobo-specific Alu subfamilies indicating the relative number of elements (size of the node) and number of mutations (line thickness) that distinguish subfamilies. **e**, A comparison of the retrotransposition rate per million years based on lineage-specific Alu insertions from a select panel of primate genomes. **f**–**h**, The percentage identity distribution (**f**) and length distribution (**g**) of segmental duplications (≥90% identify, ≥1 kb and no unplaced contigs) are shown as well as the pattern of the largest and most identical (≥10 kb and ≥98%) intrachromosomal (blue) and interchromosomal (red) segmental duplications (**h**) in the bonobo genome.

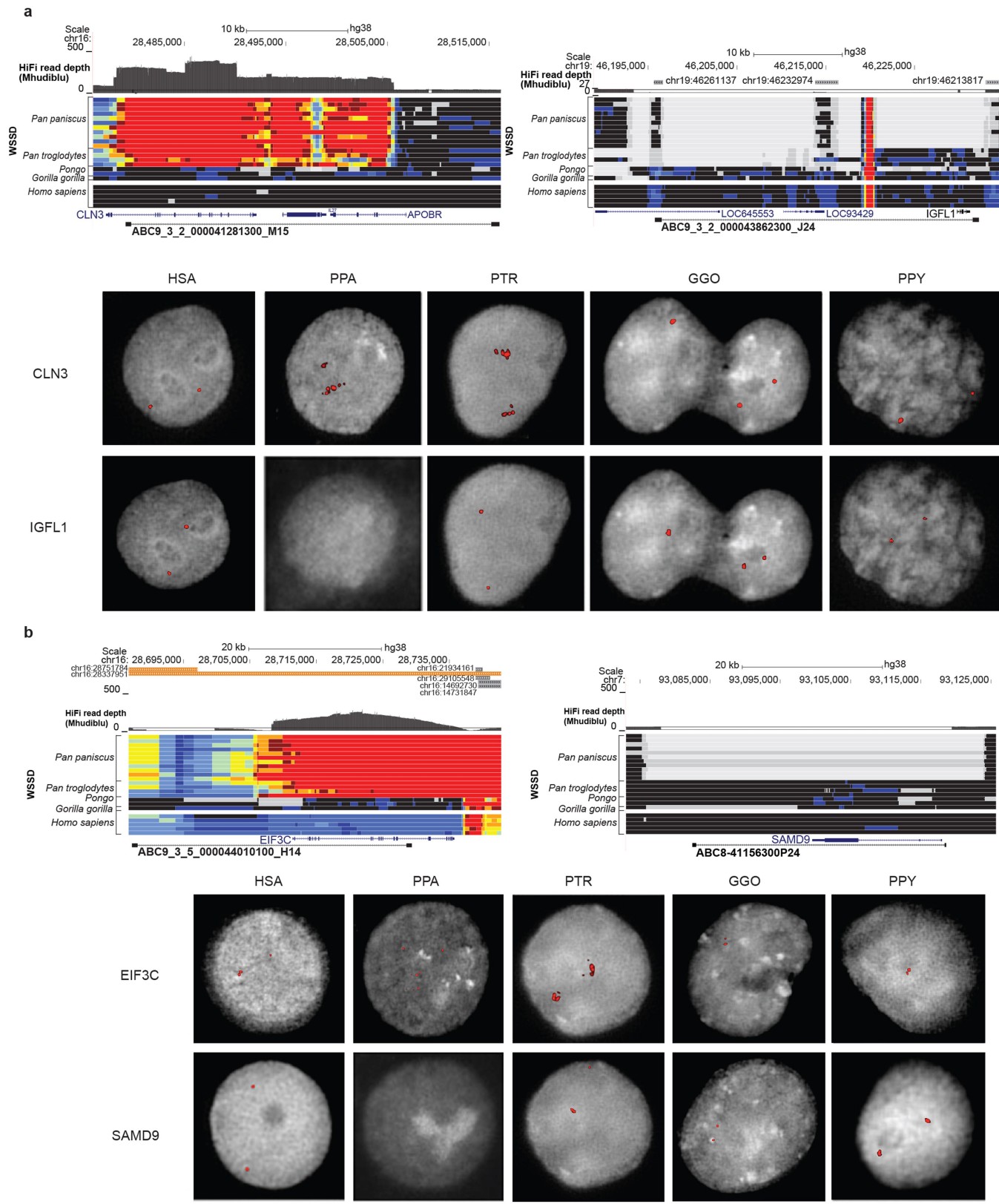

**Extended Data Fig. 4** | See next page for caption.

**Extended Data Fig. 4 | *Pan*-specific duplications and bonobo-specific deletions. a,** *Pan*-specific duplication of the *CLN3* locus and bonobo-specific deletion of *IGFL1*. HiFi read depth and whole-genome shotgun detection of bonobo, chimpanzee, orangutan, gorilla and human individuals relative to GRCh38 detect these events (top), which are validated by interphase FISH of each species using fosmid clones spanning the region (bottom). **b,** *Pan*-specific duplication of the *EIF3C* locus and bonobo-specific deletion of *SAMD9*. HiFi read depth and whole-genome shotgun detection of bonobo, chimpanzee, orangutan, gorilla and human individuals relative to GRCh38 detect these events (top), which are validated by interphase FISH of each species using fosmid clones spanning the region (bottom). Genomes were included from the following individuals (from top to bottom): bonobo (Pan_paniscus_A915_Kosana, A927_Salonga, A922_Catherine, A917_Dzeeta, A918_Hermien, A924_Chipita, A926_Natalie, A928_Kumbuka, A914_Hortense, A919_Desmond, A925_Bono); chimpanzee (Pan_troglodytes_troglodytes_A958_Doris, A957_Vaillant, A960_Clara, Pan_troglodytes_verus_Clint); orangutan (Pongo_abelii_A950_Babu, Pongo_pygmaeus_A944_Napoleon); gorilla (Gorilla_gorilla_gorilla_KB4986_Katie); human (AFR_Aari_ETAR005_F, AMR_Nahua_Mex20_M, EA_Mongola_HGDP01228_M, SA_Kalash_HGDP00328_M, WEA_FinlandFIN_HG00360_M).

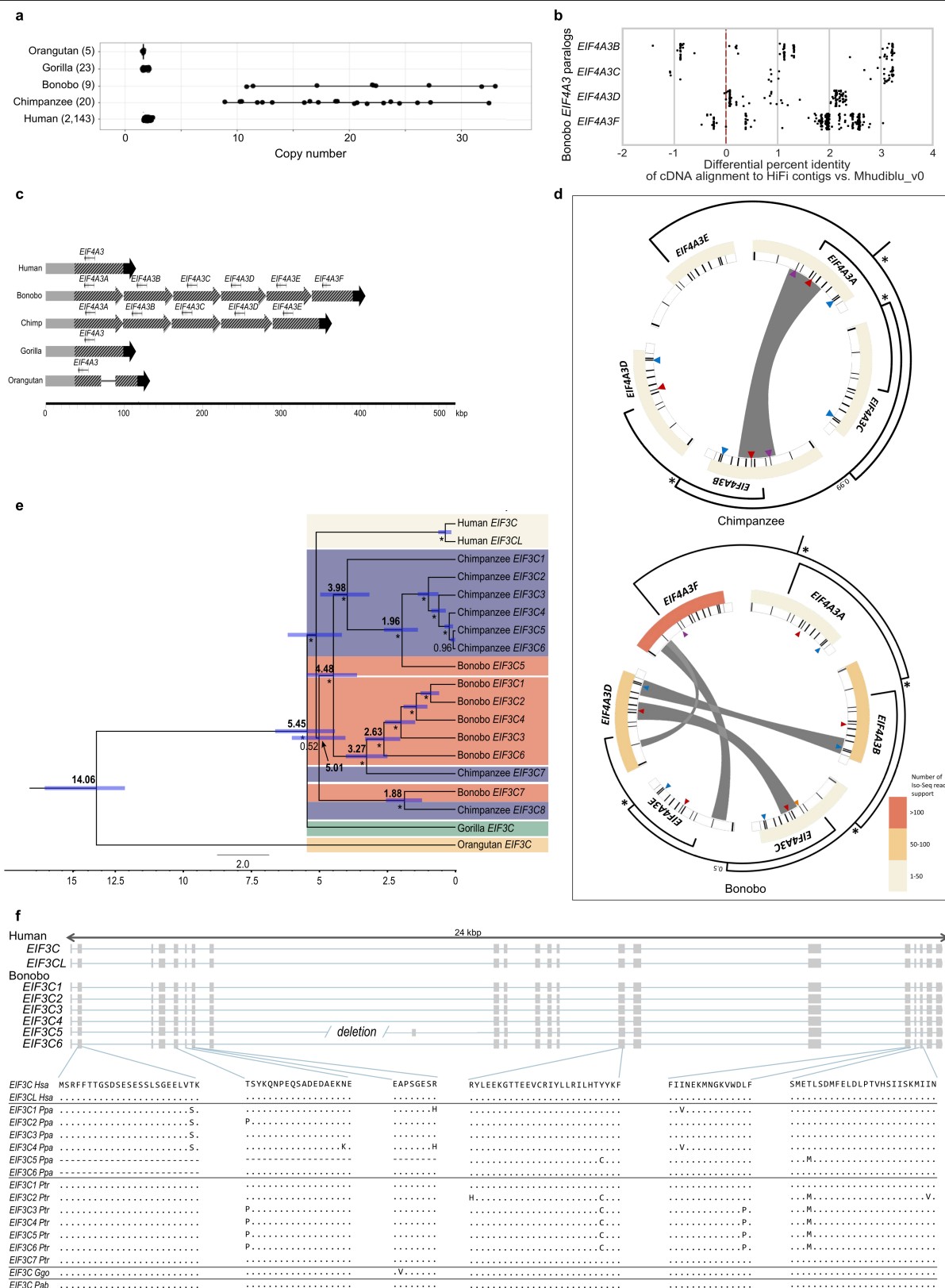

**Extended Data Fig. 5** | See next page for caption.

**Extended Data Fig. 5 | *EIF4A3* and *EIF3C* gene family expansion and sequence resolution. a**, A comparison of *EIF4A3* copy number among great apes based on a sequence-read-depth analysis confirms a variable copy number expansion in the bonobo and chimpanzee lineages (9–33 diploid copies). This recent duplication was not fully resolved initially in the bonobo reference genome (Mhudiblu_PPA_v0) because high-identity duplicated sequences were collapsed. **b**, Bonobo Iso-Seq full-length transcript reads map with higher identity to four of the paralogues compared to Mhudiblu_PPA_v0. **c**, Contigs that encompass *EIF4A3* expansions and 100 kb of the flanking regions were assembled using bonobo and chimpanzee PacBio HiFi data. The 12-kb genomic sequence of human *EIF4A3* mapped onto the assembled contigs. Six tandem copies of *EIF4A3* spanning 310 kb in bonobo and five tandem copies spanning 262 kb in chimpanzee are recovered. Schematics show structural differences in *EIF4A3* in primate genomes. Grey, black and striped arrows show different alignment blocks across the samples. A solid line connecting alignment blocks indicates an insertion event. **d**, Paralogues are expressed and show evidence of gene conversion in both bonobo and chimpanzee lineages. Analysis of bonobo Iso-Seq data confirms that five of the six *EIF4A3* copies are expressed and maintain an open-reading frame (heat map indicates the number of Iso-Seq transcripts supporting each copy; minimap2 -ax splice -G 3000 -f 1000 --sam-hit-only --secondary=no --eqx -K 100M -t 20 --cs -2 | samtools view -F 260). GENECONV software shows significant signals ($P \leq 0.05$ after multiple-test correction) of gene conversion for 16 out of 67 kb of the paralogous locus (grey bars) using multiple sequence alignment was performed using MAFFT version 7.453 (command: mafft -adjustdirection [input.fasta] > [output.msa_fasta]; GENECONV version 1.81a)). A subset of gene conversion events overlap with sites of amino acids that are specific to the *Pan* lineage. Triangles indicate the sites of amino acid change in each of the primate genomes compared to GRCh38. Different colours mark different changes: purple marks phenylalanine to leucine; yellow marks arginine to cysteine; red marks serine to arginine; teal marks tyrosine to serine. Same phylogenetic tree from Fig. 2 is reshaped to show the inferred evolutionary relationships among the paralogues. Nodes with >99% Bayesian posterior probabilities are indicated by asterisks; otherwise the actual number is shown. **e**, A phylogenetic tree was constructed from 16-kb noncoding *EIF3C* paralogues using Bayesian phylogenetic inference. This analysis was conducted using BEAST2 software. Numbers in bold on each major node denote estimated divergence time. The other numbers (not bold) indicate posterior probabilities. The blue error bar on each node indicates the 95% confidence interval of the age estimation. Bootstrap supports are reported using asterisks for nodes with posterior probability >99%. **f**, Gene models for transcribed loci based on Iso-Seq data (top). Human *EIF3C* and *EIF3CL* are compared to predicted open-reading frames for bonobo paralogues and Liftoff gene predictions for chimpanzee, orangutan and gorilla paralogues from contigs assembled from HiFi reads (bottom).

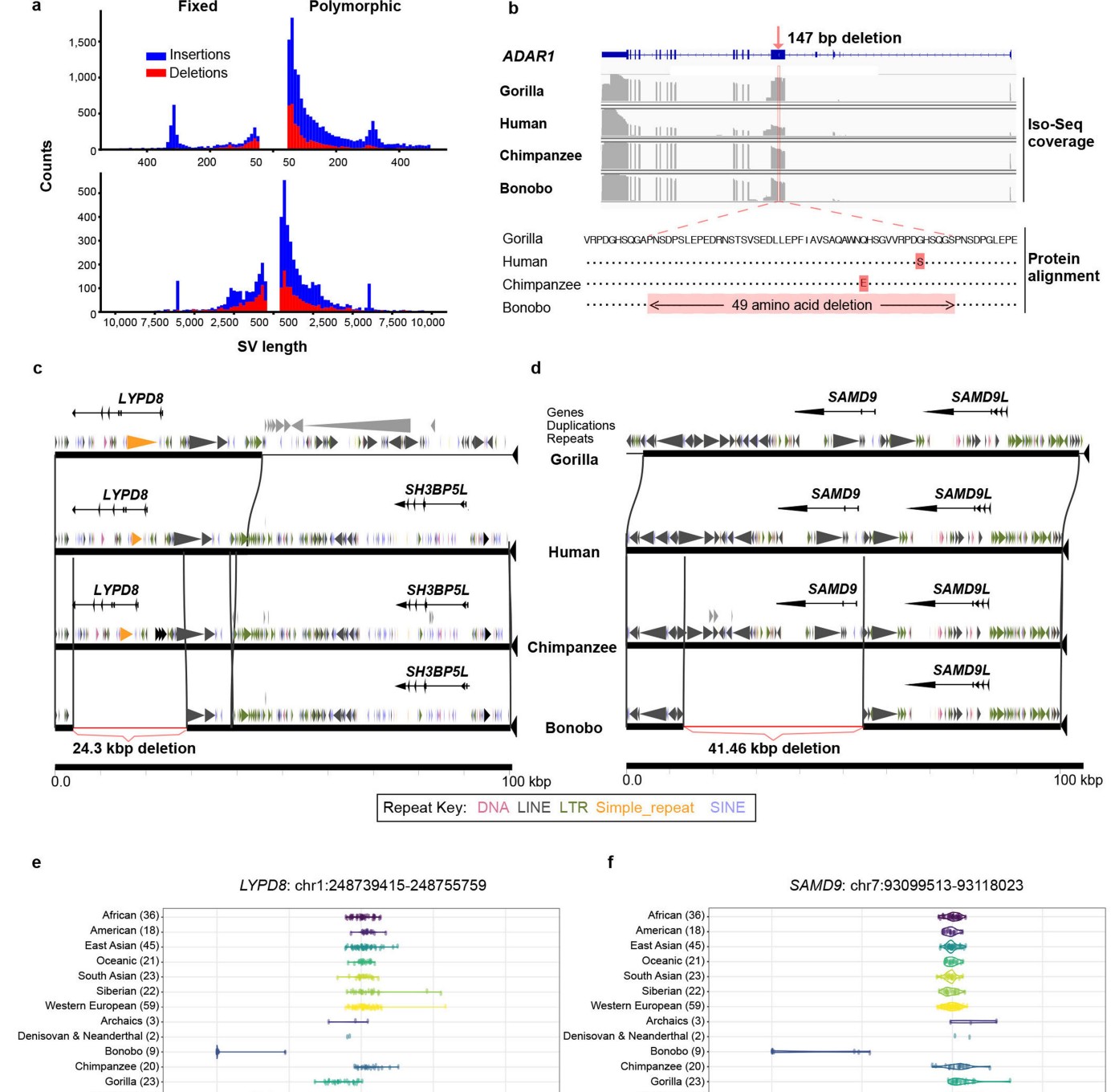

**Extended Data Fig. 6 | Bonobo structural variants and gene deletions. a**, Size distribution of fixed (left) and polymorphic (right) structural variant (SV) insertions and deletions in the bonobo genome for structural variants of 50–1,000 bp (top) or >1,000 bp (bottom) in length. Events are deemed to be specific to the bonobo lineage based on copy number genotyping against a panel of 27 ape genomes and a threshold of $F_{ST} > 0.8$ to define fixed events in bonobo. Modes are observed corresponding to full-length L1 (6 kb) and Alu (300 bp) mobile elements and are predominantly insertions reflecting the homoplasy-free nature of this class of mutation. **b**, A small fixed deletion predicts a 49 amino acid deletion in *ADAR1* in the bonobo lineage. RefSeq *ADAR1* structure is shown (top) compared with the Iso-Seq coverage of gorilla,

human, chimpanzee and bonobo (middle). The protein alignment (bottom) shows that an in-frame deletion is created. **c**, A 24.3-kb fixed deletion results in the complete loss of *LYPD8* in bonobo. Gene structure, duplication and repeat annotations are shown with respect to gorilla, human, chimpanzee and bonobo genomes. A lineage-specific duplication adjacent to *LYPD8* is present in the gorilla genome (large grey triangles). **d**, A 41.5-kb fixed deletion mediated by directly orientated L1 repeats ablates *SAMD9* leaving only *SAMD9L* in the bonobo lineage. **e**, Short-read whole-genome shotgun detection genotyping shows that *LYPD8* was lost in the bonobo lineage. **f**, Short-read whole-genome shotgun detection genotyping shows *SAMD9* was lost in the bonobo lineage.

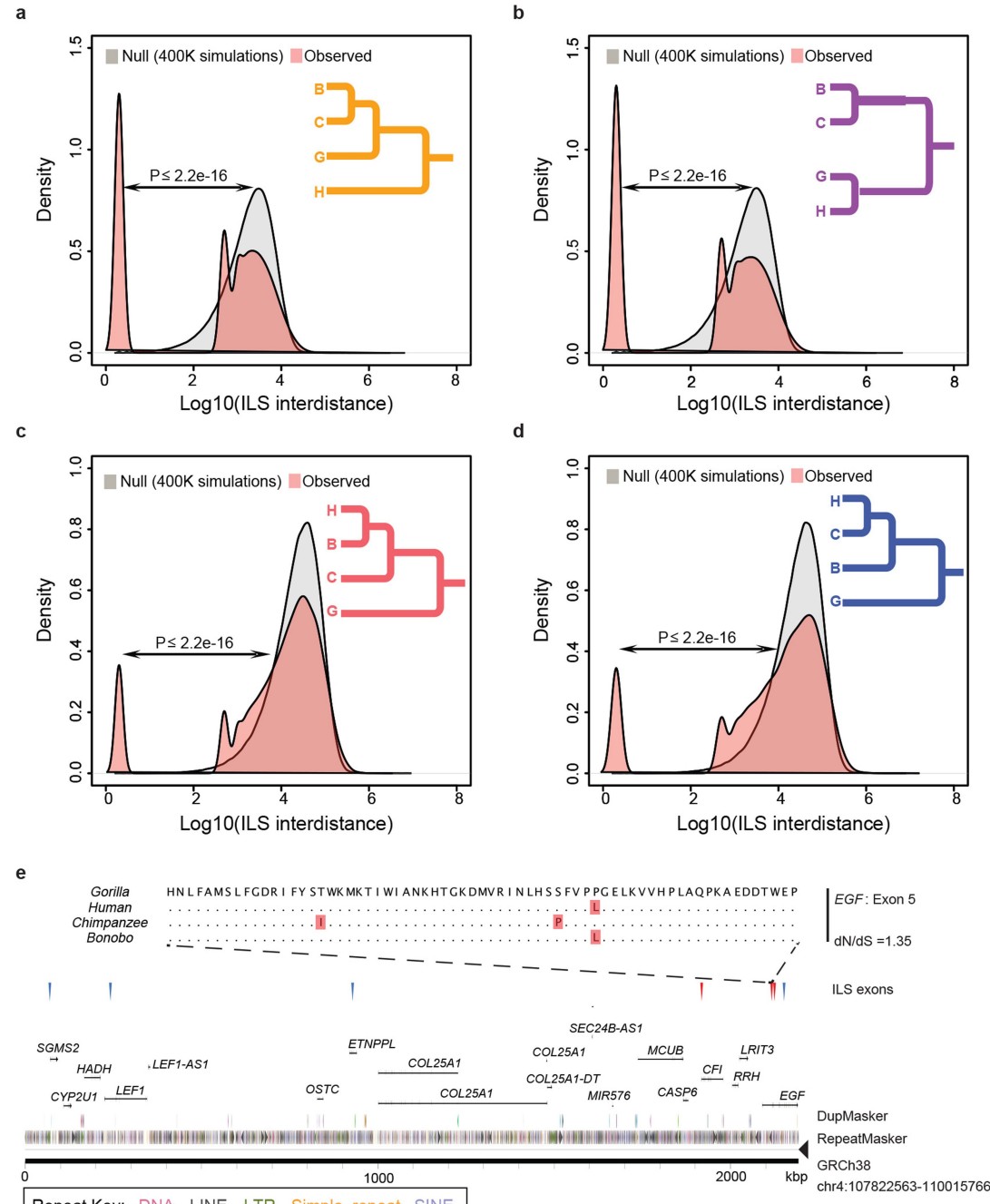

**Extended Data Fig. 7 | Hominid ILS.** The distance between adjacent ILS segments (inter-ILS) (500-bp resolution) was calculated and the distribution was compared to a simulated expectation based on a random distribution. The analysis reveals a bimodal (and possibly an emerging trimodal) pattern in which a distinct subset of ILS segments are clustered (that is, clustered ILS sites). Four different topologies were considered. **a**, A (orangutan, (((bonobo, chimpanzee), gorilla), human)) ILS topology in which 31.58% of inter-ILS is clustered is shown. **b**, A (orangutan, ((bonobo, chimpanzee), (gorilla, human)))

ILS topology in which 33.5% is clustered is shown. **c**, A (orangutan, (((bonobo, human), chimpanzee), gorilla)) ILS topology in which 8.14% is clustered is shown. **d**, A (orangutan, ((bonobo, (chimpanzee, human)), gorilla)) ILS topology in which 9.89% of sites is clustered is shown. **e**, An example of a cluster of human–bonobo (red triangles) and human–chimpanzee (blue triangles) ILS corresponding to a group of genes. A four-species alignment of one exon from *EGF* (exon 5) is shown with a nominal signal of positive selection.

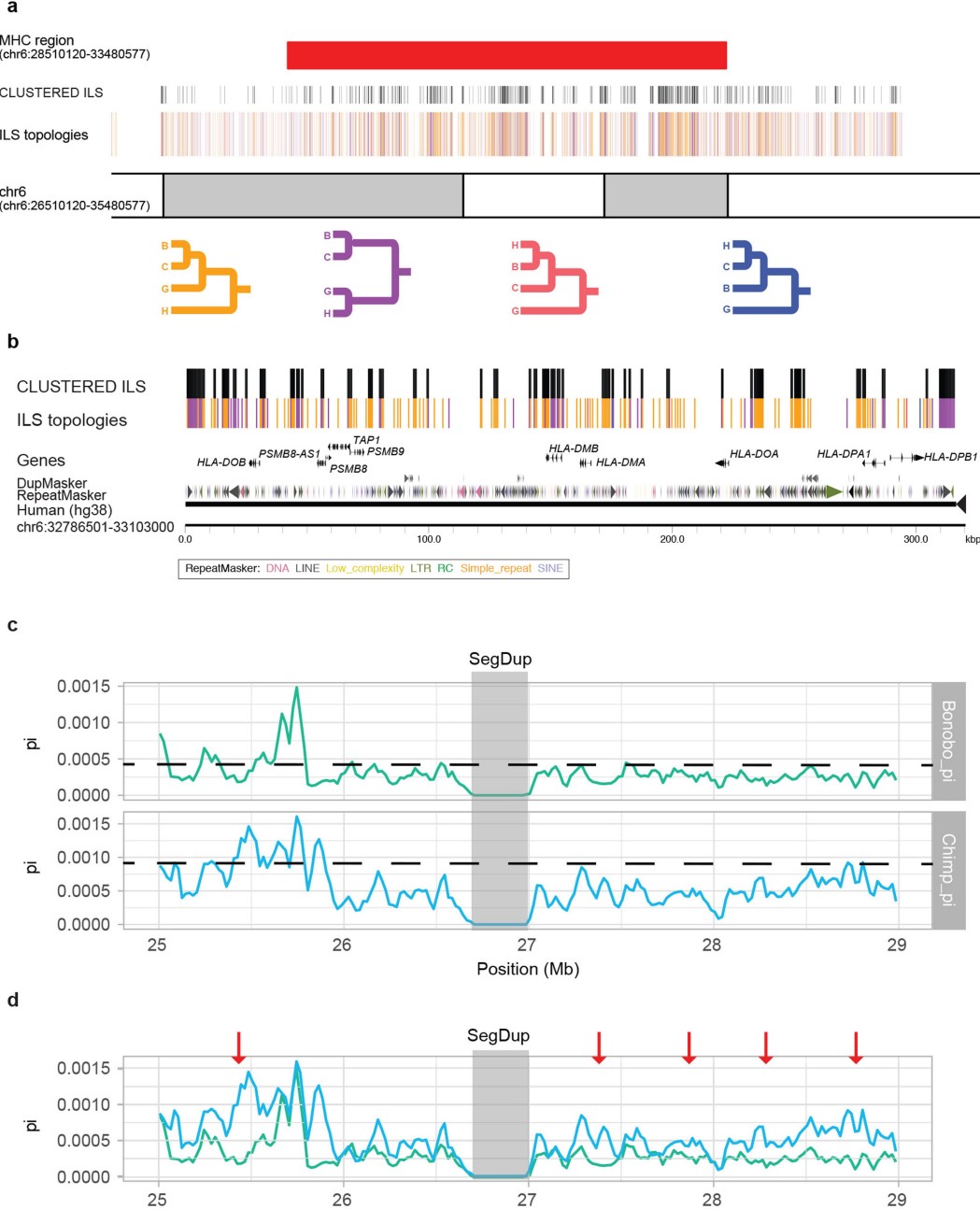

**Extended Data Fig. 8 | Ideogram of the MHC region with ILS annotations.** **a**, The four main ILS topologies are colour-coded. The four colour lines representing ILS segments are shown above the chromosome coordinate (GRCh38). The clustered ILS segments are shown above the four colour lines (black). The MHC region (red bar) corresponds to genomic coordinates on chromosome 6: 28510120–33480577. **b**, A magnified view of the MHC region (chromosome 6: 32786501–33103000) depicting clustered ILS nearby *HLA* genes. **c**, Nucleotide diversity of bonobo (green) and chimpanzee (blue) is shown based on human genomic coordinates (GRCh38, chromosome 6: 25000000–29000000). The mean (dashed line) is shown for bonobo

(mean = $4.45 \times 10^{-4}$) and chimpanzee (mean = $9.35 \times 10^{-4}$). A region of reduced diversity (grey) is shown that corresponds to a segmental duplication in which single-nucleotide polymorphisms were excluded due to potential mismapping. **d**, Same as **c** but merged onto the same scale and highlighting five regions (red arrows) in which diversity is reduced in bonobo compared to chimpanzee. Three of these correspond to previously identified regions[1]; however, they are not among the top 1% of genome candidates showing positive selection by Tajima's *D* and SweepFinder2[15]. The overall diversity of single-nucleotide polymorphisms is reduced across the region in bonobo compared to chimpanzee.

# Reporting Summary

Nature Research wishes to improve the reproducibility of the work that we publish. This form provides structure for consistency and transparency in reporting. For further information on Nature Research policies, see our Editorial Policies and the Editorial Policy Checklist.

## Statistics

For all statistical analyses, confirm that the following items are present in the figure legend, table legend, main text, or Methods section.

| n/a | Confirmed | |
|---|---|---|
| ☐ | ☒ | The exact sample size (*n*) for each experimental group/condition, given as a discrete number and unit of measurement |
| ☐ | ☒ | A statement on whether measurements were taken from distinct samples or whether the same sample was measured repeatedly |
| ☐ | ☒ | The statistical test(s) used AND whether they are one- or two-sided<br>*Only common tests should be described solely by name; describe more complex techniques in the Methods section.* |
| ☐ | ☒ | A description of all covariates tested |
| ☐ | ☒ | A description of any assumptions or corrections, such as tests of normality and adjustment for multiple comparisons |
| ☐ | ☒ | A full description of the statistical parameters including central tendency (e.g. means) or other basic estimates (e.g. regression coefficient) AND variation (e.g. standard deviation) or associated estimates of uncertainty (e.g. confidence intervals) |
| ☐ | ☒ | For null hypothesis testing, the test statistic (e.g. *F*, *t*, *r*) with confidence intervals, effect sizes, degrees of freedom and *P* value noted<br>*Give P values as exact values whenever suitable.* |
| ☐ | ☒ | For Bayesian analysis, information on the choice of priors and Markov chain Monte Carlo settings |
| ☐ | ☒ | For hierarchical and complex designs, identification of the appropriate level for tests and full reporting of outcomes |
| ☒ | ☐ | Estimates of effect sizes (e.g. Cohen's *d*, Pearson's *r*), indicating how they were calculated |

*Our web collection on statistics for biologists contains articles on many of the points above.*

## Software and code

Policy information about availability of computer code

| | |
|---|---|
| Data collection | Pacific Bioscience Sequel II Instrument Control SW (v7.1 or v8.0) and Leica Application Suite X (v3.7). |
| Data analysis | We applied Falcon (git id 53444482, dgordon branch available on 2017.06.13) to assemble the bonobo genome from SMRT sequence reads. The assembly was error-corrected using Quiver (version 0.7.6) and then further error-corrected using Pilon (version 1.21). The contigs were placed into scaffolds using the HybridScaffolds suite (pipeline version 4573) from the BioNano Genomics. Access software (pipeline version 4573, and RefAligner version 7376). To assign each contig/scaffold into unique groups corresponding to individual chromosomal homologues we used SaaRclust (github daewoooo branch available on Mar 3, 2019, version 0.99). We aligned available Strand-seq data to the Mhudiblu assembly (v0) using the BWA aligner (version 0.7.17-r1188) with default parameters for paired-end mapping. Subsequently, we used sambamba (version 0.6.8) in order to mark duplicated reads and SAMtools (version 1.9) to sort and index the final BAM file for each Strand-seq library. Segmental Duplication Assembler (SDA) (github mrvollger branch available on Mar 31, 2020) was used to identify and unpack collapsed SDs in the bonobo assembly. Repeat content of the assembled genome was analyzed using RepeatMasker (RepeatMasker-Open-4.1.0) and the Dfam3 repeat library. We assigned lineage-specific Alu and full-length LINE, SVA_D and PTERV elements to subfamilies by applying COSEG (www.repeatmasker.org/COSEGDownload.html) to determine the lineage specific subfamily composition. Genome annotation was performed using the Comparative Annotation Toolkit (CAT) v2.1. Insertions and deletions were detected in bonobo, chimpanzee and gorilla using PBSV (version 2.2.0), Sniffles (version 1.0.10) and Smartie-sv (github zeeev branch available on Mar 8, 2018) and genotyped using Paragraph (version 2.4a) against a panel of 27 Illumina WGS genomes. We searched for evidence of ILS among the chimpanzee, gorilla and human lineages applying Prank (v.140110) to construct multiple sequence alignments and using ete3 module to identify segments under ILS. All statistics analyses were performed in R (3.5.3). We applied minimiro (github mrvollger branch available on Aug 4, 2020) for plotting synteny. Splign (NCBI updated on 02/23/15) was used for gene annotation. QV analyses were run with Merqury (version 1.0). Custom codes used in this study are available at GitHub (https://github.com/EichlerLab and https://github.com/MaoYafei). |

For manuscripts utilizing custom algorithms or software that are central to the research but not yet described in published literature, software must be made available to editors and reviewers. We strongly encourage code deposition in a community repository (e.g. GitHub). See the Nature Research guidelines for submitting code & software for further information.

## Data

Policy information about availability of data

All manuscripts must include a data availability statement. This statement should provide the following information, where applicable:
- Accession codes, unique identifiers, or web links for publicly available datasets
- A list of figures that have associated raw data
- A description of any restrictions on data availability

All the data and accession codes have been reported in the manuscript

# Field-specific reporting

Please select the one below that is the best fit for your research. If you are not sure, read the appropriate sections before making your selection.

☐ Life sciences ☐ Behavioural & social sciences ☒ Ecological, evolutionary & environmental sciences

For a reference copy of the document with all sections, see nature.com/documents/nr-reporting-summary-flat.pdf

# Ecological, evolutionary & environmental sciences study design

All studies must disclose on these points even when the disclosure is negative.

| | |
|---|---|
| Study description | We sequenced and assembled a new bonobo reference genome using a multiplatform approach. The genome is more contiguous and accurate allowing more comprehensive sequence alignement. We discovered new species specific structural variants including gene family expansions and deletions in the ape lineage. We provide a more complete view of incomplete lineage sorting and its non-random clustering during ape genome evolution. |
| Research sample | We sequence a bonobo (Pygmy chimpanzee) immortalised cell line (Carbone #601152). The source of the cells was an EBV transformed lymphoblast cell line from a single female bonobo, Mhudiblu. Pygmy chimpanzee was chosen because of its importance for inferring species specific changes in both human and chimpanzee lineages. Together with chimpanzee, bonobos represent the closest great apes to human genome. The sample we sequenced is representative of Pan paniscus. |
| Sampling strategy | No sample size calculation was performed. We were searching for genomics and transcriptomics similarities/differences between Pan paniscus and other great ape genomes. For this purpose, deep whole genome long-read sequencing with the Pacific Biosciences Sequel II platform was performed and variants were then genotyped on a population of samples to confirm fixed or polymorphic status. |
| Data collection | Sequencing data for assembly were collected using Pacific Bioscience Sequel II Instrument Control SW (v7.1 or v8.0); while cytogenetics data were generated using a Leica fluorescence microscope and Leica Application Suite X (v3.7). |
| Timing and spatial scale | No Timing or spatial scale was applied |
| Data exclusions | No data were specifically excluded |
| Reproducibility | Computational experiments are deterministic and are, therefore, reproducible. Despite this expected reproducibility, computational experiments were performed multiple times with different parameters and followed up with experimental validation. All attempts at replication were successful. |
| Randomization | No randomization was performed, being a single genome sequenced and assembled. |
| Blinding | No blinding was requested, being a single genome sequenced and assembled. |

Did the study involve field work? ☐ Yes ☒ No

# Reporting for specific materials, systems and methods

We require information from authors about some types of materials, experimental systems and methods used in many studies. Here, indicate whether each material, system or method listed is relevant to your study. If you are not sure if a list item applies to your research, read the appropriate section before selecting a response.

## Materials & experimental systems

| n/a | Involved in the study |
|-----|----------------------|
| ☒ | Antibodies |
| ☐ | ☒ Eukaryotic cell lines |
| ☒ | Palaeontology and archaeology |
| ☒ | Animals and other organisms |
| ☒ | Human research participants |
| ☒ | Clinical data |
| ☒ | Dual use research of concern |

## Methods

| n/a | Involved in the study |
|-----|----------------------|
| ☒ | ChIP-seq |
| ☒ | Flow cytometry |
| ☒ | MRI-based neuroimaging |

# Eukaryotic cell lines

Policy information about cell lines

| | |
|---|---|
| Cell line source(s) | Mhudiblu (SAMN11123633), PPA Lb502 (SAMN01920504), chimpanzee (Pan troglodytes; Clint; S006007), gorilla (Gorilla gorilla; GGO9), Orangutan (Pongo abelii; Susie; PR01109) and human normal donor (with signed personal consent). All the origin of the great apes individuals we tested are reported at this link: https://www.biologiaevolutiva.org/greatape/samples.html and are available upon request according to CITES restrictions.<br>Mhudiblu (SAMN11123633) immortalized by EBV transformed lymphoblast cell line (Carbone #601152), was originally isolated from a single female bonobo (Pan paniscus), Mhudiblu (a.k.a. Mhudibluy, ISIS 601152, born April 2001 at San Diego Zoo or Muhdeblu when she was transferred to the Wuppertal Zoo in Germany).<br>PPA Lb502 (SAMN01920504), immortalized by EBV transformed lymphoblast cell line was obtained from a captive born animal and donated by Prof. Mariano Rocchi, University of Bari (Italy). GGO9 fibroblast cell lines were donated by Prof. Mariano Rocchi, University of Bari (Italy).<br>Chimpanzee (Pan troglodytes; Clint; S006007) fibroblast cells were originally obtained from a male Western chimpanzee named Clint (now deceased) at the Yerkes National Primate Research Center (Atlanta, GA) and immortalized with EBV.<br>Orangutan (Pongo abelii; Susie; PR01109) fibroblast cells were originally obtained from a female Sumatran orangutan named Susie (now deceased) at the Gladys Porter Zoo (Brownsville, TX), immortalized with EBV, and stored at the Coriell Institute for Medical Research (Camden, NJ). |
| Authentication | Mhudiblu was sequenced with Illumina whole-genome sequencing to confirm species and karyotyped. The other cell lines were authenticated via standard karyotype analysis. |
| Mycoplasma contamination | All the cell lines tested negative for Mycoplasma |
| Commonly misidentified lines<br>(See ICLAC register) | None |

