## [Peer Review File · Nature]

Manuscript Title: A high-quality bonobo genome refines the analysis of hominid evolution

Editorial Notes:

Reviewer Comments & Author Rebuttals

Reviewer Reports on the Initial Version:

Referee #1 (Remarks to the Author):

This manuscript reports a new reference assembly and series of analyses of the bonobo (*Pan paniscus*) genome. Bonobos are a critically significant species by virtue of their phylogenetic position as sister species to the common chimpanzee. This makes bonobos equally close phylogenetically to humans as are common chimpanzees (*Pan troglodytes*). But a variety of morphological and behavioral differences between bonobos and chimpanzees makes comparisons between this species and humans and all the great apes highly informative. This genome assembly seems to have been done thoroughly and accurately. The methods used are state-of-the art, and the authors employed the orthogonal technologies of optical mapping, FISH mapping and BAC sequencing to validate and improve the initial Falcon assembly of PacBio long reads. The analyses performed are extensive, interesting and will be very useful to both the comparative genomics and primatology communities. The annotation of protein coding genes and non-coding genes was done with two independent pipelines (NCBI and CAT) which is beneficial.

The authors provide a number of distinct and valuable downstream studies of this new reference genome. The analyses of ILS are particularly interesting and significant. This evolutionary process is increasingly recognized as a common phenomenon among mammals that can influence genome evolution, species differences and thus phenotypic evolution. The case of human-chimpanzee-bonobo-gorilla provides an outstanding example of the potential for ILS and this manuscript documents and explores several aspects of ILS and its consequences in detail. The results have implications for our understanding of the origin of human-specific genomic and phenotypic adaptations, and thus have particularly broad impact. The analyses of mobile element insertions and small structural variations are also valuable and informative. Overall this is an extensive and detailed analysis that provides new genomic information applicable to a number of different topics and questions. However, I do have specific comments regarding the paper.

Major comments

- 1) I am a bit confused about the details of assembly statistics for the initial assembly Mhudiblu_PPA_v0. In line 130 and the Suppl Note Table S3 the contig N50 for PPA_v0 is reported as 17.99 Mb. But both Table 1 of the main text and Suppl Table S29 indicate PPA_v0 has a contig N50 of 16.58 Mb. The information in the NCBI accession matches this latter statistic. It is not clear why the longer N50 of 17.99 Mb is presented on line 130 and in Suppl Table S3.
- 2) Given that PPA_v1 incorporates a number of useful corrections to PPA_v0, it is not clear why PPA_v0 is the assembly of record in NCBI. Can the authors demonstrate that PPA_v1 is forthcoming in NCBI? Can they include the accession number in this manuscript?
- 3) Lines 146-147: It would be valuable to state the pairwise nucleotide divergence separating the new bonobo assembly from the latest chimpanzee assembly and from GRCh38/hg38.

4) Lines 187-192: The authors should report whether there are GO categories enriched among the genes that show either gene family expansion or contraction in bonobos relative to humans or chimpanzees and especially gene family changes unique to bonobos.

5) Structural variation. The analysis of SVs is extensive and important. I recommend that the authors look at whether there is significant overlap between genomic regions that contain bonobo-specific SVs and the regions identified as introgressed into bonobos from an ancient extinct great ape (Kuhlwiilm et al. 2019, PMID31036897). Similarly, it would be useful to compare the inferred introgressed regions against the regions that show bonobo-specific changes in gene number (protein coding gene family expansions or contractions).

6) Line 478: Does the level of ILS observed have implications for inferring the effective population size of the LCA of bonobos and chimpanzees? I would not suggest that a full analysis of demographic history should be included in this report, but the authors might comment on the potential inference that N_e was large in the LCA, given that polymorphisms were retained from the time of divergence from humans about 7 mya to the divergence of bonobos from chimpanzees about 1.7 mya.

7) One of the most interesting aspects of evolution of the genus *Pan* is the behavioral differences between chimpanzees and bonobos. As described in Staes et al (2019) which is reference #7 in the present Catacchio et al. manuscript, there are specific types of social behavior and specific neural circuits that differ in bonobos vs chimpanzees. While a complete analysis is probably beyond the scope of this report, can the authors provide any novel information about newly identified genomic differences between these two species that may be related to the neuroanatomical or behavioral differences described by Staes et al.? In particular, the authors here focus on a unique deletion in the coding sequence of ADAR1 in bonobos, not shared with chimpanzees. The present authors also state that this gene may have some relationship with neurotransmitter function, and I would note that bonobos seem to have a different level serotonergic innervation in relevant neural circuits as compared with chimpanzees (Stimpson et al. 2016, PMID26475872).

Minor comments

a) In my opinion, the pale yellow lines representing deletions in Fig 1 are too pale, difficult to see.

b) The track hub from line 137 was not accessible from my browser. I can see only a text file with non-useful labels and an email address.

c) Line 179: It is not clear to me why the average increase of complete isoforms is reported as a range (1.5% to 2.1%). Why is the average not a single value? Are these the two values for NCBI and CAT annotations respectively?

d) It is not clear to this reviewer what the differences are between Suppl Tables S10, S11 and S12 in the Excel file (copy number differences bonobo vs other species). More explanation of the differences seems warranted.

e) Lines 390-391: Should the number in line 390 be 84 instead of 88 (40 + 44)?

Referee #2 (Remarks to the Author):

Catacchio et al. present a substantially-improved bonobo reference genome sequence, facilitating the analysis of polarized bonobo- and chimpanzee- lineage specific gene content variation and (with the human genome) incomplete lineage sorting events. The potential strength of the paper is not the genome assembly itself; rather, analyses that could be facilitated by the availability of a high-quality

reference genome. This is where my review comments are focused. Otherwise, the descriptive results are important to establish and generally quite solid in this paper, but they are largely accretive relative to those from the prior bonobo genome assembly and our general understandings of genome variation at this point.

Overall I was left wanting for more analytical depth and novel impact from the study. If the paper is given further consideration at Nature, I would expect deeper analyses of the structural variation, gene content variation, and incomplete lineage sorting data.

1. Gene content variation. I agree that gene loss events are potentially highly informative. Yet at present the new insight from this study is largely restricted to the development of several hypotheses in the discussion section. It may be that there were not any strongly emergent patterns from the data, but at the least a comprehensive investigation across the phylogeny on a lineage-specific (including the ancestral Pan lineage) basis should be considered and reported.

2. Incomplete lineage sorting (ILS). The statement "5.07% of the human genome has been subject to ILS" is confusing in two different ways. First, the segments with topologies of (human, bonobo), chimpanzee)) and (human, chimpanzee), bonobo)) reflect the long-term maintenance of ancestral polymorphism over the ancestral Pan lineage. Second, a much higher proportion of the human genome than this reflects incomplete lineage sorting, with estimates in the 20-30% range for the sum of (human, gorilla), chimpanzee)) and (chimpanzee, gorilla), human)).

2b. This also seems to represent a major missed opportunity to consider ILS simultaneously across these different scales, which would be more informative for understanding the process and significance; e.g. do they overlap more than expected by chance? If so, does this reflect repeat occurrence in the same region, maintained polymorphism potentially as a function of long-term balancing selection, what functional categories of genes are enriched (genes encoding proteins involved in immune responses?), potentially more insightful dN/dS analyses, integration with human, chimpanzee, bonobo, and gorilla population genomic datasets (genetic diversity, signatures of positive selection), etc. The functional enrichment analyses of the (H,B,C)) vs. (H,C),B)) regions alone are not that informative or interpretable.

Minor comment:

3. The genome assembly itself is fine, but is not as leading edge as proclaimed/implied by the text. Thus, I recommend revising the list of necessary methodological steps for high-quality assembly construction in the introduction; i.e., not only the pathway combination used in this case. I focused my review on the analytical components of the manuscript.

Referee #3 (Remarks to the Author):

Catacchio et al describe a reference grade bonobo genome assembly obtained through a mix of technologies, including PacBio Hi-Fi, optical mapping, and Strand-Seq. This assembly is a significant technical improvement over the previous version, which was produced using an older technology (Roche 454) and hence was fragmented. The new assembly thus enables the study of repeat elements (Sine, Alu, ERV), segmental duplications, and inversions.

Bonobo is particularly interesting due to its recent speciation from chimp, which provides a vantage point to hominid evolution. Capitalizing on their improved assembly, the authors spotlight several structural variants, including an exon deletion in ADAR1 and whole gene deletions (SAMMD9, LYPD9), as having potential biological significance. The most interesting findings relate to ILS, which appears to cluster in the genome and is enriched in specific pathways (photoreceptor for human-bonobo, EGF pathway for

human-chimp ILS). ILS segments, particularly those that cluster, also have higher dN/dS.

Though the work is an impressive technical feat, it is not the first to use long-read sequencing to generate reference-grade assemblies without the help of the human genome. It is also not the first assembly of the bonobo genome, albeit a much higher quality assembly than the one previously published in 2012 by the Paabo group.

The novelty then rests on the biological implications of the improved bonobo reference, and presumably the new insight that can be gained into hominid evolution. As it stands, however, the biological insights gained from this improved bonobo assembly seem minor.

Indeed, the authors are able to reconstruct many more instances of repeat elements (L1, Alu, SVA, PtERV1) and structural variants (segmental duplications, inversions), than before. However, the highlighted examples seem cherry-picked rather than nominated by a statistical model or rigorous genome-wide analysis. These anecdotes spark plausible but speculative hypotheses about the role of specific pathways (e.g. gut homeostasis or pox virus susceptibility) in bonobo / hominid evolution. The observation of the EIF3A segmental duplication is striking, but is not developed into a story. It seems that the ILS findings, including the clustering and increased dN/dS at these loci, provide the most potential for a biologically compelling narrative. However, the extent of ILS in bonobo-chimp speciation has been previously discussed (including in Prufer 2012, to considerable depth). As a result, the ILS insights seem somewhat incremental.

Specific critiques / questions:

- * Are there any new targets of positive selection? Are any of the specific variant classes (in particularly those resolved to higher fidelity with the new assembly) driving positive selection signals?
- * What is the landscape of ILS beyond coding regions? How often do these "ILS clusters" cross gene boundaries? It may be interesting to intersect some of these non-coding patterns with human regulatory annotations (eg ENCODE, Hi-C) or disease annotations (GWAS).
- * Are there regions statistically depleted in ILS suggesting selective sweeps?
- * What is the role of SVs and repeat elements in ILS? Could this be used to say something about selection on acting on these variant classes?
- * The EIF3A results are striking, but left as an isolated observation. Can the authors expand on this finding? For example, can something be said about the locus architecture, sequence features, or dynamics of this and other SD's, or their regional distribution now that they have been placed into scaffolds. Are there more that are as high level as this one?
- * More broadly, what new insight does the increased resolution of SD's in the bonobo genome give into the dynamics of SD's and gene family expansions in great apes beyond the 2009 Marques-Bonet et al paper - in particular these very high amplitude SD's. Since these are resolved at breakpoint resolution, there should be opportunities to illustrate how some of these loci are evolving in the great ape lineages. For example, EIF3A is duplicated in both chimp and bonobo to different numbers of copies - when did the individual duplications occur relative to speciation. Also, is there anything special about EIF3A that would select for this - for example, does this locus undergo SD in other mammalian lineages?
- * Prufer 2012 used the bonobo genome to show evidence of chimpanzee selective sweeps in the MHC locus and other regions. They also show that MHC is the most frequent target of ILS. But there is no mention of MHC in this work, which is surprising given these previous findings and how important MHC is in human biology. Can the authors revisit this analysis using the new assembly? Is the previous signal

missing? Can the authors confirm / revise the prior findings?

* Could the Dn/Ds-high ILS clusters be the result of missassemblies? The authors should demonstrate that ILS high vs poor regions have the same degree of assembly quality.

Minor critiques

* Supp Note Table S35 uses commas instead of periods for decimals.

Referee #4 (Remarks to the Author):

I would like to congratulate Catacchio et al., for presenting a new high-quality Bonobo genome and for treating the analysis and the presentation of the results with so much rigor and care. The manuscript presents a chromosome level genome for *Pan paniscus* – the last of the great apes to be sequenced with long-reads – where a great portion of the gaps were closed and genes were fully annotated, and half of the segmental duplications were assembled as well. They have also presented a new set of bonobo exclusive genes, have described novel gene models in the bonobo assembly thought to be related exclusively to human adaptation and have done all of this research taking into consideration IsoSeq sequencing for confirmation to these new findings. A number of segmental duplications and the chromosome fusion were further tested and confirmed with FISH experiments which brings great confidence to these findings. The work also presented a higher resolution analysis of ILS showing that a greater fraction of the hominid genome is under ILS, unlike what was estimated previously. Because of all that stated above, I consider that this work is innovative and presents a rich resource for experimental biologists who will have plenty of material to target novel genomic areas and further advance our understanding of hominid evolution and gene function.

In terms of the genome assembly – which is my main area of expertise – one point that concerns me a little is the QV ranging from 35-39 (estimated by kmer-sharing and BACs sequence comparisons). The truth is that technologies evolve, and it is likely that a 30x coverage of Pacbio HiFi would be able to take the QV to >50 and would most likely solve the remaining unresolved Segmental Duplications of this assembly. The same is true considering Hi-C reads that – particularly if sequenced from the same individual – would have high resolution to determine unconfirmed internal structure. That said, because the authors had extreme care with their claims and the genome presented is a huge improvement over the last one available, this genome should be available to the scientific community as it is and it supports the claims made by the authors. I would like to advise the authors, however, to have a look at the .bed intermediate output of merquy. This file contains the coordinates of kmers present only in the assembly, meaning they are not shared with the Illumina reads. As the authors have done so much already, it would not be too much trouble to estimate if these unique-assembly kmers are more frequent in specific genomic areas such as repeats. Further, it would be important to check if those possibly-erroneous-assembly-kmers are present in any of the 111 genes that have potential frameshifting indels that disrupt their primary isoform relative to the human reference. In addition, I would like to see a supplementary figure with the kmer plot distribution of the illumina reads used for the short-reads polishing and for the merquy QV estimation.

My two last considerations would be to (i) ask the authors to confirm they have checked that the further curated Mhudiblu_PPA_v1 version of the assembly has not disrupted any genes that the authors have investigated in Mhudiblu_PPA_v0 and described in their results. And on that point, I would suggest the authors to maybe (ii) include a last supplementary figure representing a genome assembly fluxogram – going from the Falcon assembly, pointing out the manual interventions, annotations and further improvements all the way from reads to Mhudiblu_PPA_v0 and Mhudiblu_PPA_v1. The supplementary material presented is already a great documentation for reference, but it is extensive. This added

fluxogram would be a good historical reference of the steps taken to assemble this version of the bonobo genome, and would greatly help future scientists who will be looking to further improve this assembly to find the regions more likely to contain errors.

Once more, I congratulate the authors in their great effort and relevant piece of science presented. I wish them success.

Author Rebuttals to Initial Comments:

To avoid confusion with references we reported in the rebuttal letter citations using PMID codes, while in the two other documents (main text and Supplementary Note) we used the standard Nature format.

Referees' comments:

Referee #1 (Remarks to the Author):

This manuscript reports a new reference assembly and series of analyses of the bonobo (*Pan paniscus*) genome. Bonobos are a critically significant species by virtue of their phylogenetic position as sister species to the common chimpanzee. This makes bonobos equally close phylogenetically to humans as are common chimpanzees (*Pan troglodytes*). But a variety of morphological and behavioral differences between bonobos and chimpanzees makes comparisons between this species and humans and all the great apes highly informative. This genome assembly seems to have been done thoroughly and accurately. The methods used are state-of-the art, and the authors employed the orthogonal technologies of optical mapping, FISH mapping and BAC sequencing to validate and improve the initial Falcon assembly of PacBio long reads. The analyses performed are extensive, interesting and will be very useful to both the comparative genomics and primatology communities. The annotation of protein coding genes and non-coding genes was done with two independent pipelines (NCBI and CAT) which is beneficial.

The authors provide a number of distinct and valuable downstream studies of this new reference genome. The analyses of ILS are particularly interesting and significant. This evolutionary process is increasingly recognized as a common phenomenon among mammals that can influence genome evolution, species differences and thus phenotypic evolution. The case of human-chimpanzee-bonobo-gorilla provides an outstanding example of the potential for ILS and this manuscript documents and explores several aspects of ILS and its consequences in detail. The results have implications for our understanding of the origin of human-specific genomic and phenotypic adaptations, and thus have particularly broad impact. The analyses of mobile element insertions and small structural variations are also valuable and informative. Overall this is an extensive and detailed analysis that provides new genomic information applicable to a number of different topics and questions. However, I do have specific comments regarding the paper.

Thank you. We appreciate the reviewer's recognition of the importance and impact of this work to the wider genomics and primate research community.

Major comments

1) I am a bit confused about the details of assembly statistics for the initial assembly Mhudiblu_PPA_v0. In line 130 and the Suppl Note Table S3 the contig N50 for PPA_v0 is

reported as 17.99 Mb. But both Table 1 of the main text and Suppl Table S29 indicate PPA_v0 has a contig N50 of 16.58 Mb. The information in the NCBI accession matches this latter statistic. It is not clear why the longer N50 of 17.99 Mb is presented on line 130 and in Suppl Table S3.

The contig N50 17.99 Mbp value was reported based on the initial assembly, quiver, pilon, and indel correction. The 16.58 Mbp contig N50, in contrast, was the result of post-processing (i.e., after application of Bionano Genomics optical mapping to scaffold and trim contigs that were misassembled or of insufficient quality). Since the latter numbers are more relevant, we now

refer to the post-processing contig N50 numbers throughout the main text. We have also revised accordingly Table S3 and changed the text referencing these in the Supplementary Tables.

We replaced the original supplementary tables with these updates:

Supplementary Note Table S3. Bonobo assembly statistics

Number of contigs	4,873
Total size of contigs (bp)	3,015,459,349
Longest contig	91,355,120
Shortest contig	125
Bases in Contigs > 1 kbp	3,015,433,760 (100.00%)
Bases in Contigs > 10 kbp	3,012,108,918 (99.90%)
Bases in Contigs > 100 kbp	2,886,730,091 (95.70%)
Bases in Contigs > 1 Mbp	2,686,481,600 (89.10%)
Bases in Contigs > 10 Mbp	1,969,377,575 (65.30%)
N50 contig length	17,987,413
L50 contig count	46
contig %A	29.7
contig %C	20.29
contig %G	20.28
contig %T	29.73
contig %N	0
contig %non-ACGTN	0

Falcon assembly stats for Mhudiblu_PPA_v0; statistics computed after error correction (post Quiver, Pilon and indel correction)

Supplementary Note Table S29. Final assembly statistics comparing Mhudiblu_PPA_v0, Mhudiblu_PPA_v1 and Mhudiblu_PPA_v2

	Mhudiblu_PPA_v0	Mhudiblu_PPA_v1 before adding contigs from Segmental Duplication Assembler (SDA)	SDA	Mhudiblu_PPA_v1	Mhudiblu_PPA_v2
Total scaffolds	4357	4379	1145	5524	5520
Ordered/oriented scaffolds	88	137	0	137	133
Scaffolds on chr*_random	0	108	0	108	108
Scaffolds on chrUn	4269	4134	1145	5279	5279
Contigs	4976	4977	1145	6122	6118
Ordered/oriented contigs	641	697	0	697	693
Contigs on chr*_random	0	125	0	125	125
Contigs on ChrUn	4334	4155	1145	5300	5300
non-N bases (contigs)	3,015,350,297	3,015,333,734	55,883,605	3,071,217,339	3,073,752,221
Scaffold bases (including Ns)	3,051,901,337	3,049,120,773	55,883,605	3,105,004,378	3,107,539,260
non-N bases on chromosomes	2,756,975,881	2,790,338,069	0	2,790,338,069	2,793,604,526
bases on chromosomes (including Ns)	2,787,676,126	2,918,899,387	0	2,918,899,387	2,920,672,989
bases on chr*_random (not including Ns)	0	12,455,377	0	12,455,377	12,482,156
Contig N50	16,579,680	16,579,680		16,070,023	16,076,652
Contig L50 count	48	49		50	50
Scaffold N50	68,246,502	55,818,576		53,354,638	53,386,619
Scaffold L50 count	16	18		19	19

We also changed the text in the Supplementary Note to reflect these changes: i.e., "assembly contains 3.015 Gbp distributed amongst 4,975 contigs with an N50 of 16.580

Mbp (**Supplementary Note Table S3**). There were 1,088 contigs greater than 100 kbp..”

We added a reference to Kronenberg et al. and made a note at the bottom of the Supplementary Note Table S3 to state “post Quiver, Pilon and indel correction”.

2) Given that PPA_v1 incorporates a number of useful corrections to PPA_v0, it is not clear why PPA_v0 is the assembly of record in NCBI. Can the authors demonstrate that PPA_v1 is forthcoming in NCBI? Can they include the accession number in this manuscript?

Both v0 and v1 accessions have been released (v1 GenBank accession is now SSBP000000000 and the assembly accession is GCA_013052645.2). The v1 assembly differs mainly as a result of correction of inversions and large-scale orientation issues. It is thus a more contiguous assembly.

3) Lines 146-147: It would be valuable to state the pairwise nucleotide divergence separating the new bonobo assembly from the latest chimpanzee assembly and from GRCh38/hg38.

Using minimap2, we aligned the chimpanzee (Clint_PTRv2), human (GRCh38) and bonobo (Mhudiblu_PPA_v0) genomes in 1 Mbp windows and computed pairwise nucleotide divergence for autosomes separately from the X chromosome considering SNVs as well as SNVs+INDEL differences combined (Supplementary Note Fig. S57). The primary statistics including the mean are highly consistent (see below). We investigated outliers (regions of excess divergence as suggested by the bimodal peak) on the X chromosome in smaller 100 kbp bins and find that they correspond primarily to regions of duplications and inversions where optimal pairwise alignments are more difficult to construct (Supplementary Note Fig. S58). The overall nucleotide divergence between chimpanzee and bonobo based on the latest genome assemblies is 0.421 ± 0.086 for autosomes and $0.311 \pm 0.060\%$ for the X chromosome (Supplementary Note Table S55).

We modified the main to now read:

“We estimate the overall sequence accuracy of the bonobo assembly to be 99.97-99.99% using a variety of metrics and polishing steps (Supplementary Note). The overall nucleotide divergence between chimpanzee and bonobo based on these new long-read assemblies is $0.421 \pm 0.086\%$ for autosomes and $0.311 \pm 0.060\%$ for the X chromosome (Supplementary Note Table S55).”

In addition, we added these details and figures to the Supplementary Note:

Supplementary Note Figure S57. Bonobo, chimpanzee and human nucleotide divergence. Panels show genome-wide SNV (top) and SNV +INDEL (bottom) divergence based on comparisons between the chimpanzee (Clint_PTRv2), bonobo (Mhudiblu_PPA_v0) and human genomes (GRCh38). The divergence was calculated in 1-Mbp non-overlapping windows across all autosomes and chromosome X (excluding X and Y homologous regions, analyzed region: chrX:93120350-155700620).

Supplementary Note Table S55. The genomic divergence among chimpanzee and bonobo and human

	SNV only divergence (%)		SNV+INDEL divergence	
	autosome	chrX	autosome	chrX
Bonobo_chimp	0.421±0.086	0.311±0.060	0.498±0.095	0.376±0.064
Human_bonobo	1.298±0.187	0.952±0.121	1.454±0.198	1.074±0.129
Human_chimp	1.297±0.192	0.960±0.141	1.453±0.203	1.083±0.150

Supplementary Note Figure S58. Divergence outliers on the X chromosome. Chimpanzee (orange, Clint_PTRv2) and bonobo (blue dashed lines, Mhudiblu_PPA_v0) divergence compared to human (GRCh38) X chromosome. The divergence was calculated based on analysis of non-overlapping 100 kbp windows across the X chromosome (excluding X and Y homologous regions). Regions of excess divergence frequently correspond to annotated segmental duplications (SDs, blue) or inverted (INV, green) segments in the chimpanzee genomes.

4) Lines 187-192: The authors should report whether there are GO categories enriched among the genes that show either gene family expansion or contraction in bonobos relative to humans or chimpanzees and especially gene family changes unique to bonobos.

To address this issue, we investigated gene ontology enrichment for the gene families that show evidence of expansion and contraction (see section 6.4 in the Supplementary Note) using the following databases: KEGG_2019_Human, GO_Molecular_Function_2018, GO_Biological_Process_2018, GO_Cellular_Component_2018 and Panther_2016.

Interestingly, among gene family contractions, all comparisons (bonobo vs. humans, bonobo vs. chimpanzee, bonobo vs. (chimpanzee, human)) showed a significant enrichment (after BH correction) for the pathway '*Maturity onset diabetes of the young*'. For gene family expansions, we observe no significant enrichment for bonobo-specific differences. We observed signals for methylation-dependent chromatin silencing and progesterone when comparing bonobo expansion versus human and immunity differences when comparing bonobo gene family expansion versus chimpanzee (Supplementary Note Table S47). The genes underlying the latter, however, correspond to immunoglobulin genes and are often difficult to entangle from somatic variation (VDJ recombination) as opposed to strictly germline differences. Moreover, bonobo-human differences are driven by clustered gene families (i.e., likely single events or a series of mutational events driven by recombination), and thus, these differences are less likely to be functionally informative.

We call out the contraction association with maturity onset of diabetes of the young in the main text and refer the reader to the Supplementary Note for a more detailed analysis:

“Similarly, we validated bonobo-specific gene-family contractions (*IGFL1*, *TRAV4K*, *CDK11A*) and more ancient duplications common to both chimpanzee and bonobo (e.g., *CLN3*, *EIF3C*, *RGL4*). These bonobo-contracted gene families show some GO enrichment for genes related to maturity onset diabetes of the young (Supplementary Note Table S47).”

Supplementary Note Table S47. GO enrichment analysis of gene family contractions and expansion in bonobo compared to human and chimpanzee

Term	Overlap	P-value	Adjusted P-value	Genes	Gene_set	Type	Species compared
Maturity onset diabetes of the young	8/26	9.69E-05	0.03	HHEX,BHLHA15,MAFA,MNX1,INS,NKX2-2,NEUROG3,FOXA2	KEGG_2019_Human	Contraction	chimp and human
methylation-dependent chromatin silencing (GO:006346)	4/11	4.43E-06	0.02	MBD3L4,MBD3L5,MBD3L2,MBD3L3	GO_Biological_Process_2018	Expansion	human
Progesterone-mediated oocyte maturation	7/99	1.11E-04	0.03	SPDYE2B,SPDYE2,SPDYE1,SPDYE16,SPDYE3,SPDYE6,SPDYE5	KEGG_2019_Human	Expansion	human
Fc receptor mediated stimulatory signaling pathway (GO:0002431)	5/135	8.02E-06	0.0037	IGLV6-57,IGLV3-21,IGLV1-44,IGLV7-43,IGLV3-19	GO_Biological_Process_2018	Expansion	chimp
regulation of protein processing (GO:0070613)	5/128	6.18E-06	0.0039	IGLV6-57,IGLV3-21,IGLV1-44,IGLV7-43,IGLV3-19	GO_Biological_Process_2018	Expansion	chimp
Fc-gamma receptor signaling pathway (GO:0038094)	5/134	7.73E-06	0.0039	IGLV6-57,IGLV3-21,IGLV1-44,IGLV7-43,IGLV3-19	GO_Biological_Process_2018	Expansion	chimp
(GO:0002455)	5/125	5.51E-06	0.004	IGLV6-57,IGLV3-21,IGLV1-44,IGLV7-43,IGLV3-19	GO_Biological_Process_2018	Expansion	chimp
(GO:0038096)	5/133	7.46E-06	0.0042	IGLV6-57,IGLV3-21,IGLV1-44,IGLV7-43,IGLV3-19	GO_Biological_Process_2018	Expansion	chimp
complement activation, classical pathway (GO:006958)	5/123	5.09E+06	0.0043	IGLV6-57,IGLV3-21,IGLV1-44,IGLV7-43,IGLV3-19	GO_Biological_Process_2018	Expansion	chimp
regulation of immune effector process (GO:0002697)	5/114	3.50E-06	0.0045	IGLV6-57,IGLV3-21,IGLV1-44,IGLV7-43,IGLV3-19	GO_Biological_Process_2018	Expansion	chimp
regulation of acute inflammatory response (GO:0002673)	5/121	4.70E-06	0.0048	IGLV6-57,IGLV3-21,IGLV1-44,IGLV7-43,IGLV3-19	GO_Biological_Process_2018	Expansion	chimp
regulation of humoral immune response (GO:0002920)	5/113	3.36E-06	0.0058	IGLV6-57,IGLV3-21,IGLV1-44,IGLV7-43,IGLV3-19	GO_Biological_Process_2018	Expansion	chimp
regulation of complement activation (GO:0030449)	5/109	2.81E-06	0.0072	IGLV6-57,IGLV3-21,IGLV1-44,IGLV7-43,IGLV3-19	GO_Biological_Process_2018	Expansion	chimp
Fc receptor signaling pathway (GO:0038093)	5/183	3.48E-05	0.0137	IGLV6-57,IGLV3-21,IGLV1-44,IGLV7-43,IGLV3-19	GO_Biological_Process_2018	Expansion	chimp
regulation of protein activation cascade (GO:2000257)	5/108	2.68E-06	0.0137	IGLV6-57,IGLV3-21,IGLV1-44,IGLV7-43,IGLV3-19	GO_Biological_Process_2018	Expansion	chimp
Fc-epsilon receptor signaling pathway (GO:0038095)	5/182	3.40E-05	0.0144	IGLV6-57,IGLV3-21,IGLV1-44,IGLV7-43,IGLV3-19	GO_Biological_Process_2018	Expansion	chimp
receptor-mediated endocytosis (GO:0006938)	5/188	3.96E-05	0.0144	IGLV6-57,IGLV3-21,IGLV1-44,IGLV7-43,IGLV3-19	GO_Biological_Process_2018	Expansion	chimp
serine-type peptidase activity (GO:0008236)	5/220	8.35E-05	0.0481	IGLV6-57,IGLV3-21,IGLV1-44,IGLV7-43,IGLV3-19	GO_Molecular_Function_2018	Expansion	chimp

Term: Gene classes enriched; p-value: p-value based on Fisher's test; Overlap: number of genes in the tested set overlapping with the gene category; Adjusted p-value: Benjamini-Hochberg adjusted p-value; Genes: Name of the genes in the overlap; Gene set: Gene ontology class; Type: specifies if the gene set tested is an expansion or a contraction; Species compared: Indicates if the expansion/contraction in bonobo is related to human or chimpanzee

5) Structural variation. The analysis of SVs is extensive and important. I recommend that the authors look at whether there is significant overlap between genomic regions that contain bonobo-specific SVs and the regions identified as introgressed into bonobos from an ancient extinct great ape (Kuhlwilm et al. 2019, PMID31036897). Similarly, it would be useful to compare the inferred introgressed regions against the regions that show bonobo-specific changes in gene number (protein coding gene family expansions or contractions).

We intersected all archaic regions (1,579 segments, 72.67 Mbp) identified by Kuhlilm and colleagues (see Table S7 in Kuhlilm 2019, PMID: 31036897) with fixed SVs and bonobo-specific gene expansions/contractions. We identified 52 fixed deletions (48.2 kbp) and 103 fixed insertions (98.2 kbp) overlapping archaic regions of introgression—none of which disrupted coding sequencing (Supplementary Note Table S56). Based on human ENCODE v3 annotation (Snyder et al; 2020, PMID: 32728248), we find five fixed insertions and eight fixed deletions overlapping introgressed regions and potential regulatory DNA (Supplementary Note Table S56).

To test for potential enrichment or depletion, we performed a simulation as follows: We binned the bonobo genome into 46 kbp windows (excluding regions where SVs could not be called such as centromeres) and randomly selected 1,579 windows (46 kbp*1579=72.6 Mbp). We computed the number of intersected fixed insertions and deletions as well as the number of the intersected expanded and contracted genes, constructing a distribution of observed events based on 1000 simulations (Supplementary Note Figure S59). We find no evidence of an enrichment of fixed insertions (p-value=0.168) or fixed deletion (p=0.479) among archaic introgressed segments. While we find no bonobo-specific expansions within archaic introgressed regions consistent with expectations (p=0.38), we do identify five-specific contractions (*AL513128.2*, *ACD*, *SMIM32*, *LEFTY2*, and *PTF1A*) representing a significant

depletion ($\rho=0$).

Since other researchers may find this analysis useful, we include it in the revised Supplementary Note.

Supplementary Note Figure S59. Introgressed versus SV regions in bonobo. We compared previously identified introgressed regions in bonobo (1,579 segments, 72.67 Mbp) identified by Kuhlwilm and colleagues (see Table S7 in Kuhlwilm 2019, PMID: 31036897) with regions of structural variation in the bonobo genome. We considered four bonobo categories: **a**, fixed deletions, **b**, fixed insertions, **c**, gene family expansions, and **d**, gene family contractions and identified 155 overlaps (Supplementary Table 15). We then performed simulations to assess the significance of overlap. No category showed significance other than gene family contractions which were significantly depleted in inferred archaic introgressed regions.

Supplementary Note Table S56. The intersection of archaic regions and the fixed bonobo SVs and bonobo-specific gene expansions/contractions

Hg38_CHR	START	END	SV ID	SV type	SV len	Introgresed_CHR	START	END	Annotation	genes	ENCODE_CHR	START	END	EH38D	EH38E	CCRE2020
chr12	79619095	79619096	chr12-79619095-INS-3814	INS	3814	chr12	79590000	79630000	intron_variant	PAWR	chr12	79619040	79619383	EH38D2581658	EH38E1627386	dELS
chr13	98508970	98508971	chr13-98508970-INS-1671	INS	1671	chr13	98490000	98530000	intron_variant	STK24	chr13	98508693	98509039	EH38D2683120	EH38E1691700	dELS
chr14	63563689	63563690	chr14-63563689-INS-329	INS	329	chr14	63540000	63580000	upstream_gene_variant	AL136038.2	chr14	63563648	63563997	EH38D2727834	EH38E1720568	dELS
chr21	22711061	22711062	chr21-22711061-INS-68	INS	68	chr21	22680000	22720000	intergenic_variant	NA	chr21	22710738	22711067	EH38D3328551	EH38E2133253	dELS
chr7	130894987	130894988	chr7-130894987-INS-60	INS	60	chr7	130860000	130900000	intron_variant&non_coding_transcript_variant	AC016831.1	chr7	130894941	130895285	EH38D4031127	EH38E2590655	dELS,CTCF-bound
chr1	235613896	235614669	chr1-235613896-DEL-774	DEL	774	chr1	235590000	235630000	intron_variant	GNG4	chr1	235613591	235613913	EH38D2293865	EH38E1434404	pELS
chr1	235613896	235614669	chr1-235613896-DEL-774	DEL	774	chr1	235590000	235630000	intron_variant	GNG4	chr1	235614462	235614761	EH38D2293866	EH38E1434405	pELS,CTCF-bound
chr18	5796713	5796888	chr18-5796713-DEL-176	DEL	175	chr18	5790000	5830000	intron_variant&non_coding_transcript_variant	MIR3976HG	chr18	5796835	5797176	EH38D2977591	EH38E1897042	dELS
chr19	31119429	31119616	chr19-31119429-DEL-188	DEL	188	chr19	31080000	31120000	intron_variant&non_coding_transcript_variant	AC020912.1	chr19	31119578	31119735	EH38D3054513	EH38E1948538	dELS
chr3	58537527	58541961	chr3-58537527-DEL-4435	DEL	4435	chr3	58530000	58600000	upstream_gene_variant	ACOX2	chr3	58537452	58537626	EH38D3433780	EH38E2206425	pELS
chr3	58537527	58541961	chr3-58537527-DEL-4435	DEL	4435	chr3	58530000	58600000	upstream_gene_variant	ACOX2	chr3	58537777	58538124	EH38D3433781	EH38E2206426	pELS
chr3	58537527	58541961	chr3-58537527-DEL-4435	DEL	4435	chr3	58530000	58600000	upstream_gene_variant	ACOX2	chr3	58539103	58539364	EH38D3433782	EH38E2206427	DNase-H3K4me3
chr6	53821045	53822473	chr6-53821045-DEL-1429	DEL	1429	chr6	53790000	53860000	intron_variant	LRR1	chr6	53821085	53821348	EH38D3851951	EH38E2474132	DNase-H3K4me3
chr6	53821045	53822473	chr6-53821045-DEL-1429	DEL	1429	chr6	53790000	53860000	intron_variant	LRR1	chr6	53822462	53822765	EH38D3851953	EH38E2474133	dELS,CTCF-bound
chr8	41587044	41587122	chr8-41587044-DEL-79	DEL	79	chr8	41550000	41590000	intron_variant	GPAT4	chr8	41586800	41587136	EH38D4086504	EH38E2627263	dELS
chr9	104868788	104868840	chr9-104868788-DEL-53	DEL	53	chr9	104850000	104890000	intron_variant	ABCA1	chr9	104868678	104868989	EH38D4221244	EH38E2713984	dELS
chr9	26131399	26133462	chr9-26131399-DEL-2064	DEL	2064	chr9	26130000	26170000	intergenic_variant	NA	chr9	26132733	26133022	EH38D4181843	EH38E2688252	CTCF-only,CTCF-bound

6) Line 478: Does the level of ILS observed have implications for inferring the effective population size of the LCA of bonobos and chimpanzees? I would not suggest that a full analysis of demographic history should be included in this report, but the authors might comment on the potential inference that N_e was large in the LCA, given that polymorphisms were retained from the time of divergence from humans about 7 mya to the divergence of bonobos from chimpanzees about 1.7 mya.

The relatively high proportion of ILS within the Pan genus suggests that the population predating their species divergence was relatively large, with most reductions in population size occurring more recently. To test this, we applied the pairwise sequential Markovian coalescent (PSMC) method using Illumina WGS data from bonobos and chimpanzees (Supplementary Note Table S35) mapped back to the new reference genomes and inferred changes in effective population as well as timing of population expansions (Supplementary Note Figs. S28 and S29). We considered the population split of human and chimpanzee between 4-7 million years ago (mya) and 1-2.5 mya for the split of the chimpanzee and bonobo lineages. Using a 25-year generation time and a mutation rate $\mu = 0.5 \times 10^{-9}$ mut (bp x year), we estimate a large population size for the ancestral bonobo/chimpanzee lineage ($N_e \approx 20,000$). Similarly, we estimate that Pan-Homo ancestral population size is greater than 50,000. These estimates are similar to those performed on the earlier draft versions of the bonobo and chimpanzee genomes (as reported in Prufer et al 2011 PMID: 22722832, and Prado Martinez et al. 2013 PMID: 23823723). However, it is important to note that, if the mutation rate used by Prado-Martinez et al. 2013 PMID: 23823723 and Prufer et al. 2011 PMID: 22722832 is considered ($\mu = 1 \times 10^{-9}$ mut (bp x year)) our estimates for the bonobo/chimpanzee population size are lower than those reported (23,000-37,000 and $27,000 \pm 400$, respectively), as shown in Supplementary Note Table S36. This discrepancy is likely due to the different methodologies employed, CoalHMM and CoalLS. We generated PSMC plots for comparison to the earlier work.

Since N_e estimation is not a major goal of the study, we added these analyses to the main and Supplementary Note and include a comment at the conclusion of the manuscript referring authors to it.

“We predict that a significantly greater fraction (~5.1%) of the human genome is closer to chimpanzee/bonobo when compared to previous studies (3.1%)^{1,10}. We estimate that >36.5% of the hominid genome shows ILS if we consider a deeper phylogeny including gorilla and orangutan due, in part, to the large effective population size of the common ancestor of hominids (Supplementary Note).”

Supplementary Note Table S35. Illumina WGS datasets from ape populations.

Lineage	NAME	SRA Accession	BioProject Accession
bonobo	Bono	SRS396219	PRJNA189439
	Catherine	SRS396206	
	Desmond	SRS396205	
	Dzeeta	SRS396202	
	Hermien	SRS396203	
	Hortense	SRS395319	
	Kombote	SRS396207	
	Kosana	SRS396201	
	Kumbuka	SRS396603	
	Natalie	SRS396238	
chimpanzee	Linda	ERS1286216	PRJEB15086
	Negrita	ERS1286220	
	Frederike	ERS1286243	
	Alice	ERS1286218	
	Blanquita	ERS1286221	
	Coco	ERS1286245	
	Tibe	ERS1286222	
	Ikuru	ERS1286238	
	Cleo	ERS1286240	
Bihati	ERS1286241		
gorilla	Amani	SRS396847	PRJNA189439
	Banjo	SRS396826	
	Delphi	SRS396829	
	Dian	SRS396828	
	Kaisi	SRS396605	
	Kolo	SRS396831	
	Mimi	SRS396827	

*Coverage greater than 20

a

b

c

Supplementary Note Figure S28. PSMC analysis. PSMC plots based on an analysis of Illumina WGS genomes (a) 10 bonobos, (b) 10 chimpanzees, and (c) 7 gorillas. The y-axis represents the N_e ($\times 10^4$) inferred by the PSMC and the x-axis represents the time in years. The N_e and time are scaled with generation time $g=25$ years and the mutation rate per generation $\mu=0.5 \times 10^{-9}$ mut (bp \times year) PMID: 27789843.

Supplementary Note Figure S29. Estimates of effective populations size (Ne) in demes predating divergence in Homo and Pan. Values in boxes refer to Ne ($\times 10^4$) inferred through PSMC analysis considering bonobo (red boxes) and chimpanzee (purple). We extracted size estimates from time intervals between 4-7 mya for the (human,pan) Ne and 1-2.5 mya for (*P.paniscus*,*P.troglodytes*), considering $\mu = 0.5 \times 10^{-9}$ mut (bp \times year) and generation time of 25 years. Values using $\mu = 1 \times 10^{-9}$ mut (bp \times year) are reported in **Supplementary Note Table S36**.

Supplementary Note Table S36. Estimates of effective population size (Ne $\times 10^4$) using PSMC for key temporal intervals

	$\mu = 0.5 \times 10^{-9}$ mut (bp \times year)			$\mu = 1 \times 10^{-9}$ mut (bp \times year)		
	t0	t1 (1Mya < t < 2.5Mya)	t2 (4Mya < t < 7Mya)	t0	t1 (1Mya < t < 2.5Mya)	t2 (4Mya < t < 7Mya)
Chimpanzee	1.15 (0.32-1.85)	1.97 (1.42-4.65)	7.43 (4.46-9.96)	0.57 (0.16-0.93)	1.10 (0.71-3.58)	4.8 (4.22-5.22)
Bonobo	0.22 (0.1-0.52)	2.22 (0.99-2.76)	9.50 (6.08-13.04)	0.11 (0.05-0.26)	1.66 (1.32-5.16)	5.08 (4.96-5.43)

*t=0 is the final Ne, t1 is the time predating the chimpanzee/bonobo divergence, t2 is the time interval predating the pan/homo divergence. We use a generation length of 25 years. μ =mutation rate

7) One of the most interesting aspects of evolution of the genus Pan is the behavioral differences between chimpanzees and bonobos. As described in Staes et al (2019) which is reference #7 in the present Catacchio et al. manuscript, there are specific types of social behavior and specific neural circuits that differ in bonobos vs chimpanzees. While a complete analysis is probably beyond the scope of this report, can the authors provide any novel

information about newly identified genomic differences between these two species that may be related to the neuroanatomical or behavioral differences described by Staes et al.? In particular,

the authors here focus on a unique deletion in the coding sequence of ADAR1 in bonobos, not shared with chimpanzees. The present authors also state that this gene may have some relationship with neurotransmitter function, and I would note that bonobos seem to have a different level serotonergic innervation in relevant neural circuits as compared with chimpanzees (Stimpson et al. 2016, PMID26475872).

This is an interesting suggestion. We investigated the 100 genes associated with neurobiology and social cognition suggested by Staes and colleagues (31422793) and intersected them with fixed SVs and regions where there was evidence of incomplete lineage sorting (ILS). We identified 24 fixed deletions and 26 fixed insertions mapping near these genes (15 and 18 genes, respectively) although we note that all 50 SVs mapped to introns and none intersected any predicted coding sequence. Similarly, we identified 79 genes with a nearby signal of ILS but again all were intronic. Next, we performed a simulation (100 replicates) selecting 100 RefSeq genes at random and computed the number of genes overlapping SVs and regions of ILS. The analysis initially suggested that Staes gene set was highly enriched for both SVs and ILS; however, we also noted that the genes were significantly larger than a random set of genes (typical for genes associated with neurodevelopment). Once we controlled for gene size, we find that neither the number of fixed deletions ($p=0.07$) nor insertions ($p=0.65$) are significantly enriched. Interestingly, the number of ILS segments is lower than expected for these 100 genes ($p=0.03$) perhaps reflecting the action of selection (Supplementary Note Fig. S60).

We summarize these findings in the Supplementary Note and make a note of its possible relationship to differences in serotonin innervation as pointed out by the referee.

Additionally, we integrated the reviewer's comment in the Discussion section as follows:

“While the effect of the deletion on transport, DNA binding, or RNA editing ability awaits experimentation, it is intriguing that previous comparative studies have suggested positive selection of *ADAR1* in bonobo when compared to other mammalian lineages⁵⁹. **Such a change may be related to the different levels of serotonergic innervation observed in bonobo versus chimpanzee neural circuits**⁶⁰. The gene itself has been implicated in a variety of biological activities ranging from recoding neurotransmitter function to suppression of innate immunity⁶¹.”

Supplementary Note Figure S60. Neurobehavioral genes, ILS and SV. Staes and colleagues identified 100 candidate genes that might account for neurobehavioral differences between bonobo and chimpanzee. We intersected the 100 candidate genes with our fixed SVs and 500 bp ILS regions and identified 15 genes near 26 fixed deletions, 18 genes near 26 fixed insertions, and 33 genic regions overlapping the 500 bp ILS windows, but none of the events intersected an exon. We performed a simulation intersecting 100 genes matched for gene length from RefSeq. We find that neither the number of fixed deletions ($p=0.07$) nor insertions ($p=0.65$) are significantly enriched. Notably, the number of ILS segments is lower than expected for these 100 genes ($p=0.03$), perhaps reflecting the action of selection.

Minor comments

a) In my opinion, the pale yellow lines representing deletions in Fig 1 are too pale, difficult to see.

We revised Figure 1 (see below) changing the color for deletion density to red to improve visibility and to better distinguish from PPA-specific inversions.

b) The track hub from line 137 was not accessible from my browser. I can see only a text file with non-useful labels and an email address.

The track hub is accessible at:

https://eichlerlab.gs.washington.edu/public/track_hubs/bonobo_chromosomes/hub.txt

We believe it was not accessible due to a filesystem latency issue that was discovered on the host machine. This issue has been fixed and the track hub should work as expected now. We suggest to copy and paste this address on the MytrackHubs on UCSC Genome Browser to display all relevant information. We apologize for the original issue.

c) Line 179: It is not clear to me why the average increase of complete isoforms is reported as a range (1.5% to 2.1%). Why is the average not a single value? Are these the two values for NCBI and CAT annotations respectively?

Yes that is correct, the two annotation sets gave slightly different results so we revised the sentence as follows:

“We find that 38.4% of protein-coding isoforms are more complete when mapped to the new assembly (**average increase of 1.5 to 2.1% for NCBI and CAT annotations respectively**).”

d) It is not clear to this reviewer what the differences are between Suppl Tables S10, S11 and S12 in the Excel file (copy number differences bonobo vs other species). More explanation of the differences seems warranted.

We include a description of each of the tables to better explain the differences: In details, Table S10 shows the contracted and expanded genes in bonobo compared to human, while Table S11 shows the contracted and expanded genes in bonobo compared to chimpanzee. Table S12 highlights the contracted and expanded genes in bonobo compared to both chimpanzees and humans. Thus, Table S12 is based on the intersection of Table S11 and Table S10. Due to the additional analyses the numbering of tables has changed and now these three tables are Tables S13, S14 and S15.

e) Lines 390-391: Should the number in line 390 be 84 instead of 88 (40 + 44)?

Thank you for pointing this out; the referee is correct. We revised the text to indicate 84.

Referee #2 (Remarks to the Author):

Catacchio et al. present a substantially-improved bonobo reference genome sequence, facilitating the analysis of polarized bonobo- and chimpanzee- lineage specific gene content variation and (with the human genome) incomplete lineage sorting events. The potential strength of the paper is not the genome assembly itself; rather, analyses that could be facilitated by the availability of a high-quality reference genome. This is where my review comments are focused. Otherwise, the descriptive results are important to establish and generally quite solid in this paper, but they are largely accretive relative to those from the prior bonobo genome assembly and our general understandings of genome variation at this point. Overall I was left wanting for more analytical depth and novel impact from the study. If the paper is given further consideration at Nature, I would expect deeper analyses of the structural variation, gene content variation, and incomplete lineage sorting data.

We thank the reviewer for their time and insight. We believe these advances are much more significant and that the technical advances, in fact, were key to making new biological insights. To be clear, there are three reasons we believe the work is impactful

- 1) The new genome assembly is orders of magnitude superior to the previous assembly by almost every metric and such high-quality genomes are critical for identifying the genetic differences that make us human. As an example, the original Prüfer assembly PMID: PMID: 22722832 was estimated to carry 560 Mbp of unresolved sequence (annotated as N's) while Mhudiblu assembly now has only 36.5 Mbp of unresolved sequence (N's). Specifically, we have added 255 Mbp of novel sequence to the genome including an additional 212 Mbp not assigned to chromosomes—the latter consists of various heterochromatic sequence, especially located within subterminal heterochromatic gaps but importantly is now represented (see Fig. R2-1 below). As a result of this improvement, 99.5% of the genes are now complete without frameshift, effectively improving the annotation of 38.4% of the genes. This allows the first comprehensive assessment of hominid gene loss. In addition to this carefully annotated bonobo genome 99.5% of the gaps are filled and most genes are complete. We provide numerous ancillary resources full-length cDNA sequencing (Iso-Seq), Strand-seq datasets, optical mapping data and HiFi sequencing data for the first time which facilitate all future hominid genomic comparisons. This advance in quality is especially critical for species where we are focused on the differences (as opposed to questions of conservation).
- 2) As a result of these improvements, we provide a much clearer assessment of the rapidity at which hominid genomes can change especially structurally even when separated by a short ~1.5 million years. Most of the genic differences, changes in rates of retrotransposition, and structural differences are novel and differ dramatically from previous estimations based on draft genomes and Illumina datasets.
- 3) Third, and perhaps most importantly, our comparison of bonobo to chimpanzee and human nearly doubles previous estimates of incomplete lineage sorting (ILS) among chimpanzee, human and bonobo (Prüfer et al., Nature, 2012 PMID: PMID: 22722832). This is because the more complete long-read ape genomes allowed for ~3-fold increase in the portions of the genome that could be aligned and analyzed (Table R2-1). Our analysis identifies a subset of clustered ILS that appear to be rapidly evolving. Based on the referee's suggestion we extended/replicated this observation based on a broader analysis of the ape phylogeny (including gorilla and orangutan) and now show that it holds for other ILS topologies. This finding is novel and, we believe of broader interest not only because it suggests more extensive sharing of genetic variants among apes and humans, but because it provides strong evidence that ILS is strictly not a random

process. Instead, we find a specific subset of clustered ILS between human, chimpanzee, and bonobo enriched in glycoprotein and EGF signaling (many innate immune response genes) that appear to be subject to positive selection.

Figure R2-1. Novel bonobo sequence. The new assembly adds 255 Mbp of novel sequence to 12,964 distinct locations (red vertical lines) in the long-read Mhudiblu bonobo assembly (Mhudiblu_PPA_v0) when compared to the previous bonobo genome assembly (panpan1.1/GCA_000258655.2; Prufer 2012 PMID: 22722832). In addition a large number of shorter contigs (4,271) corresponding to heterochromatic sequences are also shown as “unplaced1” and “unplaced2” (together comprising 259 Mbp of which 212 Mbp was not previously observed by Prufer, PMID: 22722832; ordered from largest to smallest contigs). A marker of subterminal cap (pCht satellite) (teal vertical lines) as well as homologous alignments shows that three-quarters of this sequence corresponds to telomeric heterochromatin.

Table R2-1. ILS comparison panpan1.1 bonobo assembly vs. Mhudiblu v0

	H-B (bp)	H-C (bp)	Total analyzed sequences (bp)
panpan1.1 (Prufer,2012,TableS8.2)	12,942,453	13,756,464	833,383,247
Mhudiblu_v0 (500 bp resolution)	51,669,000	51,098,500	2,425,788,500

*H-B and H-C ILS regions (in terms of bp) identified based on the previous Prufer bonobo and Mhudiblu_v0.

Because of gaps in early draft assemblies, only 833 Mbp (Table R2-1) of the genome could be analyzed for ILS (Prufer et al, 2012 PMID: 22722832) as compared to 2,426 Mbp with more contiguous long-read genome assemblies (Table R2-1). Concomitantly, this ~2.91-fold increase has led to a larger and more accurate estimate of ILS at 500 bp (approximately doubling the % of ILS segments).

To make this clearer, we reworked the text to highlight the non-incremental advances of the paper and importance for studies of hominid evolution. In particular, based on the referee's request for more detailed analyses, we spent the last three months focused on much more extensive analyses of the structural variation, gene content analyses, and population genetic analyses (including a broader survey of ILS) (see below). We believe the revised manuscript will have greater appeal to a broader audience.

1. Gene content variation. I agree that gene loss events are potentially highly informative. Yet at present the new insight from this study is largely restricted to the development of several hypotheses in the discussion section. It may be that there were not any strongly emergent patterns from the data, but at the least a comprehensive investigation across the phylogeny on a lineage-specific (including the ancestral Pan lineage) basis should be considered and reported.

We performed a much more systematic analysis of gene loss events across the ape phylogeny by focusing on gene-intersecting structural variation events (≥ 50 bp in length) based on a comparison of long-read assemblies of chimpanzee, bonobo, gorilla and orangutan. We also considered insertion/deletion events (< 50 bp) because such events are equally disruptive with respect to gene loss if they introduce a frameshift (Fig. R2-2). In order to validate all events (especially indels which are subject to homopolymer errors within long-read data), we generated HiFi sequencing data from the same four ape genomes (chimpanzee (40.1-fold (X) coverage, bonobo (37.9X), gorilla (31.3X) and orangutan (24.7X) (Table R2-2). HiFi data produces long-read data through circular consensus sequencing which is $>99.9\%$ accurate, allowing us to confirm structural variant and indel events discovered from the CLR-based assemblies. Mutational events were then subsequently followed up as fixed or polymorphic based on genotyping against a population of samples for each species ($n=27$ genomes; 7-10 samples per species) where Illumina WGS data were available. We provide an overview of gene loss (Fig. R2-2) and then summarize the SV and indel analyses separately.

Figure R2-2. Overview of lineage-specific fixed SV and gene loss. The schematic summarizes the total number of fixed insertions (INS) and deletions (DEL) (> 50 bp), on each branch of the ape phylogenetic tree. The number of events resulting in genic changes or leading to a frameshift based on

an SV (the first number in brackets) or indel (the second number in brackets) is indicated. The number of events on the human branch are based on a previous analysis and following our criteria in this study (Kronenberg et al., 2018, PMID: 29880660). Fixed versus polymorphic events were determined based on genotyping of 27 ape WGS Illumina samples (Supplementary Note Table S35).

Table R2-2. Summary of Ape HiFi PacBio WGS data

	Species	Coverage	Accession
Chimpanzee	Clint	40.1	SRR12517369- SRR12517374, SRR12517378, SRR12517389- SRR12517390
Bonobo	Mhudiblu	37.9	SRR13443658 SRR13446350
Gorilla	Kamila	31.3	SRR13446351 SRR13446352
Orangutan	Susie	24.7	SRR12517385- SRR12517387

Supplementary Note Table S35. Illumina WGS datasets from ape populations.

Lineage	NAME	SRA Accession	BioProject Accession
bonobo	Bono	SRS396219	PRJNA189439
	Catherine	SRS396206	
	Desmond	SRS396205	
	Dzeeta	SRS396202	
	Hermien	SRS396203	
	Hortense	SRS395319	
	Kombote	SRS396207	
	Kosana	SRS396201	
	Kumbuka	SRS396603	
	Natalie	SRS396238	
chimpanzee	Linda	ERS1286216	PRJEB15086
	Negrita	ERS1286220	
	Frederike	ERS1286243	
	Alice	ERS1286218	
	Blanquita	ERS1286221	
	Coco	ERS1286245	
	Tibe	ERS1286222	
	Ikuru	ERS1286238	
	Cleo	ERS1286240	
	Bihati	ERS1286241	
gorilla	Amani	SRS396847	PRJNA189439
	Banjo	SRS396826	
	Delphi	SRS396829	
	Dian	SRS396828	
	Kaisi	SRS396605	
	Kolo	SRS396831	
	Mimi	SRS396827	

*Coverage greater than 20

1. Lineage-specific SVs and gene disruption analyses. We applied three callers (PBSV, Sniffle, and Smartie-SV) based on a comparison of four genome assemblies (bonobo (Mhudiblu_PPA_v0), chimpanzee (Clint_PTRv2), gorilla (Kamilah_GGO_v0) and human (GRCh38)) to identify SVs and then extracted the bonobo-specific, chimpanzee-specific and pan-specific SVs--i.e. shared between chimpanzee and bonobo. Using Paragraph (PMID: 31856913), we next genotyped all SVs against Illumina WGS data available from 10 bonobos, 10 chimpanzees and 7 gorillas (PMID: 27789843, 23823723). Based on the genotypes, we calculated the F_{st} between populations and considered an event as fixed and lineage-specific if $F_{st} > 0.8$ between populations from different species. The ensembl variant effect predictor (VEP) was applied (PMID: 27268795) to annotate the SVs in order to identify SVs disrupting genes (Supplementary Note Table S50) as well as events affecting potential noncoding regulatory DNA. We validated all gene disruption events by mapping high-fidelity (HiFi) sequence reads generated from the bonobo, chimpanzee, gorilla and two human genomes back to GRCh38. Relatively few gene disruptions mediated by structural variation were discovered in the Pan lineage (eg. keratin-associated gene Supplementary Note Fig. S46) and much more common were structural changes that led to a significant modification of protein structure (eg. mucin or zinc finger genes Supplementary Note Figure S47).

Supplementary Note Figure S46. Loss of keratin-associated gene in chimpanzee and bonobo lineages. a A 25.7 kbp deletion results in the complete loss of hair keratin-associated protein (*KRTAP19-*

6) in bonobo and chimpanzee. b Sequence read-depth genotyping of deletion in human and ape Illumina WGS data (number of samples) confirms a *Pan*-specific loss fixed in both bonobo and chimpanzee.

Supplementary Note Table S50. The fixed ape SVs affecting exons

Lineage-specific	HUMAN-CHR	HUMAN-START	HUMAN-END	SV-TYPE	SIZE	ANNOTATION	GENE	WGAC	WSSD (SDA)	GENE ID	EXON	pLI
bonobo	chr1	154601820	154601966	DEL	147	inframe_deletion	ADAR	0	0	ENSG00000160710	2//15	9.91E-02
bonobo	chr1	248739523	248763827	DEL	24305	stop_lost	LYPD8	0	0	ENSG00000259823	1-7//7	NA
bonobo	chr11	63119193	63119261	DEL	69	inframe_deletion	SLC22A24	0	0	ENSG00000197658	3//10	3.09E-03
bonobo	chr3	195789477	195790190	DEL	714	inframe_deletion	MUC4	0	0	ENSG00000145113	2//25	5.45E-16
bonobo	chr7	93077971	93119434	DEL	41464	transcript_ablation	SAMD9	0	0	ENSG00000205413	1-3//3	5.21E-30
chimp	chr19	22316718	22316719	INS	84	inframe_insertion	ZNF729	84	0	ENSG00000196350	4//4	4.00E-01
chimp	chr9	113425411	113425412	INS	314	stop_gained	C9orf43	0	0	ENSG00000157653	10//14	2.27E-10
pan	chr1	248589569	248604503	DEL	14935	transcript_ablation	OR2T10	0	0	ENSG00000184022	1-2//2	7.10E-04
pan	chr16	3352155	3359732	DEL	7578	transcript_ablation	OR2C1	0	0	ENSG00000168158	1//1	3.46E-05

Coordinates based on human GRCh38 genome

* This is a partial table, please see the full table in Supplementary Note

Supplementary Note Figure S47. A Pan-specific fixed genic insertion. **a**, A 72 bp insertion in the coding sequence of *ZNF280C* in chimpanzee and bonobo based on genomic sequence alignment among bonobo, chimpanzee, gorilla, and human. **b**, A 24 amino acid insertion specific to bonobo and chimpanzee. **c**, Insert occurs at position 561 in the *ZNF280C* protein.

We also considered the potential loss of noncoding regulatory elements by intersecting lineage-specific SVs with ENCODE V3 (Snyder et al; 2020, PMID: 32728248) catalog of functional elements in humans (Supplementary Note Table S46). We assigned regulatory elements to specific genes if they occurred within the body of the gene (UTR and intron) or the elements are located within 5kb downstream/upstream of the genes. We identified 662 disruptions (fixed insertions and deletions) of noncoding regulatory elements in the bonobo lineage and 356 events in the chimpanzee (Supplementary Note Table S46). Gene ontology enrichment analyses were performed using DAVID (Huang et al; 2009, PMID: 19131956) for SVs

associated with lineage-specific gene disruptions or loss of regulatory DNA. For bonobo specific-SVs, we find genes enriched in membrane regions/topological domain: extracellular ($p=2.4E-4$), regulation (eg., phosphate-binding region ($p=7.8E-4$), zinc finger domain ($p=1.5E-2$)), and neuron-related proteins (ANK repeats, ($p=8.1E-3$), synapse ($p=4.4E-3$), dopaminergic synapse ($8.4E-2$)). Bonobo contrasts with chimpanzee-specific SVs, which show an enrichment only in the cadherin pathway ($p=6.10E-03$). Gene loss in the ancestral Pan lineage (shared between chimpanzee and bonobo) shows enrichments in postsynaptic membrane ($p=1.2E-7$), PDZ domain ($p=4.5E-5$), calcium transport ($p=2.E-3$), regulation (phosphate-binding region ($p=3.8E-3$), GTPase activator activity ($p=5.4E-3$) as well as coronary vasculature development ($p=7.9E-2$) and facial nerve structural organization ($p=4E-2$) (Supplementary Note Table S51). Although potentially interesting, it should be noted that the low number of events makes significance of all enrichments relatively modest.

Supplementary Note Table S46. Summary of lineage-specific SVs

	bonobo			chimpanzee			pan			gorilla		
	all (against hg38)	specific	fixed	all (against hg38)	specific	fixed	all (against hg38)	specific	fixed	all (against hg38)	specific	fixed
Insertion	61,078	15,786 (9.76 Mbp)	3,604 (3.3 Mbp)	63,525	17,761 (10.61 Mbp)	1,959 (1.83 Mbp)		18,742 (12.14 Mbp)	6,646 (6.27 Mbp)	72,793	42,009 (29.13 Mbp)	17,858 (15.99 Mbp)
Deletion	59,246	7,082 (6.82 Mbp)	1,965 (2.36 Mbp)	61,182	7,542 (6.89 Mbp)	1,047 (1.11 Mbp)		14,309 (16.10 Mbp)	6,852 (8.98 Mbp)	69,668	28,194 (27.60 Mbp)	12,309 (13.26 Mbp)
Disrupted exon/ UTR SVs			148			57			293			586
Disrupted exons (validate with HiFi reads)			5 (LYPD8 has half deletion in the orangutan)			2			15 (APOL1&MA GEB6 have half deletion in the orangutan)			20 (MTERF4 has half deletion in the orangutan)
Putative encode regulatory sequence			465(del)+ 197(ins)			252(del)+ 104(ins)			1,753(del)+ 404(ins)			2,408(del)+ 1,038(ins)

Supplementary Note Table S51. Gene ontology enrichment analyses for loss of functional elements

	Term	Enrichment Score	P_Value
for the genes which contains bonobo-specific SVs that intersect with ENCODE (n=381*)	Membrane region/ topological domain:Extracellular	3.28	2.40E-04
	nucleotide phosphate-binding region:ATP	2.59	7.80E-04
	Zinc finger, LIM-type	1.72	1.50E-02
	ANK repeat	1.56	8.10E-03
	ECM-receptor interaction	1.45	9.90E-03
	Host cell receptor for virus entry	1.39	2.30E-02
	Proteoglycans in cancer	1.36	1.40E-04
	ErbB signaling pathway	1.27	1.60E-03
	epidermal growth factor receptor signaling pathway	1.26	1.40E-02
	Fatty acid metabolism	1.21	1.90E-02
	Synapse	1.19	4.40E-03
	Dopaminergic synapse	1.11	8.40E-02
	metal ion-binding site:Magnesium	1.09	6.20E-02
positive regulation of endothelial cell migration	1.07	8.60E-02	
for the genes which contains chimp-specific SVs that intersect with ENCODE (n=187)	Cadherin conserved site	1.15	6.10E-03
for the genes which contains pan-specific SVs that intersect with ENCODE (n=1040)	Pleckstrin homology-like domain	6.15	1.20E-09
	postsynaptic membrane	3.32	1.20E-07
	CRAL-TRIO domain	2.47	3.00E-03
	PDZ domain	2.38	4.50E-05
	WW domain	2.37	2.80E-03
	ATPase, dynein-related, AAA domain	1.89	1.20E-03
	Calcium transport	1.71	2.00E-03
	Aminopeptidase	1.70	3.70E-02
	Calmodulin-binding	1.61	6.20E-03
	clathrin-mediated endocytosis	1.60	4.70E-02
	C2 calcium-dependent membrane targeting	1.59	7.80E-03
	coronary vasculature development	1.54	7.90E-02
	nucleotide phosphate-binding region:ATP	1.49	3.80E-03
	GTPase activator activity	1.44	5.40E-03
	domain:BEACH	1.44	3.00E-02
	Ubiquitin system component Cue	1.43	2.90E-02
	CUB domain	1.41	3.60E-02
	Phosphotyrosine interaction domain	1.39	5.90E-02
	regulation of calcium ion transport	1.37	6.80E-02
	SH3 domain	1.31	3.60E-02
	facial nerve structural organization	1.27	4.00E-02
Cyclic nucleotide-binding domain	1.24	4.70E-02	
phosphatidylinositol binding	1.22	8.60E-02	
AAA+ ATPase domain	1.18	2.00E-02	
Potassium channel, voltage dependent, KCNQ	1.15	4.20E-02	

2. Indel gene frameshift analyses with HiFi read validation

We also investigated potential gene loss as a result of indel mutation events (<50 bp) since such events are functionally equivalent to large structural variation events. We initially identified 323 frameshift mutations for 119 genes in the bonobo assembly based on comparison to human GRCh38. These events were identified from the CAT annotation of the bonobo assembly, and were filtered to include only events on the default isoform (GENCODE's MANE_select isoform) for each gene. We validated all events using HiFi sequencing data from the same source (Mhudiblu). This was done by using the HiFi data to call variants using FreeBayes and check for consistency in variant calls. As a control, we also analyzed HiFi data from two humans (Yoruban

and Puerto Rican samples) and found that only 4 of these variants were also identified as a frameshift in at least one of the two humans. We excluded these from subsequent analysis. In order to define lineage-specificity, we identified frameshift mutations in the chimpanzee and gorilla genomes as described above, and then compared those to the set of bonobo mutations. We identified 423 frameshifts corresponding to 186 genes in gorilla and 328 frameshifts corresponding to 149 genes in chimpanzee (Supplementary Note Fig. S48). We used HiFi sequencing data from an outgroup ape (orangutan) to validate lineage-specificity. Finally, we also used the 27 WGS ape short-reads to genotype these frameshifts by GATK and used the same criteria ($F_{st} \geq 0.8$) to identify the fixed frameshift events in each lineage (Supplementary Note Table S52). Please note that due to the inability to accurately map short-read Illumina data to duplicate genes we limited the analysis to potential indels and frameshifts mapping outside of segmental duplications (Supplementary Note Fig. S48)—i.e. to unique regions of the ape genome. Similar to the structural variant analyses, fixed indel events frequently occurred in genes tolerant to mutation or resulted in modifications to the carboxy terminus, with a few exceptions highlighted below (Supplementary Note Fig. S49).

Supplementary Note Figure S48. Fixed indel mutations resulting in gene frameshifts. **a**, Frameshift mutation events discovered based on CAT annotation of individual ape genomes to human GRCh38. **b**, HiFi-validated frameshift mutations mapping to unique regions of the genome (outside of SDs) and that are fixed in each population based on analysis of Illumina WGS data from 27 ape genomes (Supplementary Note Table S35). Fixed mutations show $F_{st} > 0.8$ for a given lineage. Comparisons between species were made by liftOver to GRCh38. **c**, Venn diagram of fixed lineage-specific and shared gene loss at the level of individual genes based on validated frameshifts in (b).

Supplementary Note Table S52. Fixed frameshifts in the ape lineages with HiFi and WGS validation

Lineage	Genes	Gene ID	Indel type	Human indel coords	PLI
bonobo+chimp+gorilla	WDR78	ENSG00000152763.17	Deletion	chr1:66924747-66924749	1.89E-03
bonobo+chimp+gorilla	OR11L1	ENSG00000197591.3	Deletion	chr1:247840962-247840963; chr1:247840964-247840965	5.79E-02
bonobo+chimp+gorilla	SCIMP	ENSG00000161929.15	Deletion	chr17:5210815-5210817	6.48E-03
bonobo+chimp+gorilla	GNG14	ENSG00000283980.1	Deletion	chr19:12688250-12688252	NA
bonobo+chimp+gorilla	OCSTAMP	ENSG00000149635.3	Deletion	chr20:46541566-46541568	7.13E-04
bonobo+chimp+gorilla	OR2B2	ENSG00000168131.4	Deletion	chr6:27911399-27911400; chr6:27911401-27911402	9.32E-03
bonobo+chimp+gorilla	C12orf60	ENSG00000182993.5	Deletion	chr12:14823553-14823554; chr12:14823555-14823556	4.82E-02
bonobo+chimp+gorilla	ZNF843	ENSG00000176723.10	Deletion	chr16:31436425-31436427; chr16:31436424-31436426	1.35E-03
bonobo+chimp+gorilla	CMTM5	ENSG00000166091.21	Deletion	chr14:23378759-23378761	0.32
bonobo	MTF2	ENSG00000143033.18	Deletion	chr1:93134088-93134089; chr1:93134092-93134093	1.00

*This is a partial table excerpt; full table in Supplementary Note

Supplementary Note Figure. S49. Fixed gene-disrupting indels in the *Pan* lineage. a, 1 bp deletion in *CST9L* leads to a premature stop codon, event fixed in bonobo and chimpanzee. **b**, 1 bp deletion in *RFX8* leads to a premature stop codon, fixed in bonobo and chimpanzee. **c**, 1 bp deletion in *FBXW12* leads to a premature stop codon, fixed in bonobo and chimpanzee.

In response to the request for these additional analyses (now added to the Supplementary Note), we revised the main text as follows to include validations by HiFi and a more extensive analysis of gene loss and regulatory DNA with an emphasis of events on both the bonobo and *Pan* lineage:

“Gene and regulatory DNA disruptions. We focused on a detailed analysis of gene and regulatory DNA loss on the ape lineage based on human gene annotations and SV comparisons in bonobo, chimpanzee, gorilla, and orangutan genomes¹⁵. For example, we identified 381 bonobo-specific and 185 chimpanzee-specific SVs that intersect ENCODE regulatory elements that could be assigned to a gene (Supplementary Note). Bonobo-specific events are enriched in membrane-associated genes with extracellular domains while chimpanzee-specific events are associated with cadherin-related genes (Supplementary Note Table S51). Interestingly, fixed deletions (n=1,040) on the *Pan* lineage (shared between chimpanzee and bonobo) show an enrichment for the loss of putative regulatory elements associated with post-synaptic genes (3.32 enrichment;

p = 1.2 X 10⁻⁷) and pleckstrin homology-like domains (6.15 enrichment; p = 1.20X 10⁻⁹). Disruptions of protein-coding sequence were far less abundant and we extended this analysis to include both SV and indel mutation events (<50 bp) because both can result in a gene loss or gene disruption due to premature truncation. We validated all 110 events by generating high-fidelity genomic sequencing for each of the ape reference genomes and restricting to those events that could be genotyped in a population of genomes (Supplementary Note). As expected, many fixed gene-loss events occurred in genes tolerant to mutation, redundant duplicated genes, or genes where the event simply altered the structure of the protein. For example, we identified and validated a complete

25.7 kbp gene loss of one of the keratin-associated genes (*KRTAP19-16*) associated with hair production in ancestral lineage of chimpanzee and bonobo (Supplementary Note Fig. S46). In the bonobo lineage, we identified five fixed SVs affecting protein-coding genes (Table 2 and Fig. 4 b-d), but only two of which completely ablate the gene when compared to all other apes. *LYPD8*, for example, which encodes a secreted protein that prevents gram-negative bacteria invasion of colonic epithelium, has been totally deleted by a 24.3 kbp bonobo-specific deletion (Fig. 4c). Similarly, *SAMD9* (SAMD FamilyMember

9) has been totally deleted by a 41.46 kbp bonobo-specific deletion (Fig. 4d and Supplementary Note Fig. S38) and fixed only among bonobos. The other three bonobo-specific fixed SV events in protein-coding regions all maintain the ORF, including a 49- amino acid deletion of *ADAR1*, a gene critical for RNA editing and implicated in human disease (Fig. 4b)⁵³⁻⁵⁵”

2. Incomplete lineage sorting (ILS). The statement “5.07% of the human genome has been subject to ILS” is confusing in two different ways. First, the segments with topologies of (human, bonobo), chimpanzee)) and (human, chimpanzee), bonobo)) reflect the long-term maintenance of ancestral polymorphism over the ancestral *Pan* lineage. Second, a much higher proportion of the human genome than this reflects incomplete lineage sorting, with estimates in the 20-30% range for the sum of (human, gorilla), chimpanzee)) and (chimpanzee, gorilla), human)).

Our original intention was to focus only on ILS within the terminal *Pan*/Homo lineage. The referee, however, is correct: the availability of high quality long-read genomes across the ape phylogeny now makes it possible to extend ILS analysis over the last 15 mya of evolution for all branches of the ape tree. As suggested, we repeated our analysis at a resolution of 500 bp including both orangutan (*Susie_PABv2*) and gorilla (*Kamilah_GGO_v0*) genomes. Considering only those tree topologies where there is at least 50% bootstrap support ($\geq 50\%$), we estimate that >36.5% (Supplementary Note Table S34, Supplementary Note Fig. S27) of the genome

shows evidence of ILS with 31.92% belonging to two deeper ILS topologies (orangutan,(((bonobo,chimpanzee),gorilla),human)) and (orangutan,((bonobo,chimpanzee),(gorilla,human))). These estimates are consistent with earlier estimates of 30% (Sally, Aylwyn, et al. 2012, PMID: 22398555) and ~36% (Kronenberg, et al.

2018, PMID: 29880660). Of note, if we eliminate the requirement of bootstrap support (as was done previously), the estimate of ILS, increases to 50.26%. We revised the statement in the main text to avoid this confusion and to clarify the 5.1%.

Supplementary Note Table S34. Distribution of ILS segments (500 bp) using orangutan genome (Susie_PABv2) as a root

	Tree_topology	Number of tree (BS>=50*)	Proportion (BS>=50*)	Number of tree (BS>=0)	Proportion (BS>=0)
Species tree	(O,(G,((B,C),H)))	1,581,810	63.52%	2,317,762	50.26%
ILS (discordant tree)	(O,(((B,C),G),H))	407,472	16.36%	844,133	18.30%
	(O,((B,C),(G,H)))	387,309	15.55%	827,903	17.95%
	(O,(((B,H),C),G))	34,723	1.39%	163,175	3.54%
	(O,((B,(C,H)),G))	28,603	1.15%	156,105	3.38%
	(O,(((G,H),C),B))	6,959	0.28%	46,483	1.01%
	(O,((B,(G,H)),C))	6,954	0.28%	45,414	0.98%
	(O,(((B,G),C),H))	6,030	0.24%	20,167	0.44%
	(O,((B,(G,C)),H))	5,837	0.23%	19,823	0.43%
	(O,(((C,H),G),B))	5,701	0.23%	2,608	0.57%
	(O,(((B,H),G),C))	5,522	0.22%	2,539	0.55%
	(O,((B,G),(C,H)))	4,817	0.19%	43,975	0.95%
	(O,((B,H),(C,G)))	4,569	0.18%	40,795	0.88%
	(O,(((B,G),H),C))	2,019	0.08%	17,515	0.38%
	(O,(((C,G),H),B))	1,935	0.08%	17,267	0.37%
Total number of tree/proportion		2,490,260	1	4,611,987	1
The total analyzed genome size (with respect to hg38 (3.1 Gbp))		40.16%	NA	74.39%	NA

* BS≥50 requires greater than 50% bootstrap values in support of the ML tree topology.

Supplementary Note Figure S27. Chromosome view of ILS. The schematic depicts human chromosomes 3, 4, 7 and X (GRCh38) with distribution of six different ILS shown as density plots. A subset of the major topologies are shown above and below the line (as indicated by color and arrow) and examples are shown with and without using orangutan as an outgroup.

Based on this much more extensive analysis across the ape phylogeny, we revised the abstract to clarify:

“We produce a high-resolution map of incomplete lineage sorting (ILS) estimating that ~5.1% of the human genome is closer to chimpanzee/bonobo and >36.5% of the genome shows ILS if we consider a deeper phylogeny including gorilla and orangutan.”

2b. This also seems to represent a major missed opportunity to consider ILS simultaneously across these different scales, which would be more informative for understanding the process and significance; e.g. do they overlap more than expected by chance? If so, does this reflect repeat occurrence in the same region, maintained polymorphism potentially as a function of long-term balancing selection, what functional categories of genes are enriched (genes

encoding proteins involved in immune responses?), potentially more insightful dN/dS analyses,

integration with human, chimpanzee, bonobo, and gorilla population genomic datasets (genetic diversity, signatures of positive selection), etc. The functional enrichment analyses of the (H,B,C) vs. (H,C,B) regions alone are not that informative or interpretable.

Based on this deeper phylogenetic ILS analysis, we revisited the different classes of ILS and tested whether there was evidence of clustered ILS segments as we had originally observed for chimpanzee, human and bonobo. Then, we assessed whether those clustered segments showed evidence of positive selection (as well as balancing selection) and whether the clustered sites themselves overlapped more than expected by chance.

We compared the amount of overlap for H-C and H-B classified regions in the original callset and the reclassified ILS segments after inclusion of orangutan as an outgroup. As expected (Supplementary Note Table S41), almost all of the original ILS segments (90.9% 86,342/94,964) overlapped the superset of ILS topologies when orangutan was included. However, the addition of gorilla and orangutan did lead to a reclassification of specific categories due to the presence of additional topologies. The overlap between H-C/H-B ILS topologies before and after inclusion was highly significant (Chi-square tests $p < 0.0001$) as we would have expected.

Supplementary Note Table S41. The number of ILS in without orangutan and with orangutan datasets

	ILS	H-C	H-C*	H-B	H-B**	NON-ILS	Total
Without orangutan	94,964 (3.89%)	47,832 (1.96%)	47,832 (1.96%)	47,132 (1.93%)	47,132 (1.93%)	2,348,805 (96.11%)	2,443,769
With orangutan	886,657 (36.28 %)	26,182 (1.07%)	44,200* (1.81%)	26,056 (1.07%)	43,936** (1.80%)	2,355,112 (63.72%)	2,443,769
Overlapped	86,342 (90.92%)	25,051 (52.37%)	34,384 (71.88%)	25,168 (53.40%)	34,09 (72.33%)		

Based on an analysis of 3,818,646 segments where tree topology could be assigned.

* the number of ILS contain (O,((B,C,H),G)), (O,(((G,H),C),B)), (O,(((C,H),G),B)), (O,((B,G),(Cp,H))), and (O,(((C,G),H),B))

** the number of ILS contain (O,(((B,H),C),G)), (O,((B,(G,H)),C)), (O,(((B,H),G),C)), (O,((B,H),(C,G))), and (O,(((B,G),H),C))"

Next, we restricted the clustered analysis to high-confidence ILS segments (bootstrap ≥ 50) and first tested whether those inter-ILS distances were non randomly distributed when compared to the null (Supplementary Note Fig. S33). We considered the four most abundant ILS topologies, namely:

- 1) O-H: (orangutan,(((bonobo,chimpanzee),gorilla),human)),
- 2) O-(H,G): (orangutan,((bonobo,chimpanzee),(gorilla,human))),
- 3) H-B: (orangutan,(((bonobo,human),chimpanzee),gorilla)),
- 4) H-C: (orangutan,((bonobo,(chimpanzee,human)),gorilla))).

For each topology, we observe a characteristic cluster of ILS segments that deviate significantly from the null and are not randomly distributed in the genome. We note that the proportion of

clustered ILS segments differs with older topologies (more ancient ILS) showing a greater fraction of clustered sites. For example, for the O-H and O-(H,G) topologies the proportion of clustered sites is ~32-34% while for H-B and H-C this fraction is 8-10%.

Supplementary Note Figure S33. Clustered ILS sites. The distance between adjacent ILS segments (inter-ILS) (500 bp resolution) was calculated and the distribution was compared to a simulated expectation based on a random distribution. The analysis reveals a bimodal (and possibly an emerging trimodal) pattern where a distinct subset of ILS are clustered (i.e., clustered ILS sites). Four different topologies are considered: **a**, (orangutan,(((bonobo,chimpanzee),gorilla),human)) ILS topology where 31.58% of inter-ILS are clustered; **b**, (orangutan,((bonobo,chimpanzee),(gorilla,human))) ILS topology where 33.5% are clustered; **c**, (orangutan,(((bonobo,human),chimpanzee),gorilla)) ILS topology (8.14%); and **d**, (orangutan,((bonobo,(chimpanzee,human)),gorilla)) ILS topology (9.89% of sites).

Next, we investigated whether we still observed the elevated dN/dS in clustered ILS. As before we compared the observed dN/dS values for clustered sites against a simulated set where 1000 genes were chosen at random and a genome-wide distribution was created (Supplementary Note Fig. S34) by repeating the process 100 times to generate a null distribution (mean=0.263). Using a one sample t-test statistic, we observe a significant elevated mean dN/dS in both clustered H-C & H-B ($p < 2.2e-16$, mean=0.366) and in clustered O-H & O-G-H ($p < 2.2e-16$, mean=0.316) when compared to the null. The nonclustered H-C and H-B topologies remain insignificant ($p=0.45$, mean=0.264) although non-clustered O-H & O-G-H sites now show

evidence of excess of amino acid replacement ($p < 2.2e-16$, mean=0.306) although that difference is more subtle and occurs within the last 5% of the null distribution.

Supplementary Note Figure S34. Elevated dN/dS in clustered sites of ILS. The null distribution (gray) is based on calculation of mean dN/dS for 1000 genes drawn randomly from the genome (100 simulations) (mean: 0.263). The blue solid and dashed lines represent the mean dN/dS for clustered H-C & H-B ILS (mean: 0.366, $p < 2.2e-16$) and non-clustered H-C & H-B sites (mean=0.264, $p=0.45$), respectively. The solid and dashed purple lines represent mean dN/dS of the clustered O-H & O-G-H ILS (mean=0.316, $p < 2.2e-16$) and the non-clustered O-H & O-G-H ILS (mean=0.306, $p < 2.2e-16$).

Significance performed using the t test in R although similar results based on the null distribution.

Based on this phylogenetically deeper analysis of ILS, we grouped the four most abundant ILS topologies and repeated the inter-ILS distance clustering analysis. As expected, the clustering signal became stronger suggesting long-term maintenance of ILS over specific regions of the genome as suggested by the referee (Supplementary Note Figure S35 and Figure R2-3). A GO analysis (DAVID, Huang 2009, PMID: 19131956) of the genes intersecting these combined data showed the most significant signals for immunity (eg: Glycoprotein ($p=1.3E-25$), Immunoglobulin-like fold/ FN3 ($p=2.4E-20$)), but also genes related to the transporter function (eg: transmembrane region ($p=1.3E-25$) and specifically calcium transport ($p=3.7E-8$) (Supplementary Note Table S42 and Table R2-3). Among the former, the MHC is an exemplar (positive control) and we depict the depth and diversity of ILS topologies schematically over that region.

Supplementary Note Figure S35. Clustered ILS sites of main four ILS topologies. The distance between four adjacent main ILS segments (inter-ILS) (500 bp resolution) was calculated and the distribution was compared to a simulated expectation based on a random distribution.

Supplementary Note Table S42. GO enrichments treating the four major ILS topologies as one group

Term	Enrichment Score	p_value
Glycoprotein	13.59	1.30E-25
transmembrane region	10.86	1.30E-25
Protein kinase, ATP binding site	8.13	1.50E-26
EGF-like domain	6.42	1.60E-13
Myosin head, motor domain	6.15	8.40E-12
Calcium transport	5.46	3.70E-08
Immunoglobulin-like fold/ FN3	5.28	2.40E-20
Immunoglobulin domain	5.26	3.60E-07
Sodium transport	4.88	3.90E-07
ECM-receptor interaction	4.69	3.60E-07
domain:VWFA	4.68	9.80E-05
Rho guanyl-nucleotide exchange factor activity	4.60	1.10E-07
Tyrosine-protein kinase	4.17	1.60E-06
SH3 domain	4.06	2.50E-06
MAM domain	3.68	8.60E-05
ATPase, dynein-related, AAA domain	3.64	3.60E-06
Voltage-dependent potassium channel, four helix bundle domain	3.52	1.30E-07
C2 calcium-dependent membrane targeting	3.32	4.30E-05
Deafness	3.25	2.00E-05
Complement C1r-like EGF domain	3.02	2.60E-06

Figure R2-3 The ideogram of the MHC region with ILS annotations. a) The four main ILS topologies are color-coded below. The four color lines representing ILS segments are shown above the chromosome coordinate (hg38). The clustered ILS are shown above the four color lines in black. The MHC region is in red color (chr6:28510120-33480577). b) A zoomed-in view of MHC region (chr6:32786501-33103000) showing the clustered ILS nearby *HLA* genes. c) The ideogram of the clustered immunity genes (*EGF*, *RRH* and *LRIT3*) with ILS annotations. The four main ILS topologies are color-coded below. The 4 color lines representing ILS segments are shown above the chromosome coordinate (hg38). The clustered ILS are shown above the 4 color lines in black. ILS exons (H-B in red, H-C in blue) are shown above the CLUSTERED ILS and the *EGF* exon 5 alignment are shown in the top. d) The ideogram of the clustered transporter genes (*SLC22A6*, *SLC22A8*, *SLC22A24*, *SLC22A25*, *SLC22A10*, and *SLC22A9*) with ILS

annotations. The four main ILS topologies are color-coded below. The 4 color lines representing ILS segments are shown above the chromosome coordinate (hg38). The clustered ILS are shown above the 4 color lines in black.

Table R2-3. GO enrichments treating the four major ILS topologies as one group (without MHC regions).

Term	Enrichment	p_value
Glycoprotein	13.59	9.20E-25
transmembrane region	10.3	1.30E-25
domain:EGF-like 4	8.94	1.40E-13
ATP-binding	8.08	2.10E-26
Myosin head, motor domain	6.19	7.00E-12
Calcium transport	5.51	3.10E-08
Immunoglobulin-like fold/ FN3	5.3	1.80E-20
Immunoglobulin domain	5.27	4.60E-11
Sodium transport	4.93	3.30E-07
ECM-receptor interaction	4.75	3.20E-07
domain:VWFA	4.72	2.70E-06
Rho guanyl-nucleotide exchange factor activity	4.65	9.60E-08
SH3 domain	4.13	4.00E-06
MAM domain	3.7	8.20E-05
ATPase, dynein-related, AAA domain	3.66	3.40E-06
Voltage-gated channel	3.56	9.90E-08
C2 calcium-dependent membrane targeting	3.37	3.80E-05
Deafness	3.3	1.70E-05
domain:ABC transporter 1	3.03	4.00E-08
Zymogen	3.02	3.90E-06

Finally, in response to the referee's request to perform deeper population genetic analyses beyond simple dN/dS tests of positive selection, we assessed whether there was any evidence of long-term balancing selection corresponding to regions of ILS based on genetic diversity. Here, we focused specifically on the 25,168 (H,C)B and 25,051 (H,B)C segments identified from our more extended ILS analysis (using orangutan as outgroup as described above). We identified patterns of single-nucleotide variant (SNV) diversity (GATK) genome-wide by mapping WGS data from 10 bonobos and 10 chimpanzees to human GRCh38 (Supplementary Note Table S35). We used these data, to calculate genetic diversity (π) for the bonobo & chimpanzee population and assess stratification using d_{xy} (an absolute measure of genetic divergence between incipient lineages) between bonobo and chimpanzee. We then compared patterns for H-B and H-C ILS segments, a matched randomly chosen subset and genome-wide.

Regions of long-term balancing selection are expected to have unusually high diversity within species and an excess of shared alleles between species. Previous analyses of the trans-species ABO polymorphisms have confirmed such sites through simulation and suggested that sites of balancing selection are typically small (<4 kbp) due to the action of recombination, although this may in fact aggregate in specific regions (Ségurel,2012, PMID: 23091028; Leffler, 2013, PMID: 23413192). We therefore calculated the π and d_{xy} diversity within 500 bp windows comparing clustered and non-clustered H-B/H-C ILS to null set drawn from randomly selected genome segments from the genome (Supplementary Note Fig. S36).

In general, bonobo sites (H,B),C) sites show little difference between the clustered and non-clustered sites or the null expectation--diversity is exceedingly low in all cases consistent with previous population genetic analyses of this species. In contrast, non-clustered sites in chimpanzee show the greatest population genetic diversity and, in the case of (H,B),C) non-clustered ILS regions show greater diversity than clustered regions. As expected both clustered and non-clustered ILS show significantly higher dxy values when compared to the null, although clustered sites showing significantly higher values (Supplementary Note Fig. S36). These findings are consistent with the action of long-term balancing selection resulting in greater polymorphism and higher dxy between two pop/species possibly consistent with long-term maintenance of ancestral polymorphism within the ancestral Pan lineage. Because balancing selection is typically associated with noncoding regulatory DNA (Cheng, 2019, PMID: 30380122; Teixeira, 2015, PMID: 25605789; Leffler, 2013, PMID: 23413192), we believe the observation of elevated dN/dS (positive selection) and balancing selection over the noncoding DNA are not mutually exclusive.

Supplementary Note Figure S36. Tests for balancing selection and ILS. Genetic diversity (π , d_{xy}) is compared for clustered and non-clustered ILS segments and a genome-wide null for **a**, bonobo, **b**, chimpanzee, and **c**, between the species (d_{xy}). The mean with 95% confidence intervals (log scale x-axis); p-value between ILS segments and NULL (to the right of each boxplot, Wilcoxon rank test); p-value between clustered and non-clustered (paired bracket, Wilcoxon rank test). The NULL was constructed based on 3,000 randomly sampled 500 bp segments from a total 2,443,769 aligned segments.

We also revisited the new ILS set for evidence of gene ontology enrichment. Specifically, we intersected both clustered and non-clustered H-C and H-B 500 bp segments based on GRCh38 RefSeq annotation and assessed GO enrichment using DAVID (Huang et al, 2009, PMID: 19131956). Consistent with our previous observations, the segments are enriched for immunity-

related genes (e.g., glycoprotein, and EGF-like domain, ect.) but also some signal for cell adhesion and motor function (e.g., microtubule motor activity, dynein heavy chain, domain-1, IQ motif and Laminin G domain, etc.) (Supplementary Note Table S43).

Supplementary Note Table S43. GO enrichment analysis of different classes of ILS segments overlapping with exons

	Term	Enrichment score	p_value
CLUSTERED ILS H-B (n=41) Overlapping exons	microtubule motor activity	1.21	9.40E-03
	SH3 domain	1.2	4.30E-02
CLUSTERED ILS H-C (n=36) Overlapping exons	extracellular matrix organization	2.51	3.00E-03
	Cell adhesion	2.21	3.30E-03
	Glycoprotein	1.61	8.10E-03
	Calcium/transmembrane region	1.31	1.00E-04
NON-CLUSTERED ILS H-B Overlapping exons H-B (n=765)	ATP-binding	5.05	9.30E-08
	ECM-receptor interaction	3.69	4.00E-07
	Dynein heavy chain, domain-1	3.54	2.20E-06
	SNF2-related	2.73	2.70E-05
	Laminin G domain	2.71	1.10E-08
	domain: Fibronectin type-III 3	2.55	1.90E-05
	von Willebrand factor, type A	2.39	1.00E-04
	Platelet Amyloid Precursor Protein Pathway	2.13	4.90E-05
	Epidermal growth factor-like domain	2.12	8.80E-07
Glycoprotein	2.07	8.90E-04	
NON-CLUSTERED ILS Overlapping H-C (n=806)	Pleckstrin homology-like domain	5.09	2.70E-06
	ATP-binding	3.65	2.00E-05
	EGF-like domain	2.92	4.10E-07
	Dynein heavy chain, domain-1	2.81	9.40E-05
	Rho guanyl-nucleotide exchange factor activity	2.8	1.30E-04
	WD40/YVTN repeat-like-containing domain	2.65	3.40E-06
	Extracellular matrix	2.49	5.80E-06
	Glycoprotein	2.42	6.50E-05
	IQ motif, EF-hand binding site	2.42	3.30E-05
compositionally biased region: Cys-rich	2.13	5.80E-05	

*the number of genes

With respect to the observation of balancing selection, it should be noted that ~5% of the genes associated with ILS show evidence of changes in gene structure (frameshift, premature stop, start losses). For example, restricting our analysis to ILS exons, we observe 77 CDS changes in 51 genes including stop/start-loss. Among these, 18 occur in bonobo, 32 in chimpanzee and 27 can be assigned to the ancestral Pan lineage (Supplementary Note Table S44).

Supplementary Note Table S44. Polymorphic gene disruption and ILS exons

chr	pos	ref	alt	Consequence	SYMBOL	EXON	Protein_ position	Amino_ acids	Lineage
chr1	24082032	T	TGGGGTCACCTTCCAGC CTTACCTTGACAGACCCG GGTGGGGATGGGCTGC TGAG	frameshift_variant	MYOM3	18//37	750	N/TQQPIPT RVCKVRLE GDPX	Chimp
chr1	152307613	C	A	stop_gained	FLG	3//3	2425	E//*	Chimp
chr1	152308813	CAT	C	frameshift_variant	FLG	3//3	2024	H//X	Chimp
chr1	152308819	C	G,CTG	frameshift_variant	FLG	3//3	2023	G//QX	Chimp
chr1	152311694	C	T	stop_gained	FLG	3//3	1064	W//*	Chimp
chr1	152312127	G	GCC	frameshift_variant	FLG	3//3	920	A//GX	Chimp
chr1	152312129	ATG	A	frameshift_variant	FLG	3//3	919	H//X	Chimp
chr1	155688246	A	AG	frameshift_variant	YY1AP1	1//10	73	P//PX	Chimp
chr1	159313957	G	A	stop_gained	OR10J3	1//1	235	Q//*	Pan
chr1	159314580	AC	A	frameshift_variant	OR10J3	1//1	27	V//X	Chimp
chr10	21556792	TG	T	frameshift_variant	MLLT10	4//4	131	C//X	Pan

* This is a partial table excerpt; full table in Supplementary Note

In comparison to all genes in the genome, where we identify 3,384 such polymorphic variants (693 in bonobo, 1,233 in chimpanzee, and 1,458 in Pan lineage) resulting 1,990 gene disruptions, ILS exons (77/1,446 or 5.3%) are significantly enriched when compared to the genome-average (1.5% or 3,384/222,329) ($p < 0.00001$, chi-square test) (Supplementary Note Table S45). Interestingly, these results are consistent with long-term balancing selection for gene loss partially explaining the elevated dN/dS ratio, i.e., relaxed selection.

Supplementary Note Table S45. Distribution of polymorphic gene-disruption events in ILS exons versus genome

	bonobo	chimpanzee	pan	total
ILS exons (1446*)	18	32	27	77 (51**)
Genome-wide exons (222329*)	693	1233	1458	3384 (1990**)

*the number of exons for analysis

**the number of disrupted genes

These observations further strengthen our original observations. We summarize these major findings in a new paragraph at the end of the results section of the manuscript:

“ To further investigate the functional significance of clustered ILS segments, we extended the ILS analysis (Supplementary Note) across 15 million years of hominid evolution by inclusion of orangutan and gorilla ape data. Using this deeper ape phylogeny, ILS estimates for the human genome increase to >36.5% (Supplementary Note Table S34, Supplementary Note Fig. S27) similar to (albeit still greater than) earlier estimates^{10,15}. We measured the inter-ILS distance and observed a consistent non- random pattern of clustered ILS for these deeper topologies (Supplementary Note Fig. S33) with more ancient ILS showing an even greater proportion of clustered sites (Supplementary Note Fig. S33). Once again, we observe a significant elevated mean dN/dS in clustered H-C and H-B ($p < 2.2e-16$, mean = 0.366) as well as clustered O-H and O-G-H topologies ($p < 2.2e-16$, mean = 0.316) when compared to the null distribution (Supplementary Note Fig. S34). A GO analysis⁵⁷ of the genes intersecting these combined data confirm the most significant signals for immunity (e.g., glycoprotein ($p = 1.3E-25$), immunoglobulin-like fold/FN3 ($p = 2.4E-20$)), but also genes related to

epidermal growth factor signaling ($p = 1.4E-18$), solute transporter function (e.g., transmembrane region ($p = 1.3E-25$), and specifically calcium transport ($p = 3.7E-8$) (Supplementary Note Table S42). While ILS regions, in general, show single-nucleotide polymorphism diversity patterns consistent with balancing selection, it is noteworthy that both clustered and non-clustered ILS exons show a significant excess of polymorphic gene-disruptive events consistent with the action of relaxed or balancing selection (Supplementary Note Fig. S36). An examination of these gene-rich clustered ILS regions shows a complex pattern of diverse ILS topologies consistent with deep coalescent operating across specific regions of the human genome as has been reported for major histocompatibility complex (Supplementary Note Fig. S66).”

Minor comment:

3. The genome assembly itself is fine, but is not as leading edge as proclaimed/implied by the text. Thus, I recommend revising the list of necessary methodological steps for high-quality assembly construction in the introduction; i.e., not only the pathway combination used in this case. I focused my review on the analytical components of the manuscript.

This is a fair point. Ours is only one approach and advances in technologies especially over the last six months such as the implementation of HiFi and ultralong read data from ONT and assembly algorithms that incorporate them will continue to improve genome assembly continuity and accuracy. Related to this, since our original submission we have generated an additional 40-fold High-fidelity (HiFi) sequence data by circular consensus sequencing from the same source genome (Mhudiblu) and used this to further correct remaining sequencing errors (Table R2-2). We used Racon (two rounds) to error-correct the genome eliminating some of the remaining errors for an overall accuracy 1 error every 12,882 bp (QV=41.1).

We revised the description to be more up-to-date and inclusive of new advances in the field of sequence and assembly

“The development of such new references, however, is far from an automated process. Although long-read sequencing has driven the development of more contiguous sequence, it still needs to be coupled with other orthogonal technologies, such as strand-sequencing (Strand-seq)¹⁶⁻¹⁸, optical mapping¹⁹, and molecular cytogenetics (FISH)²⁰ in order to generate chromosomal-level assemblies that are not simply “humanized” by alignment to the human reference genome. This is only one of many approaches^{21,22} being developed from advances in sequencing technologies to generate complete or nearly complete genome assemblies for the first time.”

Referee #3 (Remarks to the Author).

Catacchio et al describe a reference grade bonobo genome assembly obtained through a mix of technologies, including PacBio Hi-Fi, optical mapping, and Strand-Seq. This assembly is a significant technical improvement over the previous version, which was produced using an older technology (Roche 454) and hence was fragmented. The new assembly thus enables the study of repeat elements (Sine, Alu, ERV), segmental duplications, and inversions.

Bonobo is particularly interesting due to its recent speciation from chimpanzee, which provides a vantage point to hominid evolution. Capitalizing on their improved assembly, the authors spotlight several structural variants, including an exon deletion in ADAR1 and whole gene deletions (SAMD9, LYPD9), as having potential biological significance. The most interesting findings relate to ILS, which appears to cluster in the genome and is enriched in specific pathways (photoreceptor for human-bonobo, EGF pathway for human-chimpanzee ILS). ILS segments, particularly those that cluster, also have higher dN/dS.

Based on this referee as well as the comments of Referee #2 (see above), we extended the ILS analysis and replicate our initial observations and show that a subset of ILS segments cluster across various topologies and that these clusters are associated with elevated dn/ds (see below). We appreciate the referee's detailed review of our manuscript and address additional comments in the point-by-point response below.

Though the work is an impressive technical feat, it is not the first to use long-read sequencing to generate reference-grade assemblies without the help of the human genome. It is also not the first assembly of the bonobo genome, albeit a much higher quality assembly than the one previously published in 2012 by the Paabo group.

We agree that this is not the first time to generate a free reference-guide ape assembly, but we would like to emphasize that a high-quality bonobo assembly is critical for hominid evolutionary analyses. With respect to the previous assemblies, >99% gaps are closed and the QV of the bonobo assembly has significantly improved. Related to the quality of our assembly, since our original submission we generated an additional 40-fold High-fidelity (HiFi) sequence data by circular consensus sequencing from the same source genome (Mhudiblu) and used this to further correct remaining sequencing errors. We used Racon (two rounds) to error-correct the genome eliminating ~200,000 remaining errors for an overall accuracy of 1 error every 12,882 bp (QV=41.1).

The novelty then rests on the biological implications of the improved bonobo reference, and presumably the new insight that can be gained into hominid evolution. As it stands, however, the biological insights gained from this improved bonobo assembly seem minor.

We believe this manuscript represents both a technical as well a biological advance, which are both linked, and that the advances are much more significant. We feel this paper is of high impact for three reasons:

- 1) The new genome assembly is orders of magnitude superior to the previous assembly by almost every metric and such high-quality genomes are critical for identifying the genetic differences that make us human. As an example, the original Prufer assembly was estimated to carry 560 Mbp of unresolved sequence (annotated as N's) while Mhudiblu assembly now has only 36.5 Mbp of unresolved sequence (N's). Specifically, we have added 255 Mbp of novel sequence to the genome including an additional 212 Mbp not

assigned to chromosomes—the latter consists of various heterochromatic sequence, especially located within subterminal heterochromatic gaps but importantly is now represented (see Fig. R3-1 below). As a result of this improvement, 99.5% of the genes are now complete without frameshift, effectively improving the annotation of 38.4% of the genes. This allows the first comprehensive assessment of hominid gene loss. In addition to this carefully annotated bonobo genome 99.5% of the gaps are filled and most genes are complete. We provide numerous ancillary resources full-length cDNA sequencing (Iso-Seq), Strand-seq datasets, optical mapping data and HiFi sequencing data for the first time which facilitate all future hominid genomic comparisons. This advance in quality is especially critical for species where we are focused on the differences (as opposed to questions of conservation).

- 2) As a result of these improvements, we provide a much clearer assessment of the rapidity at which hominid genomes can change especially structurally even when separated by a short ~1.5 million years. Most of the genic differences, changes in rates of retrotransposition, and structural differences are novel and differ dramatically from previous estimations based on draft genomes and Illumina datasets.
- 3) Third, and perhaps most importantly, our comparison of bonobo to chimpanzee and human nearly doubles previous estimates of incomplete lineage sorting (ILS) among chimpanzee, human and bonobo (Prufer et al, 2012 PMID: 22722832). This is because the more complete long-read ape genomes allowed for ~3-fold increase in the portions of the genome that could be aligned and analyzed (Table R3-1). Our analysis identifies a subset of clustered ILS that appear to be rapidly evolving. Based on the referee's suggestion we extended/replicated this observation based on a broader analysis of the ape phylogeny (including gorilla and orangutan) and now show that it holds for other ILS topologies. This finding is novel and, we believe of broader interest not only because it suggests more extensive sharing of genetic variants among apes and humans, but because it provides strong evidence that ILS is strictly not a random process. Instead, we find a specific subset of clustered ILS between human, chimpanzee, and bonobo enriched in glycoprotein and EGF signaling (many innate immune response genes) that appear to be subject to positive selection.

Fig R3-1. Novel bonobo sequence. The new assembly adds 255 Mbp of novel sequence to 12,964 distinct locations (red vertical lines) in the long-read Mhudiblu bonobo assembly (Mhudiblu_PPA_v0) when compared to the previous bonobo genome assembly (panpan1.1/GCA_000258655.2; Prufer et al, 2012 PMID: 22722832). In addition a large number of shorter contigs (4,271) corresponding to heterochromatic sequences are also shown as “unplaced1” and “unplaced2” (together comprising 259 Mbp of which 212 Mbp was not previously observed by Prufer; ordered from largest to smallest contigs). A marker of subterminal cap (pCht satellite) (teal vertical lines) as well as homologous alignments shows that three-quarters of this sequence corresponds to telomeric heterochromatin.

Table R3-1. ILS comparison panpan1.1 bonobo assembly vs. Mhudiblu v0

	H-B (bp)	H-C (bp)	Total analyzed sequences (bp)
panpan1.1 (Prufer,2012,TableS8.2)	12,942,453	13,756,464	833,383,247
Mhudiblu_v0 (500 bp resolution)	51,669,000	51,098,500	2,425,788,500

*H-B and H-C ILS regions (in terms of bp) identified based on the previous Prufer bonobo and Mhudiblu_v0.

Because of gaps in early draft assemblies, only 833 Mbp (Table R3-1) of the genome could be analyzed for ILS (Prufer et al, 2012 PMID: 22722832) as compared to 2,426 Mbp with more contiguous long-read genome assemblies (Table R3-1). Concomitantly, this ~2.91 fold increase has led to a larger and more accurate estimate of ILS at 500 bp (approximately doubling the % of ILS segments).

We reworked the text to highlight the non-incremental advances of the paper and importance for studies of hominid evolution. In particular, as suggested, we worked the last three months to provide additional evolutionary insights by 1) performing a more comprehensive gene-disruptive analysis across the ape phylogeny; 2) extending the ILS analysis to include gorilla and orangutan sequence data 3) investigating other models of selections underlying ILS and its clustering and 4) working up the MHC region and EIFA3 gene family expansion to provide more complete stories (see below).

Indeed, the authors are able to reconstruct many more instances of repeat elements (L1, Alu, SVA, PtERV1) and structural variants (segmental duplications, inversions), than before. However, the highlighted examples seem cherrypicked rather than nominated by a statistical model or rigorous genome-wide analysis. These anecdotes spark plausible but speculative hypotheses about the role of specific pathways (e.g. gut homeostasis or pox virus susceptibility) in bonobo / hominid evolution.

The referee raises a fair point. In response, we performed a much more systematic analysis of gene loss events across the ape phylogeny by focusing on gene-intersecting structural variation events (≥ 50 bp in length) based on a comparison of long-read assemblies of chimpanzee, bonobo, gorilla and orangutan. We also considered insertion/deletion events (< 50 bp) because such events are equally disruptive with respect to gene loss if they introduce a frameshift (Fig. R3-2). In order to validate all events (especially indels which are subject to homopolymer errors within long-read data), we generated HiFi sequencing data from the same four ape genomes (chimpanzee (40.1-fold (X) coverage, bonobo (37.9X), gorilla (31.3X) and orangutan (24.7X) (Table R3-2). HiFi data produces long-read data through circular consensus sequencing which is $>99.9\%$ accurate, allowing us to confirm structural variant and indel events discovered from the CLR-based assemblies. Mutational events were then subsequently followed up as fixed or polymorphic based on genotyping against a population of samples for each species ($n=27$ genomes; 7-10 samples per species) where Illumina WGS data were available. We provide an overview of gene loss (Fig. R3-2) and then summarize the SV and indel analyses separately.

Table R3-2. Summary of Ape HiFi PacBio WGS data

Species	Coverage	Accession
Chimpanzee	40.1	Clint SRR12517369- SRR12517374, SRR12517378, SRR12517389- SRR12517390
Bonobo	37.9	Mhudiblu SRR13443658 SRR13446350
Gorilla	31.3	Kamila SRR13446351 SRR13446352
Orangutan	24.7	Susie SRR12517385- SRR12517387

Figure R3-2. Overview of lineage-specific fixed SV and gene loss. The schematic summarizes the total number of fixed insertion (INS) and deletion (DEL) (>50 bp), on each branch of the ape phylogenetic tree. The number of events resulting in genic changes or leading to a frameshift based on an indel is indicated (brackets). The number of events on the human branch are based on a previous analysis and following our criteria in this study (Kronenberg et al., 2018 PMID: 29880660). Fixed versus polymorphic events were determined based on genotyping of 27 ape WGS Illumina samples (Supplementary Note Table S35).

Supplementary Note Table S35. Illumina WGS datasets from ape populations.

Lineage	NAME	SRA Accession	BioProject Accession
bonobo	Bono	SRS396219	PRJNA189439
	Catherine	SRS396206	
	Desmond	SRS396205	
	Dzeeta	SRS396202	
	Hermien	SRS396203	
	Hortense	SRS395319	
	Kombote	SRS396207	
	Kosana	SRS396201	
	Kumbuka	SRS396603	
	Natalie	SRS396238	
chimpanzee	Linda	ERS1286216	PRJEB15086
	Negrita	ERS1286220	
	Frederike	ERS1286243	
	Alice	ERS1286218	
	Blanquita	ERS1286221	
	Coco	ERS1286245	
	Tibe	ERS1286222	
	Ikuru	ERS1286238	
	Cleo	ERS1286240	
	Bihati	ERS1286241	
gorilla	Amani	SRS396847	PRJNA189439
	Banjo	SRS396826	
	Delphi	SRS396829	
	Dian	SRS396828	
	Kaisi	SRS396605	
	Kolo	SRS396831	
	Mimi	SRS396827	

*Coverage greater than 20

1. Lineage-specific SVs and gene disruption analyses. We applied three callers (PBSV, Sniffle, and Smartie-SV) based on a comparison of four genome assemblies (bonobo (Mhudiblu_PPA_v0), chimpanzee (Pantro6/Clint_PTRv2), gorilla (Kamilah_GGO_v0) and human (GRCh38)) to identify SVs and then extracted the bonobo-specific, chimpanzee-specific and pan-specific SVs--i.e. shared between chimpanzee and bonobo. Using Paragraph (Chen, 2019.), we next genotyped all SVs against Illumina WGS data available from 10 bonobos, 10 chimpanzees and 7 gorillas (Prado-Martinez et al. 2013 PMID: 23823723 and De Manuel, et al, 2016 PMID: 27789843). Based on the genotypes, we calculated the Fst between populations and considered an event as fixed and lineage-specific if Fst >0.8 between populations from different species. The ensembl variant effect predictor (VEP) was applied (McLaren et al, 2016.PMID: 27268795) to annotate the SVs in order to identify SVs disrupting genes (Supplementary Note Table S50) as well as events affecting potential noncoding regulatory DNA. We validated all gene disruption events by mapping high-fidelity (HiFi) sequence reads generated from the bonobo, chimpanzee, gorilla and two human genomes back to GRCh38. Relatively few gene disruptions mediated by structural variation were discovered in the Pan lineage (eg keratin-associated gene Supplementary Note Fig. S46) and much more common were structural changes that led to a significant modification of protein structure (eg. mucin or zinc finger genes Supplementary Note Fig. S47).

Supplementary Note Table S50. The fixed ape SVs affecting exons

Lineage-specific	HUMAN-CHR	HUMAN-START	HUMAN-END	SV-TYPE	SIZE	ANNOTATION	GENE	WGAC	WSSD (SDA)	GENE ID	EXON	pLI
bonobo	chr1	154601820	154601966	DEL	147	inframe_deletion	ADAR	0	0	ENSG00000160710	2//15	9.91E-02
bonobo	chr1	248739523	248763827	DEL	24305	stop_lost	LYPD8	0	0	ENSG00000259823	1-7//7	NA
bonobo	chr11	63119193	63119261	DEL	69	inframe_deletion	SLC22A24	0	0	ENSG00000197658	3//10	3.09E-03
bonobo	chr3	195789477	195790190	DEL	714	inframe_deletion	MUC4	0	0	ENSG00000145113	2//25	5.45E-16
bonobo	chr7	93077971	93119434	DEL	41464	transcript_ablation	SAMD9	0	0	ENSG00000205413	1-3//3	5.21E-30
chimp	chr19	22316718	22316719	INS	84	inframe_insertion	ZNF729	84	0	ENSG00000196350	4//4	4.00E-01
chimp	chr9	113425411	113425412	INS	314	stop_gained	C9orf43	0	0	ENSG00000157653	10//14	2.27E-10
pan	chr1	248589569	248604503	DEL	14935	transcript_ablation	OR2T10	0	0	ENSG00000184022	1-2//2	7.10E-04
pan	chr16	3352155	3359732	DEL	7578	transcript_ablation	OR2C1	0	0	ENSG00000168158	1//1	3.46E-05

Coordinates based on human GRCh38 genome.

*This is a partial table excerpt; full table in Supplementary Note.

Supplementary Note Figure S46. Loss of keratin-associated gene in chimpanzee and bonobo lineages. a) A 25.7 kbp deletion results in the complete loss of hair keratin-associated protein (*KRTAP19-6*) in *bonobo* and *chimpanzee*. b) Sequence read-depth genotyping of deletion in human and ape Illumina WGS data (number of samples) confirms a *Pan*-specific loss fixed in both bonobo and chimpanzee.

Supplementary Note Figure S47. A Pan-specific fixed genic insertion. **a**, A 72 bp insertion in the coding sequence of *ZNF280C* in chimpanzee and bonobo based on genomic sequence alignment among bonobo, chimpanzee, gorilla, and human. **b**, A 24 amino acid insertion specific to bonobo and chimpanzee. **c**, Insert occurs at position 561 in the *ZNF280C* protein.

We also considered the potential loss of noncoding regulatory elements by intersecting lineage-specific SVs with ENCODE V3 (Snyder et al; 2020, PMID: 32728248) catalog of functional elements in humans (Supplementary Note Table S46). We assigned regulatory elements to specific genes if they occurred within the body of the gene (UTR and intron) or the elements are located within 5kb downstream/upstream of the genes. We identified 662 disruptions (insertions and deletions) of noncoding regulatory elements in the bonobo lineage and 356 events in the chimpanzee (Supplementary Note Table S46). Gene ontology enrichment analyses were

performed using DAVID (Huang et al. 2009, PMID: 19131956) for SVs associated with lineage-specific gene disruptions or loss of regulatory DNA. For bonobo specific-SVs, we find genes enriched in membrane regions/topological domain: Extracellular ($p=2.4E-4$), regulation (eg; phosphate-binding region ($p=7.8E-4$), zinc finger domain ($p=1.5E-2$)), and neuron related proteins (ANK repeats, ($p=8.1E-3$), synapse ($p=4.4E-3$), dopaminergic synapse ($8.4E-2$)). Bonobo contrasts with chimpanzee-specific SVs, which show an enrichment only in the cadherin pathway ($p=6.10E-03$). Gene loss in the ancestral Pan lineage (shared between chimpanzee and bonobo) show enrichments in postsynaptic membrane ($p=1.2E-7$), PDZ domain ($p=4.5E-5$), calcium transport ($p=2.E-3$), regulation (phosphate-binding region ($p=3.8E-3$), GTPase activator activity ($p=5.4E-3$) as well as coronary vasculature development ($p=7.9E-2$) and facial nerve structural organization ($p=4E-2$) (Supplementary Note Table S51). Although potentially interesting, it should be noted that the low number of events makes significance of all enrichments relatively modest.

Supplementary Note Table S46. Summary of lineage-specific SVs

	bonobo			chimpanzee			pan			gorilla		
	all (against hg38)	specific	fixed	all (against hg38)	specific	fixed	all (against hg38)	specific	fixed	all (against hg38)	specific	fixed
Insertion	61,078	15,786 (9.76 Mbp)	3,604 (3.3 Mbp)	63,525	17,761 (10.61 Mbp)	1,959 (1.83 Mbp)		18,742 (12.14 Mbp)	6,646 (6.27 Mbp)	72,793	42,009 (29.13 Mbp)	17,858 (15.99 Mbp)
Deletion	59,246	7,082 (6.82 Mbp)	1,965 (2.36 Mbp)	61,182	7,542 (6.89 Mbp)	1,047 (1.11 Mbp)		14,309 (16.10 Mbp)	6,852 (8.98 Mbp)	69,668	28,194 (27.60 Mbp)	12,309 (13.26 Mbp)
Disrupted exon/ UTR SVs			148			57			293			586
Disrupted exons (validate with HiFi reads)			5 (LYPD8 has half deletion in the orangutan)			2			15 (APOL1&MA GEB6 have half deletion in the orangutan)			20 (MTERF4 has half deletion in the orangutan)
Putative encode regulatory sequence			465(del)+ 197(ins)			252(del)+ 104(ins)			1,753(del)+ 404(ins)			2,408(del)+ 1,038(ins)

Supplementary Note Table S51. Gene ontology enrichment analyses for loss of functional elements

	Term	Enrichment Score	P_Value
for the genes which contains bonobo-specific SVs that intersect with ENCODE (n=381*)	Membrane region/ topological domain:Extracellular	3.28	2.40E-04
	nucleotide phosphate-binding region:ATP	2.59	7.80E-04
	Zinc finger, LIM-type	1.72	1.50E-02
	ANK repeat	1.56	8.10E-03
	ECM-receptor interaction	1.45	9.90E-03
	Host cell receptor for virus entry	1.39	2.30E-02
	Proteoglycans in cancer	1.36	1.40E-04
	ErbB signaling pathway	1.27	1.60E-03
	epidermal growth factor receptor signaling pathway	1.26	1.40E-02
	Fatty acid metabolism	1.21	1.90E-02
	Synapse	1.19	4.40E-03
	Dopaminergic synapse	1.11	8.40E-02
	metal ion-binding site:Magnesium	1.09	6.20E-02
	positive regulation of endothelial cell migration	1.07	8.60E-02
for the genes which contains chimp-specific SVs that intersect with ENCODE (n=187)	Cadherin conserved site	1.15	6.10E-03
for the genes which contains pan-specific SVs that intersect with ENCODE (n=1040)	Pleckstrin homology-like domain	6.15	1.20E-09
	postsynaptic membrane	3.32	1.20E-07
	CRAL-TRIO domain	2.47	3.00E-03
	PDZ domain	2.38	4.50E-05
	WW domain	2.37	2.80E-03
	ATPase, dynein-related, AAA domain	1.89	1.20E-03
	Calcium transport	1.71	2.00E-03
	Aminopeptidase	1.70	3.70E-02
	Calmodulin-binding	1.61	6.20E-03
	clathrin-mediated endocytosis	1.60	4.70E-02
	C2 calcium-dependent membrane targeting	1.59	7.80E-03
	coronary vasculature development	1.54	7.90E-02
	nucleotide phosphate-binding region:ATP	1.49	3.80E-03
	GTPase activator activity	1.44	5.40E-03
	domain:BEACH	1.44	3.00E-02
	Ubiquitin system component Cue	1.43	2.90E-02
	CUB domain	1.41	3.60E-02
	Phosphotyrosine interaction domain	1.39	5.90E-02
	regulation of calcium ion transport	1.37	6.80E-02
	SH3 domain	1.31	3.60E-02
	facial nerve structural organization	1.27	4.00E-02
	Cyclic nucleotide-binding domain	1.24	4.70E-02
	phosphatidylinositol binding	1.22	8.60E-02
AAA+ ATPase domain	1.18	2.00E-02	
Potassium channel, voltage dependent, KCNQ	1.15	4.20E-02	

2. Indel gene frameshift analyses with HiFi read validation.

We also investigated potential gene loss as a result of indel mutation events (<50 bp) since such events are functionally equivalent to large structural variation events. We initially identified 323 frameshift mutations for 119 genes in the bonobo assembly based on comparison to human GRCh38. These events were identified from the CAT annotation of the bonobo assembly, and were filtered to include only events on the default isoform (GENCODE's MANE_select isoform) for each gene. We validated all events using HiFi sequencing data from the same source (Mhudiblu) (Supplementary Note Table S52). This was done by using the HiFi data to call

variants using FreeBayes and check for consistency in variant calls. As a control, we also analyzed HiFi data from two humans (Yoruban and Puerto Rican samples) and found that only 4 of these variants were also identified as a frameshift in at least one of the two humans. We excluded these from subsequent analysis. In order to define lineage-specificity, we identified frameshift mutations in the chimpanzee and gorilla genomes as described above, and then compared those to the set of bonobo mutations. We identified 423 frameshifts corresponding to 186 genes in gorilla and 328 frameshifts corresponding to 149 genes in chimpanzee (Supplementary Note Fig. S48). We used HiFi sequencing data from an outgroup ape (orangutan) to validate lineage-specificity. Finally, we also used the 27 WGS ape short-reads to genotype these frameshifts by GATK and used the same criteria ($F_{st} \geq 0.8$) to identify the fixed frameshift events in each lineage (Supplementary Note Table S52). Please note that due to the inability to accurately map short-read Illumina data to duplicate genes we limited the analysis to potential indels and frameshifts mapping outside of segmental duplications (Supplementary Note Fig. S48)--i.e. to unique regions of the ape genome. Similar to the structural variant analyses, fixed indel events frequently occurred in genes tolerant to mutation or resulted in modifications to the carboxy terminus, with a few exceptions highlighted below (Supplementary Note Fig. S49).

Supplementary Note Figure S48. Fixed indel mutations resulting in gene frameshifts. **a**, Frameshift mutation events discovered based on CAT annotation of individual ape genomes to human GRCh38. **b**, HiFi-validated frameshift mutations mapping to unique regions of the genome (outside of SDs) and that are fixed in each population based on analysis of Illumina WGS data from 27 ape genomes (Supplementary Note Table S35). Fixed mutations show $F_{st} > 0.8$ for a given lineage. Comparisons between species were made by liftOver to GRCh38. **c**, Venn diagram of fixed lineage-specific and shared gene loss at the level of individual genes based on validated frameshifts in (b).

Supplementary Note Table S52. Fixed frameshifts in the ape lineages with HiFi and WGS validation

Lineage	Genes	Gene ID	Indel type	Human indel coords	PLI
bonobo+chimp+gorilla	WDR78	ENSG00000152763.17	Deletion	chr1:66924747-66924749	1.89E-03
bonobo+chimp+gorilla	OR11L1	ENSG00000197591.3	Deletion	chr1:247840962-247840963; chr1:247840964-247840965	5.79E-02
bonobo+chimp+gorilla	SCIMP	ENSG00000161929.15	Deletion	chr17:5210815-5210817	6.48E-03
bonobo+chimp+gorilla	GNG14	ENSG00000283980.1	Deletion	chr19:12688250-12688252	NA
bonobo+chimp+gorilla	OCSTAMP	ENSG00000149635.3	Deletion	chr20:46541566-46541568	7.13E-04
bonobo+chimp+gorilla	OR2B2	ENSG00000168131.4	Deletion	chr6:27911399-27911400; chr6:27911401-27911402	9.32E-03
bonobo+chimp+gorilla	C12orf60	ENSG00000182993.5	Deletion	chr12:14823553-14823554; chr12:14823555-14823556	4.82E-02
bonobo+chimp+gorilla	ZNF843	ENSG00000176723.10	Deletion	chr16:31436425-31436427; chr16:31436424-31436426	1.35E-03
bonobo+chimp+gorilla	CMTM5	ENSG00000166091.21	Deletion	chr14:23378759-23378761	0.32
bonobo	MTF2	ENSG00000143033.18	Deletion	chr1:93134088-93134089; chr1:93134092-93134093	1.00
bonobo	ZNF780B	ENSG00000128000.16	Deletion	chr19:40035339-40035340; chr19:40035342-40035343	1.28E-02
bonobo	JGSF23	ENSG00000216588.9	Deletion	chr19:44627544-44627546	0.13

* This is a partial table excerpt; full table in Supplementary Note

Supplementary Note Figure S49. Fixed gene-disrupting indels in the *Pan* lineage **a**, 1 bp deletion in *CST9L* leads to a premature stop codon, event fixed in bonobo and chimpanzee. **b**, 1 bp deletion in *RFX8* leads to a premature stop codon, fixed in bonobo and chimpanzee. **c**, 1 bp deletion in *FBXW12* leads to a premature stop codon, fixed in bonobo and chimpanzee.

We revised the main text to include a new paragraph incorporating these more complete analyses:

“Gene and regulatory DNA disruptions. We focused on a detailed analysis of gene and regulatory DNA loss on the ape lineage based on human gene annotations and SV comparisons in bonobo, chimpanzee, gorilla, and orangutan genomes¹⁵. For example, we identified 381 bonobo-specific and 185 chimpanzee-specific SVs that intersect ENCODE regulatory elements that could be assigned to a gene (Supplementary Note). Bonobo-specific events are enriched in membrane-associated genes with extracellular domains while chimpanzee-specific events are associated with cadherin-related genes (Supplementary Note Table S51). Interestingly, fixed deletions (n=1,040) on the *Pan* lineage (shared between chimpanzee and bonobo) show an enrichment for the loss of putative regulatory elements associated with post-synaptic genes (3.32 enrichment; $p = 1.2 \times 10^{-7}$) and pleckstrin homology-like domains (6.15 enrichment; $p = 1.20 \times 10^{-9}$). Disruptions of protein-coding sequence were far less abundant and we extended this analysis to include both SV and indel mutation events (<50 bp) because both can result in a gene loss or gene disruption due to premature truncation. We validated all 110 events by generating high-fidelity genomic sequencing for each of the ape reference genomes and restricting to those events that could be genotyped in a population of genomes (Supplementary Note). As expected, many fixed gene-loss events occurred in genes tolerant to mutation, redundant duplicated genes, or genes where the event simply altered the structure of the protein. For example, we identified and validated a complete 25.7 kbp gene loss of one of the keratin-associated genes (*KRTAP19-16*) associated with hair production in ancestral lineage of chimpanzee and bonobo (Supplementary Note Fig. S46). In the bonobo lineage, we identified five fixed SVs affecting protein-coding genes (Table 2 and Fig. 4 b-d), but only two of which completely ablate the gene when compared to all other apes. *LYPD8*, for example, which encodes a secreted protein that prevents gram-negative bacteria invasion of colonic epithelium, has been totally deleted by a 24.3 kbp bonobo-specific deletion (Fig. 4c). Similarly, *SAMD9* (SAMD FamilyMember 9) has been totally deleted by a 41.46 kbp bonobo-specific deletion (Fig. 4d and Supplementary Note Fig. S38) and fixed only among bonobos. The other three bonobo-specific fixed SV events in protein-coding regions all maintain the ORF, including a 49-amino acid deletion of *ADAR1*, a gene critical for RNA editing and implicated in human disease (Fig. 4b)⁵³⁻⁵⁵”

The observation of the EIF3A segmental duplication is striking, but is not developed into a story.

We performed a much more extensive analysis of this gene family (see detailed response to the specific requests below) and have developed it into a more complete story. In short, we show that the initial *EIF4A3* gene duplication occurred in the ancestral lineage of chimpanzee and bonobo approximately 2.9 mya. It then subsequently expanded and underwent gene conversion independently in the chimpanzee and bonobo lineage creating four and five copies of the *EIF4A3* gene family respectively in each lineage. In both lineages, the gene families are organized head-to-tail in direct orientation. Interestingly signals of gene conversion were detected and these correspond to a set of specific amino acid changes in the basic ancestral structure of the single ancestral copy that are now common to only chimpanzee and bonobo.

Please note that there was a typographical error in the original description (lines 273-276) which reported the gene family as *EIF3A* as opposed to the correct name *EIF4A3*. We corrected this throughout the main and supplement.

In addition to this specific example we performed a genome-wide analysis of gene expansions in both the bonobo and chimpanzee lineages. First, we identified copy number expansions and contractions in the *Pan* lineage and classified these as bonobo-specific, chimpanzee-specific, or shared (Pan-specific), compared to other hominids. This classification was based initially on short-read Illumina WGS mapping (WSSD) from 27 ape genomes (Supplementary Note Table S35) to the human reference to generate an assembly-independent assessment of copy number in order to focus on species-specific expansions as opposed to polymorphisms. Species-specific or Pan-specific events were subsequently confirmed orthogonally by read-depth analysis using the long reads and analysis of whole-genome and targeted long-read assemblies (HiFi and CLR) requiring a diploid copy number difference of at least 2. We focused on regions likely to contain genes based on Iso-Seq annotation or by Liftoff analyses (GCA_009914755.2, <https://github.com/nanopore-wgs-consortium/CHM13>). Liftoff v1.4.2 was performed with the parameters ‘-flank 0.1 -sc 0.85 -copies’ against each target genome using GRCh38 GENCODE v35 annotations as the source, in order to count the number of duplicated loci with corresponding transcript support for each gene in each assembly. To estimate number of assembled copies of each gene independent of Liftoff gene annotations, we aligned 2kbp chunks of each assembly to GRCh38 with MashMap v2.0 (Jain et al. *Bioinformatics* 2018), and merged adjacent alignments, requiring at least 6.5 kbp of contiguous sequence at 95% sequence identity. The number of assembled macaque loci corresponding to each GENCODE gene model was summarized with BEDTools. Among protein-coding gene family expansions (GRCh38 GENCODE v35), we identified 42 bonobo-specific, 12 chimpanzee-specific, and 142 shared *Pan* expansion candidates. Similarly, we identified 13 bonobo-specific, 6 chimpanzee-specific, and 56 shared *Pan* contraction candidates. For each bonobo gene duplication resolved by long-read assembly, we aligned Iso-Seq data and assessed the number of transcripts to identify predominant isoforms and potential changes in the gene structure (Supplementary Tables 11 and 12).

Supplementary Table 11. Full table of candidates: expansions

gene_ID	lineage	WSSD		Assembly CN (whole genome alignm Read depth)										Assembly CN (Liftoff)					Isoseq	Gene (hg38)	start	end	gene_type		
		Ppa	Ptr	Hsa	Ggo	Pab	HiFi CN	HiFi	CN50	v0 CN	v0 CN50	HiF CN	CLR CN	Ppa	HiFi	Hr	Hsa	Ggo						Pab	Ppa
GOLGA6L10	Ptr	13.6	16.7	15.2	12.0	13.5	6.5	7	7.5	8	4.2	3.0	3	4	2	1	1	2	1	2	1	chr15	82339993	82349475	protein_coding
PRSS57	Ptr	1.0	1.8	1.1	1.0	0.6	1.8	2	0.9	1	6.3	2.2	3	1	1	1	1	2	1	2	1	chr19	685546	695498	protein_coding
CCDC74A	Ptr	2.1	3.2	2.1	2.0	1.9	3.0	3	3.6	4	1.4	1.3	2	1	1	1	1	2	1	2	1	chr2	131527675	131533666	protein_coding
FOXDL3	Ptr	2.7	8.9	6.9	3.6	0.2	8.4	10	8.8	9	10.9	10.2	2	5	2	2	1	2	2	2	1	chr9	68302867	68305084	protein_coding

* This is a partial table excerpt; full table in Supplementary Note

As a final validation and to confirm their organization within the bonobo/chimpanzee genome, we selected five gene family expansions (*CLN3*, *EIF3C*, *RGL4*, *IGLV6-57*, *SPDYE16*) and four gene loss events (*IGFL1*, *SAMD9* (described in the original submission), *TRAV4*, *CDK11A*) for experimental validation by FISH (Supplementary Note Tables S48 and S49). Fosmid probes (n=9) corresponding to human genomic data were isolated and hybridized against human, bonobo, chimpanzee, gorilla and orangutan chromosomal metaphase spreads and interphase nuclei. Every hybridization was performed as a co-hybridization experiment combining one clone for expansion and one clone for contraction to be sure that the absence of signals expected for the contraction was due to a real absence of signals and not a technical artefact (Supplementary Note Figs. S39 and S40). This analysis confirmed all genome predictions (Supplementary Note Tables S49 and Supplementary Note Figs. S20 and S21) providing the most comprehensive resource of chimpanzee and bonobo gene family expansions. It is noteworthy that three out of four tested gene expansions show patterns of intrachromosomal interspersions and these are found adjacent to “core duplicons” (eg. *NPIP* and *GUSBP*) which

have been predicted to mediate the formation of interspersed segmental duplications in humans.

Supplementary Note Table S48. Gene functions in expanded and contracted genomic regions

Class	Gene	Description	Function	Phenotype	Notes
Expansion	CLN3	CLN3 Lysosomal/Endosomal Transmembrane Protein, Battenin	This gene encodes a protein that is involved in lysosomal function.	LOF causes neurodegenerative diseases commonly known as Batten disease or collectively known as neuronal ceroid lipofuscinoses (NCLs).	adjacent to NPIP
Expansion	EIF3C	Eukaryotic Translation Initiation Factor 3 Subunit C	EIF3C (Eukaryotic Translation Initiation Factor 3 Subunit C) is a Protein Coding gene.	Diseases associated with EIF3C include Colon Squamous Cell Carcinoma.	adjacent to NPIP
Expansion	RGL4	Ral Guanine Nucleotide Dissociation Stimulator Like 4	This oncogene encodes a protein similar to guanine nucleotide exchange factor Ral guanine dissociation stimulator. The encoded protein can activate several pathways, including the Ras-Raf-MEK-ERK cascade.	Increased expression of this gene leads to translocation of the encoded protein to the cell membrane. RGL4 expression is significantly associated with a variety of tumor-infiltrating immune cells (TIICs), particularly memory B cells, CD8+T cells and neutrophils.	adjacent to GUSBP core duplcon
Expansion	IGLV6-57	Immunoglobulin Lambda Variable 6-57	Protein Coding gene.	no phenotype associated	adjacent to a deletion
Expansion	SPDYE16	Speedy/RINGO Cell Cycle Regulator Family Member E16	Protein Coding gene. Among its related pathways are Oocyte meiosis.	no phenotype associated	high-copy duplcon
Contraction	IGFL1	IGF Like Family Member 1	The protein encoded by this gene is a member of the insulin-like growth factor family of signaling molecules. The encoded protein is synthesized as a precursor protein and is proteolytically cleaved to form a secreted mature peptide. The mature peptide binds to a receptor, which in mouse was found on the cell surface of T cells.	Increased expression of this gene may be linked to psoriasis.	
Contraction	SAMD9	Sterile Alpha Motif Domain Containing 9	This gene encodes a sterile alpha motif domain-containing protein. The encoded protein localizes to the cytoplasm and may play a role in regulating cell proliferation and apoptosis.	Mutations in this gene are the cause of normophosphatemic familial tumoral calcinosis (autosomal recessive)	
Contraction	TRAV4	T Cell Receptor Alpha Variable 4	In a single cell, the T cell receptor loci are rearranged and expressed in the order delta, gamma, beta, and alpha.	no phenotype associated	11 kbp deletion
Contraction	CDK11A	Cyclin Dependent Kinase 11A	This gene encodes a member of the serine/threonine protein kinase family. Members of this kinase family are known to be essential for eukaryotic cell cycle control.	These two genes are frequently deleted or altered in neuroblastoma.	

Supplementary Note Table S49. FISH results for expansions and contractions of bonobo and/or Pan genomes

Class	Gene	Fosmid Clones	Coords (hg38)	Heat map predictions				FISH Results										
				HSA	PPA	PTR	GGG	IPPY	HSA	PPA	PTR		GGG	IPPY				
Expansion	CLN3	170215_ABC9_3_2_000041281300_M15	chr16:28479201-28516032	S	D	D	S	S	16p	Single	XVip	Dup	XVip	Dup	XVip	Single	XVip	Single
Expansion	EIF3C	172343_ABC9_3_5_000044010100_H14	chr16:29887256-28729362	D	D	D	S	S	16p	Dup#	XVip	Dup	XVip	Dup	XVip	Single	XVip	Single
Expansion	RGL4	171515_ABC9_3_5_000046184500_C13	chr22:22879521-23714506	S	D	D	S	S	1p, 9q, 22q	Dup#	lp (weak), lXq (weak), XXIIq	Dup	lp, lqter, Vllqter, lXq, Xllq	Dup	lp, lXq, XXIIq	Dup#	Xllq	Single
Expansion	IGLV6-57	ABC8-412020005	chr22:22178597-22214773	S	SD	S	S	S	22q	Single	XXIIq	Single	XXIIq	Single	XXIIq	Single	Acrocentric chrs	Dup#
Expansion	SPDYE16	171515_ABC9_3_5_000043959400_P22	chr7:76507030-76545218	SD	D	D	SD	SD	7q	Dup	Vllq	Dup	Vllq	Dup	Vllq	Dup	Vllq	Dup
Contraction	IGFL1	170215_ABC9_3_2_000043862300_J24	chr19:46195756-46232256	S	del	del	S	S	19q	Single	No signal	del	lXXq	Single	lXXq	Single	lXXq	Single
Contraction	SAMD9	ABC8-41156300P24	chr7:93082459-93118602	S	del	S	S	S	7q	Single	No signal	del	Vllq	Single	Vllq	Single	Vllq	Single
Contraction	TRAV4	ABC8-42078300A3	chr14:21716253-21748608	S	S	del	S	S	14q	Single	XIVq (weak)	del	XIVq	Single	XIVq	Single	XIVq(weak)	Single
Contraction	CDK11A	ABC8-41133000L6	chr1:1709902-1734122	D	del	S	del	D	1p	Dup#	No signal	del	No signal	del	lp	Dup	lp	Single

Polymorphic duplication tested in three human (HG00733, GM12813 and GM24385)

§ FISH results different from predictions In bold highly duplicated pattern signals

Supplementary Note Figure S39. *Pan*-specific duplication of *CLN3* locus, and bonobo-specific deletion of *IGFL1*. HiFi read depth and WSSD of bonobo, chimpanzee, orangutan, gorilla, and human individuals relative to GRCh38 detect these events (above), which are validated by interphase FISH of each species using fosmid clones spanning the region (below).

Supplementary Note Figure S40. *Pan*-specific duplication of *EIF3C* locus, and bonobo-specific deletion of *SAMD9*. HiFi read depth and WSSD of bonobo, chimpanzee, orangutan, gorilla, and human individuals relative to GRCh38 detect these events (above), which are validated by interphase FISH of each species using fosmid clones spanning the region (below).

It seems that the ILS findings, including the clustering and increased dN/dS at these loci, provide the most potential for a biologically compelling narrative. However, the extent of ILS in bonobo-chimpanzee speciation has been previously discussed (including in Prufer 2012, to considerable depth). As a result, the ILS insights seem somewhat incremental.

We agree that the clustering of ILS and increased dN/dS of the underlying gene loci was one of the most novel findings of the work. To increase the appeal, we extended this analysis over the last 15 mya of evolution for all branches of the ape tree. As suggested by Referee #2, we repeated our analysis at a resolution of 500 bp including both orangutan (*Susie_PABv2*) and gorilla (*Kamilah_GGO_v0*) genomes. Considering only those tree topologies where there is at least 50% bootstrap support ($\geq 50\%$), we find that 36.5% (Supplementary Note Table S34, Supplementary Note Fig. S27) of the genome shows evidence of ILS with 31.92% belonging to two deeper ILS topologies (orangutan,(((bonobo,chimpanzee),gorilla),human)) and (orangutan,((bonobo,chimpanzee),(gorilla,human))). These estimates are consistent with earlier estimates of 30% (Sally, Aylwyn, et al. 2012, PMID: 22398555) and ~36% (Kronenberg, et al. 2018, PMID: 29880660). Of note, if we eliminate the requirement of bootstrap support (as was done previously), the estimate of ILS, increases to 50.26% consistent with our observation of a larger fraction of the genome under potential ILS.

Supplementary Note Table S34. Distribution of ILS segments (500 bp) using orangutan genome (Susie_PABv2) as a root

	Tree_topology	Number of tree (BS>=50*)	Proportion (BS>=50*)	Number of tree (BS>=0)	Proportion (BS>=0)
Species tree	(O,(G,((B,C),H)))	1,581,810	63.52%	2,317,762	50.26%
ILS (discordant tree)	(O,(((B,C),G),H))	407,472	16.36%	844,133	18.30%
	(O,((B,C),(G,H)))	387,309	15.55%	827,903	17.95%
	(O,(((B,H),C),G))	34,723	1.39%	163,175	3.54%
	(O,((B,(C,H)),G))	28,603	1.15%	156,105	3.38%
	(O,(((G,H),C),B))	6,959	0.28%	46,483	1.01%
	(O,((B,(G,H)),C))	6,954	0.28%	45,414	0.98%
	(O,(((B,G),C),H))	6,030	0.24%	20,167	0.44%
	(O,((B,(G,C)),H))	5,837	0.23%	19,823	0.43%
	(O,(((C,H),G),B))	5,701	0.23%	2,608	0.57%
	(O,(((B,H),G),C))	5,522	0.22%	2,539	0.55%
	(O,((B,G),(C,H)))	4,817	0.19%	43,975	0.95%
	(O,((B,H),(C,G)))	4,569	0.18%	40,795	0.88%
	(O,(((B,G),H),C))	2,019	0.08%	17,515	0.38%
(O,(((C,G),H),B))	1,935	0.08%	17,267	0.37%	
Total number of tree/proportion		2,490,260	1	4,611,987	1
The total analyzed genome size (with respect to hg38 (3.1 Gbp))		40.16%	NA	74.39%	NA

BS≥50 requires greater than 50% bootstrap values in support of the ML tree topology.

Supplementary Note Figure S27. Chromosome view of ILS. The schematic depicts human chromosomes 3, 4, 7 and X (GRCh38) with distribution of six different ILS shown as density plots. A subset of the major topologies are shown above and below the line (as indicated by color and arrow) and examples are shown with and without using orangutan as an outgroup.

Based on this ILS extended analysis, we next focused on the different classes of ILS and assessed whether there was evidence of clustered ILS segments as we had observed for chimpanzee, human & bonobo. Then, we tested whether those clustered segments showed evidence of positive selection. We restricted the clustered analysis to high-confidence ILS segments ($BS \geq 50$) and first tested whether those inter-ILS distances were non randomly distributed when compared to the null (Supplementary Note Fig. S33). We considered the four most abundant ILS topologies, namely:

- 1) O-H: (orangutan,(((bonobo,chimpanzee),gorilla),human)),
- 2) O-(H,G): (orangutan,(((bonobo,chimpanzee),(gorilla,human))),
- 3) H-B: (orangutan,(((bonobo,human),chimpanzee),gorilla)),
- 4) H-C: (orangutan,((bonobo,(chimpanzee,human)),gorilla))).

For each topology, we observe a cluster of ILS segments that deviate significantly from the null and are not randomly distributed in the genome. We note that the proportion of clustered ILS segments differs with older topologies (more ancient ILS) showing a greater fraction of clustered

sites. For example, for the O-H and O-(H,G) topologies the proportion of clustered sites is ~32-34% while for H-B & H-C this fraction is 8-10%.

Supplementary Note Figure S33. Clustered ILS sites. The distance between adjacent ILS segments (inter-ILS) (500 bp resolution) was calculated and the distribution was compared to a simulated expectation based on a random distribution. The analysis reveals a bimodal (and possibly an emerging trimodal) pattern where a distinct subset of ILS are clustered (i.e., clustered ILS sites). Four different topologies are considered: **a**, (orangutan,(((bonobo,chimpanzee),gorilla),human)) ILS topology where 31.58% of inter-ILS are clustered; **b**, (orangutan,((bonobo,chimpanzee),(gorilla,human))) ILS topology where 33.5% are clustered; **c**, (orangutan,(((bonobo,human),chimpanzee),gorilla)) ILS topology (8.14%); and **d**, (orangutan,((bonobo,(chimpanzee,human)),gorilla)) ILS topology (9.89% of sites).

Next, we investigated whether we still observed the elevated dN/dS in clustered ILS. As before we compared the observed dN/dS values for clustered sites against a simulated set where 1000 genes were chosen at random and a genome-wide distribution was created (Supplementary Note Fig. S34) by repeating the process 100 times to generate a null distribution (mean=0.263). Using a one sample t-test statistic, we observe a significant elevated mean dN/dS in both clustered H-C & H-B ($p < 2.2e-16$, mean=0.366) and in clustered O-H & O-G-H ($p < 2.2e-16$, mean=0.316) when compared to the null. The nonclustered H-C and H-B topologies remain

insignificant ($p=0.45$, $\text{mean}=0.264$) although non-clustered O-H & O-G-H sites now show evidence of excess of amino acid replacement ($p < 2.2e-16$, $\text{mean}=0.306$) although that

difference is more subtle and occurs within the last 5% of the null distribution.

Supplementary Note Figure S34. Elevated dN/dS in clustered sites of ILS. The null distribution (gray) is based on calculation of mean dN/dS for 1000 genes drawn randomly from the genome (100 simulations) (mean: 0.263). The blue solid and dashed lines represent the mean dN/dS for clustered H-C & H-B ILS (mean: 0.366, $p < 2.2e-16$) and non-clustered H-C & H-B sites (mean=0.264, $p=0.45$), respectively. The solid and dashed purple lines represent mean dN/dS of the clustered O-H & O-G-H ILS (mean=0.316, $p < 2.2e-16$) and the non-clustered O-H & O-G-H ILS (mean=0.306, $p < 2.2e-16$).

Significance performed using the t test in R although similar results based on the null distribution.

These observations further strengthen our original observations. We summarize these major findings in a new paragraph at the end of the results section of the manuscript:

“To further investigate the functional significance of clustered ILS segments, we extended the ILS analysis (Supplementary Note) across 15 million years of hominid evolution by inclusion of orangutan and gorilla ape data. Using this deeper ape phylogeny, ILS estimates for the human genome increase to >36.5% (Supplementary Note Table S34, Supplementary Note Fig. S27) similar to (albeit still greater than) earlier estimates^{10,15}. We measured the inter-ILS distance and observed a consistent non-random pattern of clustered ILS for these deeper topologies (Supplementary Note Fig. S33) with more ancient ILS showing an even greater proportion of clustered sites (Supplementary Note Fig. S33). Once again, we observe a significant elevated mean dN/dS in clustered H-C and H-B ($p < 2.2e-16$, mean = 0.366) as well as clustered O-H and O-G-H topologies ($p < 2.2e-16$, mean = 0.316) when compared to the null distribution (Supplementary Note Fig. S34). A GO analysis⁵⁷ of the genes intersecting these combined data confirm the most significant signals for immunity (e.g., glycoprotein ($p = 1.3E-25$), immunoglobulin-like fold/FN3 ($p = 2.4E-20$)), but also genes related to epidermal growth factor signaling ($p = 1.4E-18$), solute transporter function (e.g., transmembrane region ($p = 1.3E-25$), and specifically calcium transport ($p = 3.7E-8$) (Supplementary Note Table S42). While ILS regions, in general, show single-nucleotide polymorphism diversity patterns consistent with

balancing selection, it is noteworthy that both clustered and non-clustered ILS exons show a significant excess of polymorphic gene-disruptive events consistent with the action of relaxed or balancing

selection (Supplementary Note Fig. S36). An examination of these gene-rich clustered ILS regions shows a complex pattern of diverse ILS topologies consistent with deep coalescent operating across specific regions of the human genome as has been reported for major histocompatibility complex (Supplementary Note Fig. S66).”

Specific critiques / questions:

* Are there any new targets of positive selection? Are any of the specific variant classes (in particular those resolved to higher fidelity with the new assembly) driving positive selection signals?

Based on the referee’s suggestion, we performed a more systematic assessment of positive selection using both population genetic based approaches (i.e. selective sweeps) as well as more traditional approaches testing for excess of amino acid replacements (dN/dS). The latter is known to be less sensitive owing to the limited genetic distance and divergence among the apes (CSAC, 2004).

1. Tajima’s D and Sweepfinder2 analyses: For the population genetic approaches, we performed a genome-wide analysis for selective sweeps based on Illumina WGS mapped to the bonobo and chimpanzee long-read genome assemblies, namely: Mhudiblu_PPA_v0 and panTro6 (Supplementary Note Table S35). To identify potential sweeps, we applied two different site frequency spectrum (SFS)-based approaches, which search for an excess of rare variants. Briefly, Tajima’s D infers the difference between the estimates of $\Theta\pi$, the pairwise differences among individuals, and Θw , based on the number of segregating sites (Tajima, 1989, PMID: 2513255). By contrast, SweepFinder2 (DeGiorgio, et al. 2016, PMID: 27153702 and Nielsen, R. 2005, PMID: 16285858) (computes a composite likelihood ratio between the likelihood of the presence of a selective sweep at a given position and of the neutral model, modeled by the SFS of the tested sample. The latter method is more suitable for the detection of recent and stronger directional selection events.

Tajima’s D was calculated in genomic windows of 10kbp based on Illumina WGS data from 10 unrelated bonobos and 10 chimpanzees (Supplementary Note Table S35). We limited the analysis to biallelic variants with a QUAL score > 30 and where genomic data were available for at least 7 individuals for each species over that region of the genome. We limited the analysis to biallelic variants with a QUAL score > 30 and where genomic data were available for at least 7 individuals for each species over that region of the genome. All the analyses were performed with VCFtools 0.1.16. The Tajima’s D score distribution was similar between chimpanzee and bonobo (Supplementary Note Fig. S61). The Manhattan plot of the Tajima’s D values are shown in Supplementary Note Fig. S62.

Supplementary Note Figure S61. Density curves for the Tajima's D values inferred in 10 kbp genomic windows. For each species we extracted the top 100 windows, both for positive and negative values.

Supplementary Note Figure S62. An overview of Tajima's D (A-B) and SweepFinder2 analysis (C-

D) in bonobo and chimpanzee. The Manhattan plot shows Tajima's D (a & b) and Composite Likelihood Ratio (c & d) for Tajima's D and SweepFinder2 analysis, respectively.

We considered the top 100 genomic windows (negative Tajima's D) and intersected those with underlying genes. In bonobo, we found 64 discrete windows overlapping with 81 genes. We observe potential selective sweeps for *CADM2* (cell adhesion molecule 2, 2 windows D= -2.33 and -2.38, respectively)—a synaptic gene thought to be important in differentiation of synapses and behavioral responses (Stagi et al. 2010, PMID: 20368431) and *EIF4E3* (Eukaryotic Translation Initiation Factor 4E Family Member 3, D=-2.39141)—a gene whose protein product interacts with the 5' mRNA cap at the initial phase of the protein synthesis. The complementary analysis in chimpanzee showed signal for *FOXP2* (D= -2.3)—a transcription factor gene implicated in language development in humans but also shown to be under potential positive selection in chimpanzee (Nye et al. 2020, PMID: 33575612).

We also considered potential signatures of balancing selection (top 100 positive Tajima's D values) and intersected these with genes, retrieving 69 genes overlapping with 61 discrete windows (Supplementary Note Table S59). The genes included well-known examples of balancing selection such as MHC genes (*HLA-DPA1* and *HLA-DP2*, two window with D= 2.89 and 3.09) in addition to novel candidates such as *GPC5* (2 windows with D=3.1 and D=3.2, respectively) in bonobo and *KMT2C* (2 windows, D=2.16 and D=2.32), *MSH4* (2 windows, D=2.32 and D=2.15) and *OCA2* (D=2.13) genes in chimpanzee. Interestingly, *GPC5* (glypican 5) is a cell surface heparan sulfate proteoglycan important in cell growth and division while *OCA2* encodes the melanocyte P protein important in hair and skin pigmentation in humans and a subset of other primates (Supplementary Note Table S57-59).

Sweepfinder2 has the advantage over summary-based statistics like Tajima's D in that it controls from the local neutral mutation using the SFS and has the potential to identify more recent evidence of selection (Nielsen 2005, PMID: 16285858). This more advanced method has been shown to result in much higher sensitivity for detection of selective sweeps (Pavlidis et al., 2017, PMID: 28405579). We analyzed the genome using 10 kbp discrete windows for both chimpanzee and bonobo in the absence of recombination given the uncertainty of recombination rate differences and report the top 100 candidate regions (Supplementary Note Tables S61 and S62).

Supplementary Note Table S57. Summary and annotation of the top 100 negative Tajima's D values for bonobo

chr	start	end	Tajima's D	Z Tajima	rank	genes	annotation
chr8	3810000	3820000	-2.53	-2.56	1	CSMD1	CAT, RefSeq
chr5	156550000	156560000	-2.53	-2.55	2	NA	NA
chr5	79320000	79330000	-2.51	-2.52	3	NA	NA
chr3	126780000	126790000	-2.50	-2.51	4	TMCC1	RefSeq, CAT
chr4	47670000	47680000	-2.49	-2.50	5	SHROOM3, LOC112439638, RNU6-145P	CAT, RefSeq
chr13	69180000	69190000	-2.47	-2.47	6	NA	NA
chr5	80630000	80640000	-2.46	-2.45	7	NA	NA
chr10	118400000	118410000	-2.46	-2.45	8	ATE1	CAT, RefSeq
chr8	46010000	46020000	-2.45	-2.45	9	NA	NA

chr=chromosome, start= start of the genomic window, end=end of the genomic window, Tajima's D= D Value, Z Tajima= Z score of the D value, rank=Tajima's D rank, genes= name of the gene (if overlapping), Annotation= Annotation dataset considered

* This is a partial table excerpt; full table in Supplementary Note

Supplementary Note Table S58. Summary and annotation of the top 100 negative Tajima's D values for chimpanzee

chr	start	end	Tajima's D	Z Tajima	rank	genes
chr17	1900000	1910000	-2.25631	-2.555597305	70	SMG6, LOC104002749
chr20	8920000	8930000	-2.29477	-2.620705117	43	PLCB1
chr9	13250000	13260000	-2.27396	-2.585476475	52	MPDZ
chr5	14540000	14550000	-2.27377	-2.58515483	57	LOC107974634
chr14	16680000	16690000	-2.34063	-2.698340173	21	RALGAPA1
chr2A	18660000	18670000	-2.25631	-2.555597305	70	NA
chr14	18750000	18760000	-2.24322	-2.533437626	92	TTC6

chr=chromosome, start= start of the genomic window, end=end of the genomic window, Tajima's D= D Value, Z Tajima= Z score of the D value, rank=Tajima's D rank, genes= name of the gene (if overlapping)

* This is a partial table excerpt; full table in Supplementary Note

Supplementary Note Table S59. Summary and annotation of the top 100 positive Tajima's D values for bonobo

chr	start	end	Tajima's D	Z Tajima	rank	genes	annotation
chr14	60000	70000	2.97	4.84	62	FAM30A	CAT
chr1	260000	270000	2.95	4.81	68	ATAD3B, AL645728.2	RefSeq, CAT
chr7	10960000	10970000	3.04	4.92	46	LOC117980880	RefSeq
chr8	11590000	11600000	3.01	4.88	54	LOC112438728, RNA5SP253	RefSeq, CAT
chr22	13170000	13180000	2.89	4.73	88	C22H22orf42, Z83839.2, LOC117977432, C22orf42	RefSeq, CAT
chr17	15770000	15780000	3.14	5.06	26	NA	NA

chr=chromosome, start= start of the genomic window, end=end of the genomic window, Tajima's D= D Value, Z Tajima= Z score of the D value, rank=Tajima's D rank, genes= name of the gene (if overlapping), Annotation= Annotation dataset considered

* This is a partial table excerpt; full table in Supplementary Note

Supplementary Note Table S60. Summary and annotation of the top 100 positive Tajima's D values for chimpanzee

chr	start	end	Tajima's D	Z Tajima	rank	genes
chr22	180000	190000	2.17581	4.947409353	60	LRRC74B
chr8	220000	230000	2.30597	5.167753421	32	FBXO25
chr14	1850000	1860000	2.24444	5.063591078	45	NA
chr15	2500000	2510000	2.13051	4.870722305	75	OCA2
chr8	3330000	3340000	2.32861	5.206080016	29	CSMD1

chr=chromosome, start= start of the genomic window, end=end of the genomic window, Tajima's D= D Value, Z Tajima= Z score of the D value, rank=Tajima's D rank, genes= name of the gene (if overlapping)

* This is a partial table excerpt; full table in Supplementary Note

Supplementary Note Table S61. Summary and annotation of the top 100 SweepFinder2 SCLR values for bonobo

chr	start	end	SCLR-scores	alpha	rank	genes	annotation
chr8	89500757	89520757	65.916254	4.05E-05	1.00E+02	AC104211.1, AC117834.1, TRIQK	CAT, RefSeq
chr16	44468885	44488885	66.248473	7.97E-05	9.90E+01	NA	NA
chr5	132888717	132908717	67.294062	2.61E-05	9.80E+01	SPOCK1	RefSeq, CAT
chr3	56233799	56253799	67.379568	2.15E-05	9.70E+01	ERC2	RefSeq, CAT
chr4	130094997	130114997	67.860031	1.75E-05	9.60E+01	AC131956.3	CAT

chr2a	66410028	66430028	67.965253	4.24E-05	9.50E+01	MIR4778	CAT
chr8	93971159	93991159	72.623264	3.08E-05	9.40E+01	LOC117981463	RefSeq
chr8	89530760	89550760	72.963749	2.06E-05	9.30E+01	AC117834.1, TRIQK	CAT, RefSeq
chr18	22390222	22410222	78.042919	5.33E-05	9.20E+01	N/A	CAT

* This is a partial table excerpt; full table in Supplementary Note

Supplementary Note Table S62. Summary and annotation of the top 100 SweepFinder2 SCLR values for chimpanzee

chr	start	end	SCLR-scores	alpha	rank	genes	annotation
chr2b	543921	563921	38.94	0.0007074009	35	NA	NA
chr11	1565539	1585539	18.73	0.0004040051	8.60E+01	LOC112204781	RefSeq
chr2b	8054317	8074317	52.09	0.0001238779	2.20E+01	NA	NA
chr15	8404246	8424246	17.57	3.07E-06	9.30E+01	NA	NA
chr15	8414247	8434247	22.45	3.12E-06	7.50E+01	NA	NA
chr15	8424247	8444247	27.12	3.17E-06	5.30E+01	NA	NA
chr15	8434248	8454248	31.56	3.22E-06	39	NA	NA
chr15	8444249	8464249	35.76	3.28E-06	37	NA	NA

* This is a partial table excerpt; full table in Supplementary Note

For bonobo, we observed the strongest signal for chromosome 2b (75820999-76221031), within a region containing *DIRC1* (Disrupted In Renal Carcinoma 1) and *GULP1* (GULP PTB Domain Containing Engulfment Adaptor 1). *DIRC1* is expressed at low level in several tissues, while *GULP1* encodes an adapter protein involved in the phagocytosis of apoptotic cells and is ubiquitously expressed. High SweepFinder2 Composite Likelihood Ratio (SCLR) values were also observed for three windows (chr8: 46946928-47006932) within the gene *SNTG1*, encoding for the neuronal syntrophin protein associated with subcellular localization of proteins and neurotrophic signaling (Supplementary Note Fig. S62). On the same chromosome, putative selected regions are also observed in association with *PINX1* (PIN2/TERF1-interacting telomerase inhibitor 1) encoding a telomerase inhibitor and *SOX7* (SRV-related HMG-box 7) a transcription factor associated with embryonic development and in the determination of the cell fate, and the *TRIQQ* (triple QxxK/R motif-containing protein)—another gene potentially important in embryonic development.. For chimpanzee, we observed the strongest signal for the *TM4SF4* (Transmembrane 4 L Six Family Member 4) gene (chr3:147550781-147570782), encoding a transmembrane protein of the tetraspanin family thought to be important for cell proliferation especially in the gut (Supplementary Note Fig. S62).

2. dN/dS Positive Selection. We also searched for evidence of an excess of amino acid replacements in protein coding genes on the bonobo and hominid lineages. We applied a branch-site model of selection to all single-copy orthologs for 12,175 single copy gene orthologs (identified by Orthofinder¹⁹) based on available RefSeq annotations of human, chimpanzee, bonobo and gorilla. 2,322 single-copy orthologs showed some evidence of selection based on the aBSREL (adaptive branch-site random effects likelihood_ model implemented in the HyPhy software package with Bonferroni correction (false discovery rate < 0.05)²⁰. We then applied the PAML branch-site model to estimate selection of 2,322 single-copy orthologs, manually excluding alignment and isoform ambiguities. We identified 45 single-copy orthologs as significant using both the aBSREL model (HyPhy) and branch-site model (PAML). We classified genes into two categories: those with multiple amino acid replacements ($n \geq 5$) and the others likely resulting from a single mutational event ($n < 5$) (Supplementary Note Tables S63 and S64). Inspection of the latter suggested that multiple amino acid replacements changes most from a single frameshift event producing a cluster of amino-acid replacements (eg., *IFT80*) (Supplementary Note Fig. S63).

Supplementary Note Table S63. Summary of genes in the Pan lineage with excess amino acid replacement

	bonobo	chimp	pan	total
Multiple events (n>=5)	20	15	5	40
Single amino acid changes (n<5)	2	2	1	5
All	22	17	6	45

Supplementary Note Table S64. Candidate genes showing excess of amino acid replacement on specific branche

Lineage	Gene	HUMAN_refseq	BONOBO_refseq	CHIMP_refseq	GORILLA_refseq	ORANGUTAN_refseq	Alignment
bonobo	BAIAP2L1	NM_018842.5	XM_034963621.1	XM_016945059.2	XM_031006653.1	XM_002817703.4	Single amino acid changes
bonobo	SLC15A5	NM_001170798.1	XM_034935426.1	XM_001142606.4	XM_031000605.1	XM_002822990.3	
chimp	EXD3	NM_017820.5	XM_034929641.1	XM_024346011.1	XM_031014734.1	XM_024252353.1	
chimp	STRC	NM_153700.2	XM_034938649.1	XM_024353823.1	XM_031006451.1	XM_024232864.1	
pan	VSIG8	NM_001013661.1	XM_034938323.1	XM_016949587.2	XM_031011334.1	XM_002809931.2	
bonobo	C17orf99	NM_001163075.2	XM_034942992.1	XM_511708.6	XM_031010589.1	XM_002827888.1	
bonobo	C2CD4C	NM_001136263.2	XM_034950970.1	XM_016934474.2	XM_031006675.1	XM_024237544.1	
bonobo	CD6	NM_006725.5	XM_034932717.1	XM_001144310.3	XM_031016447.1	XM_024255879.1	
bonobo	COA6	NM_001206641.3	XM_034949257.1	XM_001152917.4	XM_004028612.3	XM_002809287.3	

* This is a partial table excerpt; full table in Supplementary Note

a

```

Ift80/1-772 1 MRLKSLLEPFHGLVSCVGMTEAEIYSCSDHDIWKWLLTSETQIVKLPDIYVDFHWFFKSLGVKQDQAEFVLTSSDGKPHLSKLGREKVSV 302
chmp2/1-772 1 MRLKSLLEPFHGLVSCVGMTEAEIYSCSDHDIWKWLLTSETQIVKLPDIYVDFHWFFKSLGVKQDQAEFVLTSSDGKPHLSKLGREKVSV 302
homo/1-772 1 MRLKSLLEPFHGLVSCVGMTEAEIYSCSDHDIWKWLLTSETQIVKLPDIYVDFHWFFKSLGVKQDQAEFVLTSSDGKPHLSKLGREKVSV 302
pan/1-772 1 MRLKSLLEPFHGLVSCVGMTEAEIYSCSDHDIWKWLLTSETQIVKLPDIYVDFHWFFKSLGVKQDQAEFVLTSSDGKPHLSKLGREKVSV 302
Ift80/1-777 1 MRLKSLLEPFHGLVSCVGMTEAEIYSCSDHDIWKWLLTSETQIVKLPDIYVDFHWFFKSLGVKQDQAEFVLTSSDGKPHLSKLGREKVSV 302
chmp2/1-777 1 MRLKSLLEPFHGLVSCVGMTEAEIYSCSDHDIWKWLLTSETQIVKLPDIYVDFHWFFKSLGVKQDQAEFVLTSSDGKPHLSKLGREKVSV 302
homo/1-777 1 MRLKSLLEPFHGLVSCVGMTEAEIYSCSDHDIWKWLLTSETQIVKLPDIYVDFHWFFKSLGVKQDQAEFVLTSSDGKPHLSKLGREKVSV 302
pan/1-777 1 MRLKSLLEPFHGLVSCVGMTEAEIYSCSDHDIWKWLLTSETQIVKLPDIYVDFHWFFKSLGVKQDQAEFVLTSSDGKPHLSKLGREKVSV 302
Ift80/1-782 388 KAHCGLACRNNYICTALVTVCEGDIKIKWSTCMLESTIADQCIPVYVAVGDPSEKVLVYACGLIKIYKIQRAKVLOWAHDGIIKLVVNSVNDLIL 294
chmp2/1-782 388 KAHCGLACRNNYICTALVTVCEGDIKIKWSTCMLESTIADQCIPVYVAVGDPSEKVLVYACGLIKIYKIQRAKVLOWAHDGIIKLVVNSVNDLIL 294
homo/1-782 388 KAHCGLACRNNYICTALVTVCEGDIKIKWSTCMLESTIADQCIPVYVAVGDPSEKVLVYACGLIKIYKIQRAKVLOWAHDGIIKLVVNSVNDLIL 294
pan/1-782 388 KAHCGLACRNNYICTALVTVCEGDIKIKWSTCMLESTIADQCIPVYVAVGDPSEKVLVYACGLIKIYKIQRAKVLOWAHDGIIKLVVNSVNDLIL 294
Ift80/1-797 205 SACDCCKYKWSYQSLVYVSPDHPHFTVAVAPDGLFAVCSFHTLRLCDKCGVYALEKPTGCFIYAWAIDCTQIACACNCHVVFAHVYEQHWK 306
chmp2/1-797 205 SACDCCKYKWSYQSLVYVSPDHPHFTVAVAPDGLFAVCSFHTLRLCDKCGVYALEKPTGCFIYAWAIDCTQIACACNCHVVFAHVYEQHWK 306
homo/1-797 205 SACDCCKYKWSYQSLVYVSPDHPHFTVAVAPDGLFAVCSFHTLRLCDKCGVYALEKPTGCFIYAWAIDCTQIACACNCHVVFAHVYEQHWK 306
pan/1-797 205 SACDCCKYKWSYQSLVYVSPDHPHFTVAVAPDGLFAVCSFHTLRLCDKCGVYALEKPTGCFIYAWAIDCTQIACACNCHVVFAHVYEQHWK 306
Ift80/1-802 302 NQVLTTRKAMRVKLVNDVLDLEFRDVKASLNVAHVVSTSLGCVYFSTKNWTFPIIFDLKCTVSLIQAERHFLVDCGSSFLYVYEGRFISSPK 408
chmp2/1-802 302 NQVLTTRKAMRVKLVNDVLDLEFRDVKASLNVAHVVSTSLGCVYFSTKNWTFPIIFDLKCTVSLIQAERHFLVDCGSSFLYVYEGRFISSPK 408
homo/1-802 302 NQVLTTRKAMRVKLVNDVLDLEFRDVKASLNVAHVVSTSLGCVYFSTKNWTFPIIFDLKCTVSLIQAERHFLVDCGSSFLYVYEGRFISSPK 408
pan/1-802 302 NQVLTTRKAMRVKLVNDVLDLEFRDVKASLNVAHVVSTSLGCVYFSTKNWTFPIIFDLKCTVSLIQAERHFLVDCGSSFLYVYEGRFISSPK 408
Ift80/1-817 408 PFCMRDIIAGDVYSLSNDTIARDKAEKIIIFLEASTCFPGLDGFISHNENILEIADQKCLTADRKAIFIDKNDLICTVSRKCFKEQIIKICTMWH 510
chmp2/1-817 408 PFCMRDIIAGDVYSLSNDTIARDKAEKIIIFLEASTCFPGLDGFISHNENILEIADQKCLTADRKAIFIDKNDLICTVSRKCFKEQIIKICTMWH 510
homo/1-817 408 PFCMRDIIAGDVYSLSNDTIARDKAEKIIIFLEASTCFPGLDGFISHNENILEIADQKCLTADRKAIFIDKNDLICTVSRKCFKEQIIKICTMWH 510
pan/1-817 408 PFCMRDIIAGDVYSLSNDTIARDKAEKIIIFLEASTCFPGLDGFISHNENILEIADQKCLTADRKAIFIDKNDLICTVSRKCFKEQIIKICTMWH 510
Ift80/1-832 511 TLAMNDTCNILECGDQTFIIVWYVNTVYVDRIPLTLYERDASEFKNFHVSFGVQDVIIRADCSLVHSISFFPAIHVEYSSSWEDAVLRCFVK 612
chmp2/1-832 511 TLAMNDTCNILECGDQTFIIVWYVNTVYVDRIPLTLYERDASEFKNFHVSFGVQDVIIRADCSLVHSISFFPAIHVEYSSSWEDAVLRCFVK 612
homo/1-832 511 TLAMNDTCNILECGDQTFIIVWYVNTVYVDRIPLTLYERDASEFKNFHVSFGVQDVIIRADCSLVHSISFFPAIHVEYSSSWEDAVLRCFVK 612
pan/1-832 511 TLAMNDTCNILECGDQTFIIVWYVNTVYVDRIPLTLYERDASEFKNFHVSFGVQDVIIRADCSLVHSISFFPAIHVEYSSSWEDAVLRCFVK 612
Ift80/1-847 613 ECTHWALCAAMAVANRDMTAEIAYAAIGEDKQYINSIKNLPSEKSMARIILFSGNIQAEIYLQALQVQAIDINILYVWRALILAVYETHVD 714
chmp2/1-847 613 ECTHWALCAAMAVANRDMTAEIAYAAIGEDKQYINSIKNLPSEKSMARIILFSGNIQAEIYLQALQVQAIDINILYVWRALILAVYETHVD 714
homo/1-847 613 ECTHWALCAAMAVANRDMTAEIAYAAIGEDKQYINSIKNLPSEKSMARIILFSGNIQAEIYLQALQVQAIDINILYVWRALILAVYETHVD 714
pan/1-847 613 ECTHWALCAAMAVANRDMTAEIAYAAIGEDKQYINSIKNLPSEKSMARIILFSGNIQAEIYLQALQVQAIDINILYVWRALILAVYETHVD 714
Ift80/1-862 715 LATRQFLFETCKQETNKRYLHYAAGLQDWEKIKAKIEMITEREQSSSIFRARV----- 772
chmp2/1-862 715 LATRQFLFETCKQETNKRYLHYAAGLQDWEKIKAKIEMITEREQSSSIFRARV----- 772
homo/1-862 715 LATRQFLFETCKQETNKRYLHYAAGLQDWEKIKAKIEMITEREQSSSIFRARV----- 772
pan/1-862 715 LATRQFLFETCKQETNKRYLHYAAGLQDWEKIKAKIEMITEREQSSSIFRARV----- 772

```

Ift80: selection in pan lineage

b

```

Aft80a_mus_0014899426.1-1-583 1 MSVDFDITDEEVHSHSIEKKTVEHICDCESSHWKIQVIGCLLVEICRFT 58
chmp2_mus_001142000.41-582 1 MSVDFDITDEEVHSHSIEKKTVEHICDCESSHWKIQVIGCLLVEICRFT 58
homo_mus_001170706.11-578 1 MSVDFDITDEEVHSHSIEKKTVEHICDCESSHWKIQVIGCLLVEICRFT 58
pan/1-577 1 MSVDFDITDEEVHSHSIEKKTVEHICDCESSHWKIQVIGCLLVEICRFT 58
Aft80a_mus_0014899426.1-1-583 37 FEVVCNMPFETIKLYVNCQAAIILNICTSLTPWVQVTDVYVGRNLY 112
chmp2_mus_001142000.41-582 37 FEVVCNMPFETIKLYVNCQAAIILNICTSLTPWVQVTDVYVGRNLY 112
homo_mus_001170706.11-578 37 FEVVCNMPFETIKLYVNCQAAIILNICTSLTPWVQVTDVYVGRNLY 112
pan/1-577 37 FEVVCNMPFETIKLYVNCQAAIILNICTSLTPWVQVTDVYVGRNLY 112
Aft80a_mus_0014899426.1-1-583 111 CLFLHFLCTALLSVVAFPLEDFLCTYHAVNRIPTQGRHLYVALLICLCVCC 388
chmp2_mus_001142000.41-582 111 CLFLHFLCTALLSVVAFPLEDFLCTYHAVNRIPTQGRHLYVALLICLCVCC 388
homo_mus_001170706.11-578 111 CLFLHFLCTALLSVVAFPLEDFLCTYHAVNRIPTQGRHLYVALLICLCVCC 388
pan/1-577 111 CLFLHFLCTALLSVVAFPLEDFLCTYHAVNRIPTQGRHLYVALLICLCVCC 388
Aft80a_mus_0014899426.1-1-583 169 IRAIVCFPCAFGLQYEGQKTFSTFNFWYVWLNINATVFLCISVIQHSQAWALV 224
chmp2_mus_001142000.41-582 169 VRAIVCFPCAFGLQYEGQKTFSTFNFWYVWLNINATVFLCISVIQHSQAWALV 224
homo_mus_001170706.11-578 169 VRAIVCFPCAFGLQYEGQKTFSTFNFWYVWLNINATVFLCISVIQHSQAWALV 224
pan/1-577 169 IRAIVCFPCAFGLQYEGQKTFSTFNFWYVWLNINATVFLCISVIQHSQAWALV 224
Aft80a_mus_0014899426.1-1-583 221 LIPFMSIMAVITLHMVYVNLVQSEKRSVLTSGVGLVSAIKTCHQYCHC 280
chmp2_mus_001142000.41-582 221 LIPFMSIMAVITLHMVYVNLVQSEKRSVLTSGVGLVSAIKTCHQYCHC 280
homo_mus_001170706.11-578 221 LIPFMSIMAVITLHMVYVNLVQSEKRSVLTSGVGLVSAIKTCHQYCHC 280
pan/1-577 221 LIPFMSIMAVITLHMVYVNLVQSEKRSVLTSGVGLVSAIKTCHQYCHC 280
Aft80a_mus_0014899426.1-1-583 281 TSIDHAKENGCYSSELWEDDTFFLTLPLFVQLYRMCIMQIFSYLQTMN 336
chmp2_mus_001142000.41-582 281 TSIDHAKENGCYSSELWEDDTFFLTLPLFVQLYRMCIMQIFSYLQTMN 336
homo_mus_001170706.11-578 281 TSIDHAKENGCYSSELWEDDTFFLTLPLFVQLYRMCIMQIFSYLQTMN 336
pan/1-577 281 TSIDHAKENGCYSSELWEDDTFFLTLPLFVQLYRMCIMQIFSYLQTMN 336
Aft80a_mus_0014899426.1-1-583 337 SNLNDGFLIPAVMNAISLPLLAFFPEVFTCLFPRSRVGFSTICIAGN 390
chmp2_mus_001142000.41-582 337 SNLNDGFLIPAVMNAISLPLLAFFPEVFTCLFPRSRVGFSTICIAGN 390
homo_mus_001170706.11-578 337 SNLNDGFLIPAVMNAISLPLLAFFPEVFTCLFPRSRVGFSTICIAGN 390
pan/1-577 337 SNLNDGFLIPAVMNAISLPLLAFFPEVFTCLFPRSRVGFSTICIAGN 390
Aft80a_mus_0014899426.1-1-583 389 FAALSVVMAACFEIHRKHFPVQDPVSKVLVTSMPFCVLIQYVILGVAETLVN 448
chmp2_mus_001142000.41-582 389 FAALSVVMAACFEIHRKHFPVQDPVSKVLVTSMPFCVLIQYVILGVAETLVN 448
homo_mus_001170706.11-578 389 FAALSVVMAACFEIHRKHFPVQDPVSKVLVTSMPFCVLIQYVILGVAETLVN 448
pan/1-577 389 FAALSVVMAACFEIHRKHFPVQDPVSKVLVTSMPFCVLIQYVILGVAETLVN 448
Aft80a_mus_0014899426.1-1-583 489 PALSVVYVYVYVSVYKSMNLTFLNFCCTGALLVGLVLSGDNFPNLTN 504
chmp2_mus_001142000.41-582 489 PALSVVYVYVYVSVYKSMNLTFLNFCCTGALLVGLVLSGDNFPNLTN 504
homo_mus_001170706.11-578 489 PALSVVYVYVYVSVYKSMNLTFLNFCCTGALLVGLVLSGDNFPNLTN 504
pan/1-577 489 PALSVVYVYVYVSVYKSMNLTFLNFCCTGALLVGLVLSGDNFPNLTN 504
Aft80a_mus_0014899426.1-1-583 505 GNLESFFFFLALLLNVLGFWVSQRVYCNHFNAGNIRGSLRETLHLESK 560
chmp2_mus_001142000.41-582 505 GNLESFFFFLALLLNVLGFWVSQRVYCNHFNAGNIRGSLRETLHLESK 560
homo_mus_001170706.11-578 505 GNLESFFFFLALLLNVLGFWVSQRVYCNHFNAGNIRGSLRETLHLESK 560
pan/1-577 505 GNLESFFFFLALLLNVLGFWVSQRVYCNHFNAGNIRGSLRETLHLESK 560
Aft80a_mus_0014899426.1-1-583 561 FYGIDFSSSDIWETALCNV 583
chmp2_mus_001142000.41-582 561 FYGIDFSSSDIWETALCNV 583
homo_mus_001170706.11-578 561 FYGIDFSSSDIWETALCNV 583
pan/1-577 561 FYGIDFSSSDIWETALCNV 583

```

Selection in bonobo: SLC15A5

Supplementary Note Figure S63. Candidate positive selection genes with excess amino acid replacement.

a, Multiple protein sequence alignment (top panel) shows signals of positive selection (PAML, bottom panel) in *IFT80* in the Pan lineage (chimpanzee and bonobo) resulting in a cluster of

amino acid replacements in the carboxy terminus (middle panel). *IFT80* is involved in the function of motile and sensory cilia and bone development. **b**, An example of a gene under positive selection (PAML, bottom panel) encoding the SLC15A5 protein with three amino acid replacement changes (top left) mapping to a transmembrane domain (top right). The gene is highly expressed in fat tissue and is associated with dicarboxylic aminoaciduria and hydranencephaly. 95% selection possibility from PAML model is shown in orange, 99% selection possibility from PAML model is shown in blue.

3. Comparison of candidate genes among positive selection tests. We compared the various tests for positively selected genes to determine if any genes were observed by more than one test (Supplementary Note Table S65 and Supplementary Note Fig. S64).

Supplementary Note Figure S64. Upset plot of multiple intersections among selection tests and ILS coordinates. The barplot shows the amount of overlapping base pairs resulting from the intersection of the tests/ILS scan indicated by the connecting points.

We were specifically interested in genes that showed evidence of positive selection by both negative Tajima's D values and SweepFinder2, focusing on the top 1% of signals (Supplementary Note Tables S65 and S66). Among the intersecting 50 windows for bonobo, we identified two genes related to lipid metabolism: 2-arachidonoyl-glycerol, an endocannabinoid (interacting with cannabinoid receptors) (*DAGLA* = chr11: 56979557 - 57046589, Tajima's D value=-1.99, SCLR= 13.5) and *ABHD2* = chr15: 67780452-67891154. Tajima's D value=-2.29, SCLR= 8.54) Interestingly, we also identified signatures of positive selection for *CAMK2D* gene (chr4: 106083972- 106103972, Tajima's D = -2.11, SCLR = 6.99), an upstream regulator of *DAGLA* activity suggesting that the pathway may be under selection in bonobo. We also identified a putative selected window within the *CEP164* (*chr11: 112185192- 112205192, Tajima's D=-2.02, SCLR= 15.7*) gene, involved in microtubule organization. Within the chimpanzee lineage, we found both signals of selection corresponding to the *GRIA4* (chr11: 101388489- 101694639, Tajima's D= -1.92, SCLR=3.84) which encodes for the glutamate receptor and found evidence of selection in genes related to chromatin structure: *PHF2* (chr9:65812964-65914169, Tajima's D=-2.07, SCLR= 9.64) and *HIST1H1C* (chr6:19089567-19090347, Tajima's D= -2.36, SCLR= 5.24).

Supplementary Note Table S65. Summary and annotation of the bonobo putative selected regions (in the top 1%) in Tajima's D and SweepFinder2

chr	D_start	D_end	TajimaD	rank_D	Sfstart	Sfend	CLR-score	alpha	rank_SF	gene
chr15	67840000	67850000	-2.29	164	67827363	67847363	8.56	0.0002	834	ABHD2
chr10	70940000	70950000	-1.98	2067	70937105	70957105	25.71	0.0006	229	ADK
chr10	70960000	70970000	-2.23	305	70947106	70967106	6.86	0.0001	1131	ADK
chr1	3440000	3450000	-2.30	151	3446220	3466220	11.96	0.0004	545	AJAP1
chr15	58250000	58260000	-2.03	1553	58246187	58266187	29.05	0.0000	209	ANKRD34C
chr15	58260000	58270000	-2.06	1204	58246187	58266187	29.05	0.0000	209	ANKRD34C

chr: chromosome; Dstart=Tajima's D window start, Dend=Tajima's D window end, TajimaD= Tajima's D value, rank D: rank for Tajima's D value; Sfstart: start of the SweepFinder2 region, Sfend: end of the SweepFinder2 region, SCLR-score, alpha=alpha score, SF rank=rank for SweepFinder2 SCLR; gene=name of the gene (NA indicates that the region do not overlap any known gene).

* This is a partial table excerpt; full table in Supplementary Note

Supplementary Note Table S66. Summary and annotation of the chimpanzee putative selected regions (in the top 1%) in Tajima's D and SweepFinder2

chr	D_start	D_end	TajimaD	rank_D	Sfstart	Sfend	SCLR-score	alpha	rank_SF	gene
chr1	45070000	45080000	-1.87	2467	45071243	45091243	5.68	0.0009	492	RAD54L
chr1	51210000	51220000	-1.93	1615	51201406	51221406	8.46	0.0001	233	PRPF38A
chr1	128770000	128780000	-1.88	2238	128753474	128773474	4.55	0.0004	771	CRTC2
chr1	128770000	128780000	-1.88	2238	128753474	128773474	4.55	0.0004	771	SLC39A1
chr1	128770000	128780000	-1.88	2238	128753474	128773474	4.55	0.0004	771	CREB3L4
chr10	51590000	51600000	-2.04	730	51582330	51602330	7.01	0.0026	335	NA

chr: chromosome; Dstart=Tajima's D window start, Dend=Tajima's D window end, TajimaD= Tajima's D value, rank D: rank for Tajima's D value; Sfstart: start of the SweepFinder2 region, Sfend: end of the SweepFinder2 region, SCLR-score, alpha=alpha score, SF rank=rank for SweepFinder2 SCLR; gene=name of the gene (NA indicates that the region do not overlap any known gene).

Based on this intersection set of genes (n=21), we searched for gene ontology and gene expression enrichment. For gene ontology enrichment analysis, we applied enrichr²¹, testing our gene set against five different annotations libraries (KEGG_2019_Human, GO_Molecular_Function_2018 GO_Biological_Process_2018, GO_Cellular_Component_2018, and Panther_2016 Mi, et al. 2021 PMID: 33290554) as described for expansions and contractions (see section 6.4.1). Acylglycerol lipase activity (GO Molecular Function 2018), Lipase activity (GO Molecular Function 2018) and 2-arachidonoylglycerol biosynthesis²² were significantly enriched GO categories (Supplementary Note Table S67). By contrast, no GO category was enriched for positively selected genes (n=32) in the chimpanzee.

Supplementary Note Table S67. GO enrichment analysis of putative selected genes in bonobo

	Overlap	P-value	Adjusted P-value	Odds Ratio	Combined Score	Genes	Gene_set
acylglycerol lipase activity (GO:0047372)	2/11	5.7E-05	2.1E-03	2.3E+02	2280.8	DAGLA; ABHD2	GO_Molecular_Function_2018
lipase activity (GO:0016298)	2/43	9.2E-04	1.7E-02	5.1E+01	357.7	DAGLA; ABHD2	GO_Molecular_Function_2018

2-arachidonoylglycerol biosynthesis Homo sapiens P05726	1/6	6.3E-03	1.9E-02	2.0E+02	1012.6	DAGLA	Panther_2016
---	-----	---------	---------	---------	--------	-------	--------------

Gene classes enriched; p-value: p-value based on Fisher's test; Overlap: number of genes in the tested set overlapping with the gene category; Adjusted p-value: Benjamini-Hochberg adjusted p-value; Genes: Name of the genes in the overlap; Gene set: Gene ontology class.

We added these new analyses to the Supplementary Note and added a paragraph to the Gene annotation results section. While interesting, we feel that additional experimental work is required to validate the functional significance of such signals.

“The availability of a more complete gene annotation in bonobo as well as other apes such as chimpanzee allows for more comprehensive analyses of positive selection. We performed a genome-wide analysis to identify genes showing an excess of amino acid replacements in the chimpanzee lineage as well as potential selective sweeps based on analysis of sequencing data from 20 bonobo and chimpanzee population samples to the new references. The latter analyses identified numerous candidate genes for selective sweeps in bonobo (*DIRC2*, *GULP1*, *ERC2*; Supplementary Note Tables S57, S59, S61 and S65) and chimpanzee (*KIAA040*, *TM4SF4*, *FOXP2*; Supplementary Note Tables S58, S60, S62 and S66). In bonobo, we observed an enrichment of genes subject to selective sweeps associated with lipid metabolism (e.g., *DAGLA*, *CAMK2D* and *ABHD2*). Similarly, candidate regions and genes underlying sites of balancing selection were identified based on a Tajima’s D analysis. While additional investigations will be required to assess the functional significance of these in each lineage, and with larger sample sizes, most of these candidate genes are novel.”

* What is the landscape of ILS beyond coding regions? How often do these "ILS clusters" cross gene boundaries? It may be interesting to intersect some of these non-coding patterns with human regulatory annotations (eg ENCODE, Hi-C) or disease annotations (GWAS).

This is an interesting suggestion. Since bonobo noncoding regulatory DNA annotations are not available, we intersected both clustered and non-clustered ILS segments with both genes (RefSeq) and ENCODE (V3) regulatory regions based on human annotation.

Using human gene annotation (RefSeq GRCh38), we classify 1.37 Gbp (45.2%) of the genome as intragenic and 1.66 Gbp (54.8%) as intergenic. With respect to chimpanzee/human ILS, we find that 19,607 clustered H-B (total: 29,691) and 19,930 clustered H-C (total: 30,056) correspond to intergenic regions, respectively. Based on a null distribution (randomly choose 30,000 segments (500bp) compute the mean 100 times) (mean=17384.9), we find that both clustered H-B(19,607 (66%), empirical p=0) /H-C (19930 (66%), empirical p=0) ILS are more likely to be located in the intergenic regions.

With respect to noncoding regulatory DNA, we considered the 926,536 annotated regulatory elements from ENCODE (V3) database and found that 4,070 clustered H-B and 4,083 clustered H-C are intersected with regulatory elements, respectively. Similarly, we find 13,728 nonclustered H-B and 13,772 nonclustered H-C intersect with regulatory elements, respectively. To ask whether the clustered H-C/H-B are more/less likely intersected with the regulatory elements with respect to the genome-wide or nonclustered H-C/H-B, we randomly chose 1,000 segments from each type (clustered H-C/H-B, non clustered H-C/H-B, and genome-wide), and calculated the number of intersection between the 1,000 segments and regulatory elements. We repeated this process 100 times and compared the distributions. We found that clustered H-B ($p < 2.2e-16$)/H-C ($p < 2.2e-16$) are less likely to intersect with the regulatory elements with respect to genome-wide or the nonclustered H-B/H-C. Yet, interestingly, we found that nonclustered H-B ($p = 0.00005$)/H-C ($p = 0.001$) are more likely to intersect with the regulatory elements with respect to genome-wide (Supplementary Note Fig. S32b and c).

Supplementary Note Figure S32. b, Clustered H-B/H-C ILS are less likely intersected with regulatory elements (ENCODE V3) with respect to genome-wide or non-clustered H-B/H-C. **c**, (Non)clustered H- B/H-C ILS less likely intersected with exons (RefSeq) with respect to genome-wide or non-clustered H- B/H-C.

With respect to exons, we repeated the same process using RefSeq definitions. As we expected, the H-B/H-C are less likely to intersect with exons (RefSeqs) no matter whether they are clustered or not. Interestingly, clustered H-B/H-C are less likely to intersect with exons with respect to the nonclustered H-B/H-C.

We added this analysis to the Supplementary Note and as statement to the main text summarizing these findings:

“and specifically calcium transport ($p = 3.7E-8$) (Supplementary Note Table S42). While ILS regions, in general, show single-nucleotide polymorphism diversity patterns consistent with balancing selection, it is noteworthy that both clustered and non-clustered ILS exons show a significant excess of polymorphic gene-disruptive events consistent with the action of relaxed or balancing selection (Supplementary Note Fig. S36). An examination of these gene-rich clustered ILS regions shows a complex pattern of diverse ILS topologies consistent with deep coalescent operating across specific regions of the human genome as has been reported for major histocompatibility complex (Supplementary Note Fig. S66).”

* Are there regions statistically depleted in ILS suggesting selective sweeps?

To address this question, we also searched for regions significantly depleted for ILS (ILS

deserts) by calculating the inter-ILS distance and selecting regions within the lowest 1% of that

distribution. We identified 892 and 909 ILS deserts (H-B and H-C respectively). Next we estimated diversity (π) in both chimpanzee and bonobo comparing it to the genome-wide average. We observed that both H-B and H-C ILS deserts show reduced genetic diversity although are not significantly different from each other. These results are consistent with these regions being targets of selective sweeps or background selection regions in the Pan lineage (Supplementary Note Figure S32d). Thus, we intersected ILS deserts with regions identified by Sweepfinder2 (above). We found 40 ($p=0.29$) and 41 ($p=0.23$) bonobo selective sweeps regions intersected with H-B and H-C desert regions, respectively; while 55 ($p=0.17$) and 45 ($p=0.61$) chimpanzee selective sweeps regions intersected with H-B and H-C deserts, respectively. These data suggest that ILS deserts are not more likely to be associated with selective sweeps in bonobo and chimpanzee. We have added this analysis to the Supplementary Note.

Supplementary Note Figure S32. d, ILS deserts and reduced genetic diversity. Distribution of ILS deserts was defined as the top 1% of ILS deserts (top panel) for H-B (red) and H-C ILS (blue) regions. Genetic diversity (π) is compared for bonobo (left) and chimpanzee (right panel) for H-B and H-C deserts to a randomly simulated set and the genome wide average based on autosomal regions.

* What is the role of SVs and repeat elements in ILS? Could this be used to say something about selection on acting on these variant classes?

We did not highlight this in the main text but we investigated this previously. We found that fixed insertions are enriched (1.46-fold higher P-value < 0.001 ; chi-square) but fixed deletions are significantly reduced 0.34-fold lower (P-value < 0.001) in ILS regions. Based on the referee's

suggestion, we further investigated the two major common repeat classes and found that both Alu (1.065-fold, $P < 0.001$) and L1 elements (1.33-fold, $P < 0.001$) are significantly higher within regions of ILS. These data are consistent with ILS regions in general being under more relaxed selection (Supplementary Table 22).

Supplementary Table 22. ILS intersections with SVs and common repeats.

	Total	Intersect with 500 bp ILS	Intersect with 500 bp ILS	Intersect with Inversion
Insertion (Fixed)	3604	110	157	376
Deletion (Fixed)	1965	8	26	195
Insertion (polymorphic)	12182	268	283	982
Deletion (polymorphic)	5117	89	126	531
Alu	1108093	30007	29861	NA
L1	963794	32152	32797	NA

We added a statement to the main text summarizing these findings:

“and specifically calcium transport ($p = 3.7E-8$) (Supplementary Note Table S42). While ILS regions, in general, show single-nucleotide polymorphism diversity patterns consistent with balancing selection, it is noteworthy that both clustered and non-clustered ILS exons show a significant excess of polymorphic gene-disruptive events consistent with the action of relaxed or balancing selection (Supplementary Note Fig. S36). An examination of these gene-rich clustered ILS regions shows a complex pattern of diverse ILS topologies consistent with deep coalescent operating across specific regions of the human genome as has been reported for major histocompatibility complex (Supplementary Note Fig. S66).”

* The EIF3A results are striking, but left as an isolated observation. Can the authors expand on this finding? For example, can something be said about the locus architecture, sequence features, or dynamics of this and other SD's, or their regional distribution now that they have been placed into scaffolds. Are there more that are as high level as this one?

We performed a much more extensive analysis of this gene family and have developed it into a more complete story which we now highlight with a revised Figure 3 in the main. Please note that there was a typographical error in the original description (lines 273-276) which reported the gene family as *EIF3A* as opposed to the correct name *EIF4A3*. We corrected this throughout the main and supplement.

We targeted this region for complete assembly using HiFi sequence data and were able to reconstruct the complete locus in bonobo, chimpanzee, gorilla, and orangutan identifying five full-length gene copies (262 kbp total length) in chimpanzee and six copies in bonobo (310 kbp in bonobo). In both chimpanzee lineages, the gene families are organized head-to-tail in direct orientation. (Supplementary Note Fig. S41).

Supplementary Note Figure S41. Recent expansion of EIF4A3 genes in the Pan lineage. Contigs that encompass *EIF4A3* expansions and 100 kbp of the flanking regions were assembled using bonobo and chimpanzee PacBio HiFi data. A 12 kbp segment of genomic sequence representative of human *EIF4A3* is mapped onto the assembled contigs. Six tandem copies of *EIF4A3* spanning 310 kbp in bonobo and five tandem copies spanning 262 kbp in chimpanzee are recovered organized in a head-to-tail configuration. Gray, black, and striped arrows show synteny across the ape genomes.

We used the high quality sequence to generate an MSA and then construct a phylogeny estimating that the initial *EIF4A3* gene duplication occurred in the ancestral lineage of chimpanzee and bonobo approximately 2.9 mya. The locus subsequently expanded before and after chimpanzee and bonobo speciation to create the multiple copies (Supplementary Note Fig. S42).

Supplementary Note Figure S42. EIF4A3 primate phylogeny. The *EIF4A3* duplication results from multiple expansions before and after chimpanzees–bonobo speciation. A phylogenetic tree was constructed from 22 kbp noncoding sequence of *EIF4A3* paralogs using Bayesian phylogenetic inference. This analysis is conducted using BEAST2 software. Bolded numbers on each major node denote estimated divergence time. The blue error bar on each node indicates 95% confidence interval of the age estimation. Bayesian posterior probabilities are reported using asterisks for nodes with posterior probability >99%.

Sequence analysis using GeneConv suggests independent gene conversion events in each lineage. A subset of these events correspond to a set of Pan-specific amino-acid changes in the basic ancestral structure of the single ancestral copy that are now common to only chimpanzees and humans (Supplementary Note Figure S43).

Supplementary Note Figure S43. Gene conversion of *EIF4A3*. Paralogs are expressed and show evidence of gene conversion in both bonobo and chimpanzee lineages. Analysis of bonobo Iso-Seq data confirms that five of the six *EIF4A3* copies are expressed and maintain an open reading frame (heatmap indicates the number of Iso-Seq transcripts supporting each copy; minimap2 -ax splice -G 3000 -f 1000 --sam-hit-only --secondary=no --eqx -K 100M -t 20 --cs -2 | samtools view -F 260). GENECONV software shows significant signals ($p \leq 0.05$ after multiple test correction) of gene conversion for 16/67 kbp of the paralogous locus (gray bars) (MSA was performed using MAFFT version 7.453 (command: mafft --adjustdirection [input.fasta] > [output.msa_fasta]; GENECONV version 1.81a). A subset of gene conversion events overlap with sites of amino-acid specific to the Pan lineage. Triangles indicate the sites of amino acid change in each of the primate genomes compared to GRCh38. Different colors mark different changes: purple marks phenylalanine to leucine; yellow marks arginine to cysteine; red marks serine to arginine; teal marks tyrosine to serine. Same phylogenetic tree from **Figure 3c** is reshaped to show the inferred evolutionary relationships among the paralogs. Nodes with >99% Bayesian posterior probabilities are indicated by asterisks; otherwise the actual number is shown.

As an aside, we investigated the copy number of *EIF4A3* in other mammalian lineages. Specifically, we mapped (blat -stepSize=5 -minScore=1000 -repMatch=2253 -minScore=20 -minIdentity=0) human *EIF4A3* genomic sequence onto genome assemblies of mouse lemur (MicMur2), mouse (mm39), opossum (monDom5), cow (bosTau9), and dog (canFam5). In all other lineages we were able to identify only one copy of the *EIF4A3* gene from each of the species suggesting that the expansion is specific to the *Pan* lineage.

We revised Figure 3 in the main, added the gene conversion analysis to the supplement, and elaborated in more detail in the main text on these new findings regarding the *EIF4A3* gene family.

“Among the Pan-specific Eukaryotic Translation Initiation Factor 4 Subunit A3 (*EIF4A3*) gene family, there is evidence that five out of the six paralogs are expressed and encode a full-length open reading frame (ORF; Fig. 3). Based on our assembly of the locus in both chimpanzee and bonobo, we show that the initial *EIF4A3* gene duplication occurred in the ancestral lineage approximately 2.9 mya. It then subsequently expanded and experienced gene conversion events independently in the chimpanzee and bonobo lineages creating five and six copies of the *EIF4A3* gene family, respectively. In both lineages, the gene families are organized head-to-tail in direct orientation. Interestingly,

some of the gene conversion signals correspond to a set of specific amino-acid changes in the basic ancestral structure that are now common to only chimpanzee and bonobo.

Each of the chimpanzee specific copies carries an 18 bp VNTR motif in the 5' UTR, which frequently differs among the copies."

Figure 3. *EIF4A3* gene family expansion and sequence resolution. **a**, A comparison of *EIF4A3* copy number among apes based on a sequence read-depth analysis confirms a variable copy number expansion in the bonobo and chimpanzee lineage (9-33 diploid copies). This recent duplication was not fully resolved initially in the bonobo reference genome (Mhudiblu_PPA_v0) because high identity duplicated sequences were collapsed. **b**, Contigs which encompass *EIF4A3* expansions and 100 kbp of the flanking regions were assembled using bonobo and chimpanzee PacBio HiFi data. The 12 kbp genomic sequence of human *EIF4A3* is mapped onto the assembled contigs. 6 tandem copies of the *EIF4A3* gene spanning 310 kbp in bonobo and 5 tandem copies spanning 262 kbp in chimpanzee are recovered. Schematics show structural differences of *EIF4A3* in primate genomes. Grey, black, and striped arrows show different alignment blocks across the samples. A solid line connecting alignment blocks indicates insertion event. **c**, Multiple sequence alignment shows *EIF4A3* amino acid differences between the human, the Mhudiblu_PPA assembled paralogs, chimpanzee assembled paralogs and other great apes. A polymorphic 18 bp motif VNTR is located at the 5' UTR of non-human primate *EIF4A3* gene and accounts for most of the differences between different isoforms. A phylogenetic tree is built from neutral sequences of *EIF4A3* paralogs using Bayesian phylogenetic inference. This analysis is conducted using BEAST2 software. Numbers on each major node denotes estimated divergent time. Blue error bar on each node indicates 95% confidence interval of the age estimation. Bayesian posterior probabilities are reported using asterisks for nodes with posterior probability >99%. **d**, Bonobo Iso-Seq full-length transcript reads map with higher identity to four of the paralogs when compared to Mhudiblu_PPA_v0. **e**, FISH on metaphase chromosomes and interphase nuclei with human fosmid probe WI2-3271P14 confirms an *EIF4A3* subtelomeric expansion of chromosome 17 in bonobo and chimpanzee relative to human, gorilla and orangutan.

* More broadly, what new insight does the increased resolution of SD's in the bonobo genome give into the dynamics of SD's and gene family expansions in great apes beyond the 2009

Marques-Bonet et al paper - in particular these very high amplitude SD's. Since these are resolved at breakpoint resolution, there should be opportunities to illustrate how some of these

loci are evolving in the great ape lineages. For example, EIF3A is duplicated in both chimpanzee and bonobo to different numbers of copies - when did the individual duplications occur relative to speciation. Also, is there anything special about EIF3A that would select for this - for example, does this locus undergo SD in other mammalian lineages?

In addition to this specific example we performed a genome-wide analysis of gene expansions in both the bonobo and chimpanzee lineage. First, we identified copy number expansions and contractions in the *Pan* lineage and classified these as bonobo-specific, chimpanzee-specific, or shared (Pan-specific), compared to other hominids. This classification was based initially on short-read Illumina WGS mapping (WSSD) from 27 ape genomes (Supplementary Note Table S35) to the human reference to generate an assembly-independent assessment of copy number in order to focus on species-specific expansions as opposed to polymorphisms. Species-specific or Pan-specific events were subsequently confirmed orthogonally by read-depth analysis using the long reads and analysis of whole-genome and targeted long-read assemblies (HiFi and CLR) requiring a diploid copy number difference of at least 2. We focused on regions likely to contain genes based on Isoseq annotation or by liftoff analyses (GCA_009914755.2, <https://github.com/nanopore-wgs-consortium/CHM13>). Liftoff v1.4.2 was performed with the parameters ‘-flank 0.1 -sc 0.85 -copies’ against each target genome using GRCh38 GENCODE v35 annotations as the source, in order to count the number of duplicated loci with corresponding transcript support for each gene in each assembly. To estimate number of assembled copies of each gene independent of Liftoff gene annotations, we aligned 2kbp chunks of each assembly to GRCh38 with MashMap v2.0 (Jain et al. Bioinformatics 2018), and merged adjacent alignments, requiring at least 6.5 kbp of contiguous sequence at 95% sequence identity. The number of assembled macaque loci corresponding to each GENCODE gene model was summarized with BEDTools. Among protein-coding gene family expansions (GRCh38 GENCODE v35), we identified 42 bonobo-specific, 12 chimpanzee-specific, and 142 shared *Pan* expansion candidates. Similarly, we identified 13 bonobo-specific, 6 chimpanzee-specific, and 56 shared *Pan* contraction candidates. For each bonobo gene duplication resolved by long-read assembly, we aligned Iso-Seq data and assessed the number of transcripts to identify predominant isoforms and potential changes in the gene structure (Supplementary Tables 11 and 12).

Supplementary Table 11. Full table of candidates: expansions

gene_ID	lineage	WSSD		Assembly CN (whole genome align Read depth)										Assembly CN (Liftoff)					Isoseq	Gene (hg38)		start	end	gene_type
		Ppa	Ptr	Hsa	Ggo	Pab	HiFi CN	HiFi CN50	v0 CN	v0 CN50	HiFi CN	CLR CN	Ppa_HiFi	Ppa_HiFi	Hsa	Ggo	Pab	Ppa_CLR		Ppa_SDA	Loci			
GCLGABL10	Ptr	13.6	16.7	15.2	12.0	13.5	6.5	7	7.5	8	4.2	3.0	3	4	2	1	1	2	1	chr15	82339993	82349475	protein_coding	
PKSS27	Ptr	1.0	1.9	1.1	1.0	0.6	1.8	2	0.9	1	6.3	2.2	3	1	1	1	1	2	1	chr19	685546	695498	protein_coding	
CCDC74A	Ptr	2.1	3.2	2.1	2.0	1.9	3.0	3	3.6	4	1.4	1.3	2	1	1	1	1	2	1	chr2	131527675	131533666	protein_coding	
FOXD4L3	Ptr	2.7	8.9	6.9	3.6	0.2	8.4	10	8.8	9	10.9	10.2	2	5	2	2	1	2	1	chr9	68302867	68305084	protein_coding	
...																								

* This is a partial table excerpt; full table in Supplementary Note

As a final validation and to confirm their organization within the bonobo/chimpanzee genome, we selected five gene family expansions (*CLN3*, *EIF3C*, *RGL4*, *IGLV6-57*, *SPDYE16*) and four gene loss events (*IGFL1*, *SAMD9* (described in the original submission), *TRAV4*, *CDK11A*) for experimental validation by FISH (Supplementary Note Tables S48 and S49). Fosmid probes (n=9) corresponding to human genomic data were isolated and hybridized against human, bonobo, chimpanzee, gorilla and orangutan chromosomal metaphase spreads and interphase nuclei. Every hybridization was performed as a co-hybridization experiment combining one clone for expansion and one clone for contraction to be sure that the absence of signals expected for the contraction was due to a real absence of signals and not a technical artefact (Supplementary Note Figs. S39 and S40). This analysis confirmed all genome predictions

(Supplementary Note Tables S49 and Supplementary Note Figs. S20 and S21) providing the most comprehensive resource of chimpanzee and bonobo gene family expansions. It is noteworthy that three out of four tested gene expansions show patterns of intrachromosomal interspersion and these are found adjacent to “core duplicons” (eg. *NPIP* and *GUSBP*) which have been predicted to mediate the formation of interspersed segmental duplications in humans.

Supplementary Note Table S48. Gene functions in expanded and contracted genomic regions

Class	Gene	Description	Function	Phenotype	Notes
Expansion	CLN3	CLN3 Lysosomal/Endosomal Transmembrane Protein, Battenin	This gene encodes a protein that is involved in lysosomal function.	LOF causes neurodegenerative diseases commonly known as Batten disease or collectively known as neuronal ceroid lipofuscinoses (NCLs).	adjacent to NPIP
Expansion	EIF3C	Eukaryotic Translation Initiation Factor 3 Subunit C	EIF3C (Eukaryotic Translation Initiation Factor 3 Subunit C) is a Protein Coding gene.	Diseases associated with EIF3C include Colon Squamous Cell Carcinoma.	adjacent to NPIP
Expansion	RGL4	Ral Guanine Nucleotide Dissociation Stimulator Like 4	This oncogene encodes a protein similar to guanine nucleotide exchange factor Ral guanine dissociation stimulator. The encoded protein can activate several pathways, including the Ras-Raf-MEK-ERK cascade.	Increased expression of this gene leads to translocation of the encoded protein to the cell membrane. RGL4 expression is significantly associated with a variety of tumor-infiltrating immune cells (TICs), particularly memory B cells, CD8+T cells and neutrophils.	adjacent to GUSBP core duplicon
Expansion	IGLV6-57	Immunoglobulin Lambda Variable 6-57	Protein Coding gene.	no phenotype associated	adjacent to a deletion
Expansion	SPDYE16	Speedy/RINGO Cell Cycle Regulator Family Member E16	Protein Coding gene. Among its related pathways are Oocyte meiosis.	no phenotype associated	high-copy duplicon
Contraction	IGFL1	IGF Like Family Member 1	The protein encoded by this gene is a member of the insulin-like growth factor family of signaling molecules. The encoded protein is synthesized as a precursor protein and is proteolytically cleaved to form a secreted mature peptide. The mature peptide binds to a receptor, which in mouse was found on the cell surface of T cells.	Increased expression of this gene may be linked to psoriasis.	
Contraction	SAMD9	Sterile Alpha Motif Domain Containing 9	This gene encodes a sterile alpha motif domain-containing protein. The encoded protein localizes to the cytoplasm and may play a role in regulating cell proliferation and apoptosis.	Mutations in this gene are the cause of normophosphatemic familial tumoral calcinosis (autosomal recessive)	
Contraction	TRAV4	T Cell Receptor Alpha Variable 4	In a single cell, the T cell receptor loci are rearranged and expressed in the order delta, gamma, beta, and alpha.	no phenotype associated	11 kbp deletion
Contraction	CDK11A	Cyclin Dependent Kinase 11A	This gene encodes a member of the serine/threonine protein kinase family. Members of this kinase family are known to be essential for eukaryotic cell cycle control.	These two genes are frequently deleted or altered in neuroblastoma.	

Supplementary Note Table S49. FISH results for expansions and contractions of bonobo and/or Pan genomes

Class	Gene	Fosmid Clones	Coords (hg38)	Heat map predictions				FISH Results										
				HSA	PPA	PTR	GGO	PPY	HSA	PPA	PTR	GGO	PPY					
Expansion	CLN3	170215 ABC9_3_2_000041281300_M15	chr16:28479201-28516032	S	D	D	S	S	16p	Single	XXlp	Dup	XVlp	Dup	XVlp	Single	XVlp	Single
Expansion	EIF3C	172243 ABC9_3_5_000044010100_H14	chr16:28687256-28729352	D	D	D	S	S	16p	Dup#	XXlp	Dup	XVlp	Dup	XVlp	Single	XVlp	Single
Expansion	RGL4	171515 ABC9_3_5_000046184500_C13	chr22:23675621-23714508	S	D	D	S	S	1p, 9q, 22q	Dup\$	lp (weak), IXq (weak), XXIIq	Dup	lp, later, VIIpter, IXq, XIIq	Dup	lp, IXq, XXIIq	Dup\$	XIIq	Single
Expansion	IGLV6-57	ABC8-412020005	chr22:22178597-22214773	S	S/D	S	S	S	22q	Single	XXIIq	Single	XXIIq	Single	XXIIq	Single	XXIIq	Acrocentric chrs
Expansion	SPDYE16	171515 ABC9_3_5_000043959400_P22	chr7:76507030-76545218	S/D	D	D	S/D	S/D	7q	Dup	VIIq	Dup	VIIq	Dup	VIIq	Dup	VIIq	Dup
Contraction	IGFL1	170215 ABC9_3_2_000043862300_J24	chr19:46195756-46232256	S	del	del	S	S	19q	Single	No signal	del	IXXq	Single	IXXq	Single	IXXq	Single
Contraction	SAMD9	ABC8-41156300P24	chr7:83082459-83118602	S	del	S	S	S	7q	Single	No signal	del	VIIq	Single	VIIq	Single	VIIq	Single
Contraction	TRAV4	ABC8-42078300A3	chr14:21716253-21749608	S	S/del	S	S	S	14q	Single	XIVq (weak)	del	XIVq	Single	XIVq	Single	XIVq (weak)	Single
Contraction	CDK11A	ABC8-41133000L6	chr1:1700902-1734122	D	del	S/del	D	S	1p	Dup#	No signal	del	No signal	del	lp	Dup	lp	Single

Polymorphic duplication tested in three human (HG00733, GM12813 and GM24385)

§ FISH results different from predictions In bold highly duplicated pattern signals

Supplementary Note Figure S39. *Pan*-specific duplication of *CLN3* locus, and bonobo-specific deletion of *IGFL1*. HiFi read depth and WSSD of bonobo, chimpanzee, orangutan, gorilla, and human individuals relative to GRCh38 detect these events (above), which are validated by interphase FISH of each species using fosmid clones spanning the region (below).

Supplementary Note Figure S40. *Pan*-specific duplication of *EIF3C* locus, and bonobo-specific deletion of *SAMD9*. HiFi read depth and WSSD of bonobo, chimpanzee, orangutan, gorilla, and human individuals relative to GRCh38 detect these events (above), which are validated by interphase FISH of each species using fosmid clones spanning the region (below).

Because of our discovery of a chimpanzee/bonobo expansion of the *EIF4A3* gene family, we focused on the *EIF3C* gene family expansion confirmed by FISH in both chimpanzee and bonobo. Unlike the *EIF4A3* gene family which expanded in tandem, this locus expanded in an interspersed fashion along the short arm of chromosome XVI (phylogenetic group chromosome 16) likely as a result of its association with *NPIP*. We performed a similar phylogenetic reconstruction (see *EIF4A3* above) and found that while the initial duplication of this locus occurred ~5.01 mya, subsequent duplications occurred independently in the bonobo and chimpanzee lineage (<1.5 mya) (Supplementary Note Figs. S44 and S45).

Supplementary Note Figure S44. *EIF3C* primate phylogeny. A phylogenetic tree was constructed from 16 kbp neutral sequences of *EIF3C* paralogs using Bayesian phylogenetic inference. This analysis is conducted using BEAST2 software. Bolded numbers on each major node denote estimated divergence time. The blue error bar on each node indicates 95% confidence interval of the age estimation. Bootstrap supports are reported using asterisks for nodes with posterior probability >99%.

Supplementary Note Figure S45. EIF3C coding variation. Gene models for transcribed loci based on Iso-Seq data (above). Human *EIF3C* and *EIF3CL* are compared to predicted open reading frames for bonobo paralogs and Liftoff gene predictions for chimpanzee, orangutan, and gorilla paralogs from contigs assembled from HiFi reads.

In addition to including these findings in the supplementary material, we added a paragraph describing our more comprehensive analysis of SDs and incorporated the most interesting observations regarding the *EIF4A3* locus into the main text:

“We focused on creating a more comprehensive list of gene family expansions and contractions specifically in the *Pan* lineage by reconciling counts from the genome assembly first by sequencing read-depth estimates (Supplementary Tables 11 and 12) and then following up with experimental validation by FISH. A complete list of genes expanded and contracted in bonobo compared to other apes based on sequence read depth is provided, including 10 gene families predicted to have expanded specifically in bonobo since its divergence from chimpanzee (Supplementary Tables 13-15). We tested by FISH 20 SDA-positive regions containing genes and confirmed their duplication status in the bonobo genome as gene family expansions (Supplementary Table 16). Similarly, we validated bonobo-specific gene-family contractions (*IGFL1*, *TRAV4K*, *CDK11A*) and more ancient duplications common to both chimpanzee and bonobo (e.g., *CLN3*, *EIF3C*, *RGL4*). These bonobo-contracted gene families show some GO enrichment for genes related to maturity onset diabetes of the young (Supplementary Note Table S47). It is noteworthy that three out of four tested gene expansions show patterns of intrachromosomal interspersions and these are found adjacent to “core duplicons” (e.g., *NPIP* and *GUSBP*) that have been predicted to mediate the formation of interspersed SDs independently in humans⁴⁶. Among the Pan-specific Eukaryotic Translation Initiation Factor 4 Subunit A3 (*EIF4A3*) gene family, there is evidence that five out of the six paralogs are expressed and encode a full-length open reading frame (ORF; Fig. 3). Based on our assembly of the locus in both chimpanzee and bonobo, we show that the initial *EIF4A3* gene duplication occurred in the ancestral lineage approximately 2.9 mya. It then subsequently expanded and experienced gene conversion events independently in the chimpanzee and bonobo lineages creating five and six copies of the *EIF4A3* gene family, respectively. In both lineages, the gene families are organized head-to-tail in direct orientation. Interestingly, some of the gene conversion signals correspond to a set of specific amino-acid changes in the basic ancestral structure that are now common to

only chimpanzee and bonobo. Each of the chimpanzee specific copies carries an 18 bp VNTR motif in the 5' UTR, which frequently differs among the copies.”

* Prufer 2012 used the bonobo genome to show evidence of chimpanzee selective sweeps in the MHC locus and other regions. They also show that MHC is the most frequent target of ILS. But there is no mention of MHC in this work, which is surprising given these previous findings and how important MHC is in human biology. Can the authors revisit this analysis using the new assembly? Is the previous signal missing? Can the authors confirm / revise the prior findings?

We revisited the MHC locus and performed a much more detailed analysis with a specific focus on evidence of selection between our study and Prufer (PMID: 22722832). We began by first comparing the degree of completion in this region and found 291 gaps in the Prufer assembly (red bars Supplementary Note Figure S65) versus two gaps in the Mhudiblu assembly (purple bars).

Supplementary Note Figure S65. Dot matrix comparison of MHC region. The MHC region of the Mhudiblu_PPA_v0 bonobo assembly compared with the panpan1.1 bonobo assembly from Prufer et al. (2012). The current bonobo assembly contig gaps are shown along the x-axis in purple. The Prufer et al. (2012) assembly is represented along the y-axis, with the contig gaps shown in red. In the MHC region, there are two gaps in the Mhudiblu_PPA_v0 assembly and 291 gaps in the Prufer et al. (2012) assembly. Alignment between the two genomes is represented in blue with each dot representing 1 kbp of alignment.

As expected, we observed strong signals of balancing selection (Tajima's D values for the two significant 10kbp windows chr6:32650000-32660000 and chr6:32660000-32670000 are 2.89 and 3.10, respectively) and clustered ILS of various topologies across multiple regions within the MHC locus (Supplementary Note Fig. S66). These findings are generally consistent with previous reports from Prufer and colleagues. The strongest signals were observed for bonobo orthologs of the MHC genes (*HLA-DPA1* and *HLA-DP2*).

a

b

Supplementary Note Figure S66. Ideogram of the MHC region with ILS annotations. a, The four main ILS topologies are color-coded below. The four color lines representing ILS segments are shown above the chromosome coordinate (hg38). The clustered ILS are shown above the four color lines (black). The MHC region (red bar) corresponds to genomic coordinates chr6:28510120-33480577. **b,** A zoomed-in view of the MHC region (chr6:32786501-33103000) depicts clustered ILS nearby *HLA* genes.

The previous study, however, previously reported regions of reduced diversity in bonobo based on a comparison to chimpanzee. We do not find compelling evidence that these sites are under positive selection based on SweepFinder2 or Tajima's D analyses. We further followed this up by directly comparing the genetic diversity (π) bonobo vs chimpanzee. With one exception, we observed no regions of significantly reduced diversity. The one exception where both chimpanzee and bonobo show a reduction of single-nucleotide polymorphisms (SNPs) corresponds to a segmental duplication (chr6: 26666991-27002570) where SNPs were removed in our VCF due to paralogy. Overall SNP diversity is reduced across the region in bonobo when compared to chimpanzee and there are five regions (red arrows) (Supplementary Note Figure S67) where diversity is the greatest between chimpanzee and bonobo. Three of these correspond to regions identified by Prufer, however, they are not among the top 1% of genome windows showing positive selection.

Supplementary Note Figure S67. Chimpanzee versus bonobo MHC upstream diversity. **a**, Nucleotide diversity of bonobo (green) and chimpanzee (blue) are shown based on human genomic coordinates (hg38, chr6:25000000-29000000). The mean (dashed line) is shown for bonobo (mean=4.45e-4) and chimpanzee (mean=9.35e-4). A region of reduced diversity (gray) is shown but corresponds to an SD where SNPs were excluded due to potential mismapping. **b**, Same as (a) but merged onto the same scale and highlighting five regions (red arrows) where diversity is reduced in bonobo when compared to chimpanzee. Three of these correspond to regions identified by Prufer; however, they are not among the top 1% of genomes candidates showing positive selection by Tajima's D and SweepFinder2. Overall SNP diversity is reduced across the region in bonobo when compared to chimpanzee.

We include this detailed analysis of the MHC in the main and Supplementary Note and add a note to the discussion regarding the comparisons between the Prufer and our new genome assembly.

“In the case of major histocompatibility complex, we effectively close 289 of the 291 gaps across this region and, while we still detect strong signatures of balancing selection, we no longer identify potential selective sweeps in the top 1% of regions as reported previously¹ (Supplementary Note Figs. S65-67).”

* Could the Dn/Ds-high ILS clusters be the result of missassemblies? The authors should demonstrate that ILS high vs poor regions have the same degree of assembly quality.

To address this question, we computed the sequence accuracy in ILS regions compared to the entire genome (Mercury QV estimation based on Illumina WGS data). ILS regions are estimated to be more accurate than the genome-wide average and clustered and non-clustered sites show comparably degrees of assembly quality so it is unlikely that the dN/dS difference observed for clustered sites is due to misassembly (Table R3-3).

Table R3-3. Comparison sequence accuracy of clustered vs nonclustered ILS regions

	QV	number of regions	number of bases
clustered human-bonobo ILS regions	47.5	31,134	15,768,607
unclustered human-bonobo ILS regions	44.6	76,74	38,901,166
clustered human-chimp ILS regions	46.7	31,576	15,962,619
unclustered human-chimp ILS regions	43.6	77,587	39,375,096
entire bonobo assembly	39.2	1	3,015,333,734

Minor critiques

* Supp Note Table S35 uses commas instead of periods for decimals.

We thank the reviewer for pointing out this discrepancy; it has now been resolved (the table is now Supp Note Table S38).

Supplementary Note Table S38. The functional annotation clustering of genes that contain at least two exons under ILS

	Functional cluster	Enrichment Score	P_Value
Human-bonobo ILS (n=40)	Photoreceptor activity	2.28	1.60E-04
	Glycoprotein	1.57	9.60E-03
	Proteinaceous extracellular matrix	1.3	1.10E-02
	Lysosome	1.28	1.20E-02
	Cytoskeleton	1.2	3.40E-02
Human-chimpanzee ILS (n=44)	EGF-like	3.73	1.90E-06
	Disulfide bond	3.38	9.20E-06
	CUB domain	3.14	1.90E-04
	Transport	2.39	1.30E-03
	Sushi	2.18	5.60E-03
	Transmembrane region	1.7	2.40E-03
	Hypertrophic cardiomyopathy (HCM)	1.26	1.20E-02
	Calmodulin-binding	1.25	6.10E-04
Union human-chimpanzee and human-bonobo ILS (n=143)	Glycoprotein	4.67	6.10E-10
	CUB domain	3.83	4.60E-05
	EGF-like domain	3.13	5.80E-06
	domain:VWFC 2	2.58	2.80E-04
	Sushi	2.49	5.90E-04
	Myosin tail	2.34	3.20E-04
	Calmodulin-binding	2.27	6.40E-04
	ATP binding	2.24	9.90E-04
	actin filament binding	1.69	2.70E-03
	Peptidase M12B, ADAM/reprolysin	1.67	2.90E-03
	C-type lectin fold	1.5	1.30E-03

Referee #4 (Remarks to the Author):

I would like to congratulate Catacchio et al., for presenting a new high-quality Bonobo genome and for treating the analysis and the presentation of the results **with so much rigor and care**. The manuscript presents a chromosome level genome for *Pan paniscus* – the last of the great apes to be sequenced with long-reads – where a great portion of the gaps were closed and genes were fully annotated, and half of the segmental duplications were assembled as well. They have also presented a new set of bonobo exclusive genes, have described novel gene models in the bonobo assembly thought to be related exclusively to human adaptation and have done all of this research taking into consideration IsoSeq sequencing for confirmation to these new findings. A number of segmental duplications and the chromosome fusion were further tested and confirmed with FISH experiments which brings great confidence to these findings. The work also presented a higher resolution analysis of ILS showing that a greater fraction of the hominid genome is under ILS, unlike what was estimated previously. Because of all that stated above, I consider **that this work is innovative and presents a rich resource for experimental biologists who will have plenty of material to target novel genomic areas and further advance our understanding of hominid evolution and gene function.**

We thank the reviewer for appreciating the value of this work. All responses to remaining comments are discussed point-by-point below

In terms of the genome assembly – which is my main area of expertise – one point that concerns me a little is the QV ranging from 35-39 (estimated by kmer-sharing and BACs sequence comparisons). The truth is that technologies evolve, and it is likely that a 30x coverage of Pacbio HiFi would be able to take the QV to >50 and would most likely solve the remaining unresolved Segmental Duplications of this assembly. The same is true considering Hi-C reads that – particularly if sequenced from the same individual – would have high resolution to determine unconfirmed internal structure. That said, because the authors had extreme care with their claims and the genome presented is a huge improvement over the last one available, this genome should be available to the scientific community as it is and it supports the claims made by the authors.

This is a good point. To improve the quality of our assembly, since our original submission, we generated an additional 40-fold HiFi sequence data by circular consensus sequencing from the same source genome (Mhudiblu) and used this to further correct remaining sequencing errors. We used Racon (two rounds) to error correct the genome eliminating ~128,000 remaining errors for an overall accuracy 1 error every 12,882 bp (improving QV from 39 to 41.1). This improved quality assembly is being released as Mhudiblu_PPA_v2 (Accession number pending).

I would like to advise the authors, however, to have a look at the .bed intermediate output of merquy. This file contains the coordinates of kmers present only in the assembly, meaning they are not shared with the Illumina reads. As the authors have done so much already, it would not be too much trouble to estimate if these unique-assembly kmers are more frequent in specific genomic areas such as repeats.

Based on the reviewer's suggestions, we intersected the .bed intermediate output of merquy (which contains these unique assembly k-mer (UAK)'s) with Rpmasker bed file, SD bed file as well as GC content. We determined that 61% of unique assembly k-mers (4.6 out of 7.5 million) map to common repeats with 3.1 million mapping to segmental duplications. 68% (5.1 million) map to unplaced contigs which are largely composed of heterochromatic DNA. Thus, nearly all the assembly-only kmers map to repetitive DNA (Fig. R4-1). Most of the UAKs do not map to

chromosomes but instead are assigned to the unplaced chromosomes (5.1 million or 68%) consistent with their heterochromatic and repetitive content (Fig. R4-1).

Fig. R4-1. Genomic distribution of unique assembly kmers (or UAK) present in assembly but not supported by Illumina WGS from the same reference source (Mhudiblu). “GC”: GC content of k-mer is

<20% or >80%. “Seg dup”: the k-mer intersects a segmental duplication (according to WGAC) with identity $\geq 90\%$ and length ≥ 1 kbp. “Repeat”: the k-mer intersects a repeat. 1,019,180 (13.5%) of the UAKs map to unique regions of the genome without extreme GC content.

Further, it would be important to check if those possibly-erroneous-assembly-kmers are present in any of the 111 genes that have potential frameshifting indels that disrupt their primary isoform relative to the human reference. In addition, I would like to see a supplementary figure with the kmer plot distribution of the illumina reads used for the short-reads polishing and for the merqury QV estimation.

This was a good suggestion. We performed the comparison of the unique kmers not supported by Illumina WGS and identified 4 genes (corresponding to 17 distinct events) in the set of genes where frameshifts were detected (Table R4-1). Since these represent duplicate genes, we restricted our final report to those frameshift events identified in unique genes, validated by HiFi, and by genotyping in a population of bonobo samples (see above).

We also revised the Merqury k-mer distribution plots of the Illumina reads used for the short-read polishing and for the Merqury QV estimation as a supplementary figure (Supplementary Note Figure S11).

Supplementary Note Figure S11. Merqury k-mer distribution of bonobo assembly. Merqury was run on bonobo genome assembly Mhudiblu_PPA_v0 with the Illumina reads used to polish the assembly. The number of distinct Illumina k-mers (“Count”) is compared against its occurrence in Illumina WGS (“kmer multiplicity”). Colored lines indicate the number of times a k-mer is found within the assembly. The black line indicates k-mers unique to Illumina WGS. The blue and red boxes (at kmer_multiplicity = 0) indicate unique assembly k-mers (UAK) not found in the Illumina reads.

Table R4-1. The bonobo-specific frameshifts intersecting UAKs from Merqury.

bonobo_indel_coords	human_indel_coords	source_gene	indel type	overlap SD (WGAC)	fixed bonobo
002331F_46097_qpd_scaf:45013-45013	chr2:102338272-102338273;chr2:102338270-102338271	ENSG00000115602.17	CodingDeletion	yes	NA
chr12:8354470-8354470	chr11:71799562-71799563	ENSG00000158483.16	CodingDeletion	yes	NA
chr12:8354476-8354476	chr11:71799561-71799562	ENSG00000158483.16	CodingDeletion	yes	NA
chr20:28615808-28615808	chr20:32326771-32326773	ENSG00000101350.8	CodingDeletion	yes	NA
chr20:28615816-28615816	chr20:32326781-32326783	ENSG00000101350.8	CodingDeletion	yes	NA
chr20:28615818-28615819	chr20:32326783-32326785	ENSG00000101350.8	CodingInsertion	yes	NA
chr20:28615839-28615839	chr20:32326804-32326806	ENSG00000101350.8	CodingDeletion	yes	NA
chr20:28616657-28616657	chr20:32327611-32327612;chr20:32327613-32327614	ENSG00000101350.8	CodingDeletion	yes	NA
chr20:28616692-28616692	chr20:32327647-32327649	ENSG00000101350.8	CodingDeletion	yes	NA
chr20:28619165-28619165	chr20:32330151-32330152;chr20:32330153-32330154	ENSG00000101350.8	CodingDeletion	yes	NA
chr20:28619227-28619227	chr20:32330212-32330214	ENSG00000101350.8	CodingDeletion	yes	NA
chr20:28619314-28619314	chr20:32330297-32330299	ENSG00000101350.8	CodingDeletion	yes	NA
chr20:28619320-28619320	chr20:32330304-32330306	ENSG00000101350.8	CodingDeletion	yes	NA
chr20:28619327-28619327	chr20:32330312-32330313;chr20:32330314-32330315	ENSG00000101350.8	CodingDeletion	yes	NA
chr20:28620278-28620278	chr20:32331283-32331285	ENSG00000101350.8	CodingDeletion	yes	NA
chr20:28620282-28620282	chr20:32331288-32331289;chr20:32331290-32331291	ENSG00000101350.8	CodingDeletion	yes	NA
chr5:739670-739670	chr5:677939-677941	ENSG00000171368.12	CodingDeletion	yes	NA

My two last considerations would be to (i) ask the authors to confirm they have checked that the further curated Mhudiblu_PPA_v1 version of the assembly has not disrupted any genes that the authors have investigated in Mhudiblu_PPA_v0 and described in their results.

We performed this analysis. We checked that the Mhudiblu_PPA_v1 version of the assembly had not disrupted any of the genes investigated in Mhudiblu_PPA_v0: in the RefSeq gene set, two putative genes of unknown function were, in fact, disrupted (gene_id: LOC117980845, LOC100977127); in the final CAT gene set seven genes were interrupted (gene_id: Bonobo_T0015403, Bonobo_T0015688, Bonobo_T0026896, AC136431.2-201, Bonobo_T0078976, Bonobo_T0091676, PMS2CL-204)]. Most of these “broken gene models”,

with the exception of Bonobo_T0026896 or ASH2, do not have strong support and were novel predictions based solely on Augustus PB. We include these updates in the supplementary text.

Please see Supplementary Note section 4.4

And on that point, I would suggest the authors to maybe (ii) include a last supplementary figure representing a genome assembly fluxogram – going from the Falcon assembly, pointing out the manual interventions, annotations and further improvements all the way from reads to Mhudiblu_PPA_v0 and Mhudiblu_PPA_v1. The supplementary material presented is already a great documentation for reference, but it is extensive. This added fluxogram would be a good historical reference of the steps taken to assemble this version of the bonobo genome, and would greatly help future scientists who will be looking to further improve this assembly to find the regions more likely to contain errors.

Thank you for this suggestion. We created a fluxogram including our additional polishing using HiFi data. It is included in the text to provide an overview of the steps associated with process of initial contig assembly (v0), order and orientation (v1) and polishing (v2) (Supplementary Note Fig. S26).

Supplementary Note Figure S26. Processing steps to create the reference sequences Mhudiblu_PPA_v0, Mhudiblu_PPA_v1, and Mhudiblu_PPA_v2.

Once more, I congratulate the authors in their great effort and relevant piece of science presented. I wish them success

Thank you again.

Reviewer Reports on the First Revision:

Referee #1 (Remarks to the Author):

This manuscript is a revised version of a paper reporting a new whole genome reference assembly for bonobo (*Pan paniscus*). This is a valuable information resource for genetic and genomic analyses, and the authors have also performed a range of analyses comparing this genome to other apes and to humans. I find the changes made to the manuscript in response to the first round of reviews to be extensive and satisfactory. The authors have quite adequately addressed all my specific concerns. I further believe the authors have provided satisfactory responses to the critiques from the other reviewers.

Referee #2 (Remarks to the Author):

The authors responded to reviewer comments in a very thorough manner. The results reported in the manuscript have now been examined in acceptable detail, with logical gaps filled, and some new insights have certainly become apparent through this process. The results will be quite valuable to several different research communities. Are the results particularly striking or transformative in terms of biological knowledge? While the answer to that question for me is still no, I am at least now satisfied by the quality and depth of analyses supporting the results that are presented. I probably did envision a more comprehensive revision to the direction of the main manuscript text than what was resubmitted, but I recognize that the manuscript that I would write is not necessarily the one that the authors would. Whether the overall significance of the results now passes the threshold for acceptance is an editorial decision.

Referee #3 (Remarks to the Author):

The authors have fully addressed all of my critiques. This includes substantial additional analyses and very interesting findings on ILS and selection. Overall the work is rigorous and fascinating. Well done

Referee #4 (Remarks to the Author):

I would like to congratulate Catacchio et al., once again for this rigorous and relevant piece of work. They have responded well to all my concerns. They've improved the genome accuracy by sequencing an extra 40-fold of Pacbio HiFi data, which eliminated ~128,000 remaining errors in the genome and lead to a final version Mhudiblu_PPA_v2 to be released. They've also made an analysis of remaining assembly-only kmers (likely to be errors) and concluded that most of them are placed on repeats, which is good to see and further guarantees the quality of the assembly. In addition, they revisited genes with potential frameshifting indels using HiFi data and revised their list of those genes, guaranteeing that frameshift events identified are most reliable.

The assembly is of high-quality – it includes different data types and methods to validate finds. This genome will greatly benefit further studies into hominid evolution. The authors present a thorough analysis of ILS among chimpanzee, human and bonobo almost doubling previous estimates. Their data also shows that a portion of the genomic regions subjected to ILS are not randomly distributed and exons on those regions show elevated rates of amino acid replacement. I can't wait to see this paper out there to continue this discussion with the wider scientific community. I advise the acceptance of the current version of the manuscript.

Dr Marcela Uliano da Silva